# AdamO: A Collapse-Suppressed Optimizer for Offline RL

**Nan Qiao** [1 2]   **Sheng Yue** [3]   **Shuning Wang** [1 2]   **Ju Ren** [1]

## Abstract

Offline reinforcement learning (RL) can fail spectacularly when bootstrapped temporal-difference (TD) updates amplify their own errors, driving the critic toward extreme and unusable Q-values. A key counterintuitive insight of this work is that collapse is not only a property of the backup rule or network architecture: optimizer dynamics themselves can directly trigger or suppress instability. From a control-theoretic viewpoint, we model offline TD learning as a feedback system and analyze Adam-based critic updates. This yields a necessary and sufficient condition for stability of the induced local update dynamics: within the regime we analyze, these dynamics are stable if and only if the spectral radius of the corresponding update operator is strictly below one. Further analysis suggests that standard Adam updates can inadvertently distort the parameter geometry, motivating explicit orthogonality constraints to prevent TD error amplification. To this end, we propose *AdamO*, an Adam-based optimizer with a decoupled orthogonality correction regulated by a strict task-alignment budget. We prove that this design theoretically guarantees worst-case task safety and preserves Adam's continuous-time dissipative dynamics. Empirically, AdamO is broadly compatible with diverse offline RL baselines, improving stability and returns across a broad suite of benchmarks, and our project page is publicly available at https://jo-nan.github.io/AdamO/.

## 1. Introduction

Offline reinforcement learning (RL) aims to learn from fixed datasets without interaction, typically addressing severe distribution shift through mechanisms such as explicit policy constraints, conservative value regularization, or uncertainty-driven pessimism. (Tarasov et al., 2023; Lyu et al., 2022; An et al., 2021). Despite recent progress, the critic learning remains the main bottleneck, as temporal difference (TD) learning dynamics can trigger the deadly triad[1], leading to instabilities and ultimate *value collapse*. (Baird, 1995; Chen et al., 2023; Kumar et al., 2022).

Empirically, this value collapse is typically preceded by a breakdown of the TD learning signal itself (Tarasov et al., 2022; Sun, 2023). Once the bootstrapped target becomes an unstable feedback path, TD errors can be amplified rather than corrected, and the critic is pushed toward extreme and eventually unusable Q-values (Kumar et al., 2022). A traditional way to stabilize deep Q-learning is to weaken or delay error propagation using target networks and Double-Q style updates (Hasselt, 2010; Fujimoto et al., 2018). More recently, empirical evidence suggests that architectural choices, especially normalization, can substantially improve stability in both online and offline settings (Bhatt et al., 2019; Nikulin et al., 2022; Ball et al., 2023; Kang et al., 2023; Kumar et al., 2023; Yue et al., 2023). At the same time, a growing line of work studies collapse more directly by attributing it to unstable generalization dynamics or harmful correlations introduced by squared TD objectives, proposing remedies through auxiliary regularization (Qiao et al., 2026b; Kumar et al., 2022). However, these methods serve primarily as mitigation rather than a cure, as they treat the optimization mechanism as a black box, leaving the critical interplay between optimizer dynamics and the TD objective unaddressed. In this paper, we ask the following question: what are the *necessary and sufficient conditions* for TD error collapse from the perspective of optimization dynamics, and how do modern optimizers like Adam (Kingma & Ba, 2017) interact with bootstrapping to trigger or suppress such collapse?

To this end, we study critic collapse through a control theory viewpoint by treating offline TD learning as a feedback system, where the learner is trained on its own bootstrapped predictions without external correction. Focusing on the optimizer that is used in practice, we analyze the critic up-

---

[1]Tsinghua University, China [2]Central South University, China [3]Sun Yat-sen University Shenzhen Campus, China. Correspondence to: Sheng Yue <yuesh5@mail.sysu.edu.cn>.

*Proceedings of the 43rd International Conference on Machine Learning*, Seoul, South Korea. PMLR 306, 2026. Copyright 2026 by the author(s).

---

[1]The deadly triad describes the combination of function approximation, bootstrapping, and off-policy training, which can lead to divergent value estimates in RL (Van Hasselt et al., 2018).

dates under Adam (Kingma & Ba, 2017; Robbins & Monro, 1951). We derive a closed-form linear recurrence for the TD error and prove a necessary and sufficient characterization of stability: *learning critic is stable if and only if the spectral radius of an explicit augmented update matrix is strictly below one.* This yields a simple mechanism: value collapse occurs exactly when bootstrapping turns into positive feedback that amplifies TD errors faster than the update dynamics can "suppress" them.

This characterization also exposes which design knobs can suppress collapse and why. We derive a tractable sufficient condition for the Hurwitz requirement that separates two effects: a bootstrapped scale term and a geometric term quantifying how severely the parameter mapping deviates from isometry. The scale term can be controlled by standard normalization practices, while the geometric term is directly governed by the properties of layer-wise weights, motivating parameter orthogonality as a principled way to prevent error amplification. Crucially, standard loss-based penalties inevitably corrupt Adam's adaptive moment estimates, so orthogonality should be imposed as a decoupled correction at the optimizer level. Guided by this insight, we propose AdamO, an Adam-based optimizer that adds a conservative orthogonality correction on selected layer-wise weight matrices, limits its interference with task descent through a per-layer budget, and keeps the correction magnitude bounded relative to the Adam step. AdamO reduces to Adam when the orthogonality strength is set to zero.

We provide theory that matches this design at the level of optimization dynamics. In a conflict-free mode where the orthogonality correction is not allowed to oppose the task gradient, AdamO is guaranteed to not increase the next step task loss compared with Adam under a standard smoothness condition and an explicit step size requirement. When a positive budget is allowed, exact next step non-inferiority cannot hold in general, and we instead derive an explicit upper bound that quantifies the worst case single step degradation as a function of the allowed conflict and the correction magnitude. We further relate these regimes to a continuous time energy interpretation of Adam, where the conflict-free mode preserves monotone decrease and the budgeted mode yields a quantified relaxation. Empirically, these theoretical guarantees translate into robust practice: simply replacing the critic optimizer with AdamO effectively suppresses value collapse across diverse D4RL benchmarks. As a result, AdamO achieves higher returns than both standard optimizers and stability-oriented alternatives across a broad suite of algorithms, as detailed in Table 1.

## 2. Related Work

**Offline reinforcement learning.** Offline reinforcement learning fundamentally aims to extract optimal policies from fixed, pre-collected datasets without the ability to interact with the environment for correction (Qiao & Yue, 2026; Lillicrap et al., 2015; Fujimoto et al., 2019; Qiao et al., 2026a). To handle the inevitable distribution shift between the learned policy and the static dataset, the field has largely converged on methods utilizing policy constraints (Tarasov et al., 2023; Wu et al., 2019; Nair et al., 2020), conservative value regularization (Kumar et al., 2020; Kostrikov et al., 2022; Lyu et al., 2022), or uncertainty-driven pessimism (Qiao et al., 2025; An et al., 2021). However, the core challenge in these off-policy algorithms often stems from the instability of the value function itself (Kumar et al., 2019; Chen et al., 2023). Specifically, when combined with bootstrapping and deep neural function approximation, offline learning becomes highly susceptible to the "deadly triad," a phenomenon characterized by unbounded value divergence and maximization bias (Sutton & Barto, 2018; Baird, 1995; Tsitsiklis & Van Roy, 1996; Van Hasselt et al., 2018). Beyond standard control benchmarks, learning or post-training from fixed high-stakes datasets has also been explored in medical AI and language-model adaptation, including LLM-guided prototype refinement, medical vision-language model post-training for diagnosis, collaborative fine-tuning between small and large language models, and security risks in fine-tuned generative language models (Zhu et al., 2025; 2026a; Lin et al., 2026; Zhu et al., 2026b; Ma et al., 2026a;b). These works are complementary to our focus, as we study optimizer-induced TD feedback instability in offline RL rather than domain-specific semantic reasoning or visual-language alignment.

**Value-function collapse in reinforcement learning.** To counteract such instability in deep Q-learning, standard protocols have historically relied on heuristic mechanisms like separate target networks and Double-Q learning to stabilize error propagation (Hasselt, 2010; Fujimoto et al., 2018). Beyond these algorithmic adjustments, recent empirical work suggests that architectural interventions, particularly normalization techniques like CrossNorm (Bhatt et al., 2019) and LayerNorm, can significantly enhance training stability in both online and offline settings (Nikulin et al., 2022; Ball et al., 2023; Kang et al., 2023; Kumar et al., 2023). While these methods provide empirical relief, attention has recently shifted toward understanding the underlying mechanics of why representations degrade. $C^4$ attributes collapse to harmful TD cross covariance effects induced by the squared TD loss, and proposes controlling this structure via clustered replay and regularization (Qiao et al., 2026b). DR3 highlights an implicit regularization effect that aligns features across backup pairs and can collapse representations, and counteracts it with an explicit feature similarity penalty (Kumar et al., 2022). Conversely, Yue et al. (2023) analyze divergence as a self excitation feedback in Q updates, introduce an NTK based predictor of divergence, and

empirically show that LayerNorm can suppress collapse. However, these approaches generally rely on auxiliary regularization or architectural modifications without identifying the necessary and sufficient conditions for value collapse. Crucially, they overlook the potential of suppressing these instabilities directly from the optimizer.

**Optimizers for reinforcement learning.** Stochastic gradient descent (SGD) is the classical stochastic-approximation workhorse for learning from noisy gradients (Robbins & Monro, 1951), and modern deep RL commonly adopts adaptive first-order methods such as Adam (Kingma & Ba, 2017) to train both policy and value networks. Beyond generic first-order updates, ACKTR performs scalable trust-region / natural-gradient optimization in actor-critic methods via a Kronecker-factored curvature approximation (Wu et al., 2017). Motivated by the non-stationary and bootstrapped nature of RL objectives, Asadi et al. (2023) show that reusing Adam-style moment statistics across rapidly shifting loss landscapes can be harmful and propose resetting optimizer states to improve value-based deep RL stability, while TRAC designs a parameter-free optimizer inspired by online convex optimization to improve adaptation under continual distribution shifts and mitigate plasticity loss (Muppidi et al., 2024; Wang et al., 2026). In parallel, learned-optimizer approaches aim to meta-learn update rules specialized to RL dynamics, including Optim4RL (Lan et al., 2024) and OPEN (Goldie et al., 2024). Most recently, Stable Gradients studies performance collapse when scaling deep RL and proposes gradient-flow interventions, including a Kronecker-factored optimizer (Kron), to stabilize training at large depth/width (Castanyer et al., 2025). Feedback-driven optimization has also been explored in generative and agentic visual systems (Liang et al., 2026a;b; 2025a). However, these optimizer-centric studies largely improve stability through heuristic interventions (e.g., curvature approximations, state resets, or meta-learned updates) and do not characterize the concrete conditions that trigger *value-function collapse* in offline RL. In contrast, we identify a necessary and sufficient condition for collapse, derive a tractable sufficient condition, and enforce it through an Adam-style modification to suppress collapse in the offline setting.

## 3. Preliminary

**Offline RL.** We consider a fixed offline dataset given by $\{(s_i, a_i, s_{i+1}, r_i)\}_{i=1}^M$. Define the input set by $X = \{x_i = (s_i, a_i)\}_{i=1}^M$ and the reward vector $r = [r_1, \ldots, r_M]^\top$. Let $Q_\theta(\cdot)$ denote the network output parameterized by $\theta \in \mathbb{R}^P$, evaluated on a finite set of inputs and stacked into a vector in $\mathbb{R}^M$. For instance, $Q_\theta(X)$ is the vector of Q-values on the dataset pairs. The discount factor is $\gamma \in (0, 1)$. To cover both on-policy and off-policy one-step TD up-

dates, we introduce a target action selection rule at the next state. For each iterate $\theta_t$, define the greedy policy $\hat{\pi}_{\theta_t}(s) = \arg\max_a Q_{\theta_t}(s, a)$. In addition, when the offline data are provided as trajectories, we let $\tilde{a}_i$ denote the behavior action taken at the next state $s_{i+1}$ in the dataset, i.e., the action following $s_{i+1}$ along the same trajectory. We then define, for each transition $i$, a target next-action $a'_{i,t}$, e.g.,

$$a'_{i,t} = \begin{cases} \hat{\pi}_{\theta_t}(s_{i+1}), & \text{(Q-learning / off-policy TD),} \\ \tilde{a}_i, & \text{(SARSA / on-policy TD),} \end{cases} \quad (1)$$

and form the corresponding target set $X'_t = \{x'_{i,t}\}_{i=1}^M$ where $x'_{i,t} = (s_{i+1}, a'_{i,t})$. We then define the TD error:

$$\mathbf{e}_t = Q_{\theta_t}(X) - \left( r + \gamma Q_{\theta_t}(X'_t) \right), \quad (2)$$

and the corresponding step-dependent squared TD loss

$$L_t(\theta) = \frac{1}{2} \left\| Q_\theta(X) - \left( r + \gamma Q_{\theta_t}(X'_t) \right) \right\|_2^2. \quad (3)$$

Note that the target term uses $\theta_t$,[2] matching the usual semi-gradient TD objective. The above formulation specializes to Q-learning and SARSA under the respective choices of $a'_{i,t}$.

**Adam optimizer.** In the context of RL, Adam is arguably the most widely used optimizer due to its adaptive learning rate properties (Sokar et al., 2023; Fujimoto & Gu, 2021; Kostrikov et al., 2022). In our standard implementation, let $g_t = \nabla_\theta L_t(\theta_t)$ denote the gradient of the loss function with respect to the parameters $\theta$ at step $t$. The algorithm maintains exponential moving averages of the gradient's first and second moments, $m_t$ and $v_t$, to adaptively scale the updates. These moments are updated as:

$$m_t = \beta_1 m_{t-1} + (1 - \beta_1)g_t, \quad (4)$$

$$v_t = \beta_2 v_{t-1} + (1 - \beta_2)g_t^{\odot 2}, \quad (5)$$

where $g_t^{\odot 2}$ represents the element-wise square of the gradient. To counteract initialization bias, these estimates are corrected as $\hat{m}_t = m_t/(1 - \beta_1^t)$ and $\hat{v}_t = v_t/(1 - \beta_2^t)$. Consequently, the parameters are updated via the adaptive diagonal preconditioner $D_t = \text{Diag}\left(1/(\sqrt{\hat{v}_t} + \epsilon)\right)$ according to:

$$\theta_{t+1} = \theta_t - \eta D_t \hat{m}_t, \quad (6)$$

where $\eta$ is the learning rate and $\epsilon$ is a small constant for numerical stability (Kingma & Ba, 2017).

**Neural Tangent Kernel.** To analyze the optimization dynamics in the function space, we utilize the Neural Tangent Kernel (NTK) perspective (Jacot et al., 2018). Intuitively, the NTK characterizes the similarity between inputs

---

[2]We do not backpropagate through the bootstrapped target.

through the lens of the parameterized network, where the Jacobian matrix serves as the feature extraction map. Let $Z_t(X) = \nabla_\theta Q_{\theta_t}(X) \in \mathbb{R}^{P \times M}$ denote the Jacobian of the $Q$-function. Consider the dynamics of the network output under a parameter shift $\Delta\theta$. By applying a first-order Taylor expansion, the change in prediction can be approximated as: $\Delta Q_\theta(X) \approx Z_t(X)^\top \Delta\theta$. This expansion suggests that the evolution of the function values depends heavily on the alignment of the gradients. When considering gradient-based optimization, this interaction is governed by the correlation between these gradients. Consequently, the dynamics are summarized by the Gram matrix, which is defined as the inner product of the Jacobians:

$$G_t(X) = Z_t(X)^\top Z_t(X). \qquad (7)$$

This matrix $G_t$ captures the pairwise similarity and dictates the convergence properties by characterizing how modifications in the parameter space translate to updates in the function space (Yue et al., 2023).

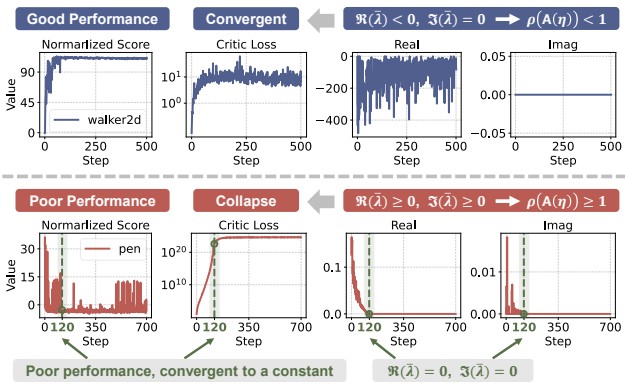

*Figure 1.* Convergence vs. collapse of critic loss using TD3+BC. **Top:** On the *walker2d-medium-expert* task, the update operator satisfies the contraction condition $\rho(\mathsf{A}(\eta)) < 1$. This suppresses bootstrapping errors, which maintains a convergent critic loss. **Bottom:** On the *pen-human* task, the expansive regime $\rho(\mathsf{A}(\eta)) > 1$ triggers a value explosion (critic loss collapse). The green vertical line highlights the critical boundary state where $\rho(\mathsf{A}(\eta)) = 1$, leading to collapse stagnation and resulting in poor performance.

## 4. A Linearized Divergence Criterion for Adam in TD Learning

We further develop a local, operator-level stability and divergence criterion for applying Adam to the semi-gradient squared TD objective introduced in the preliminary section. To keep the main text concise, we present a simplified operator-level derivation here and defer the full technical development to Appendix D. We rely on standard local analysis assumptions (detailed in Appendix B.1), including: the stepsize $\eta$ is small enough to permit a first-order Taylor approximation, the action gap is sufficient to keep the greedy policy stable, and the process has entered a "terminal phase." In this phase, the bootstrapped targets $X'$, the network Jacobians $Z(\cdot)$, and the Adam preconditioner $D$ can

be treated as effectively frozen constants. In particular, for any finite input sets $X_1$ and $X_2$, define the preconditioned Gram operator $\mathsf{K}(X_1, X_2) \triangleq Z(X_1)^\top D\, Z(X_2)$. Because TD learning is bootstrapped through the greedy next-action targets, the same parameter update also propagates through $X'$, leading to the linearized TD-error dynamics

$$\mathbf{e}_{t+1} = \mathbf{e}_t + \eta\, \mathsf{S}\, \bar{\mathbf{e}}_t + o(\eta), \qquad (8)$$

where $\bar{\mathbf{e}}_t$ is Adam's exponential moving average of past TD errors and the TD dynamics operator is

$$\mathsf{S} \triangleq \gamma\, \mathsf{K}(X', X) - \mathsf{K}(X, X). \qquad (9)$$

By coupling this TD-error propagation with the EMA recursion for $\bar{\mathbf{e}}_t$, we obtain a closed-form linear recurrence for TD error $\mathbf{e}_t$. The following theorem provides the exact stability criterion for this linearized system.

**Theorem 4.1.** *Under the assumptions of linearization, greedy stability, and terminal freezing (Assumptions B.1–B.3 in Appendix), for all $t \geq t_0$ the TD error satisfies*

$$\mathbf{e}_{t+1} = \Big((1+\beta_1)I + \eta(1-\beta_1)\mathsf{S}\Big)\mathbf{e}_t - \beta_1\mathbf{e}_{t-1} + o(\eta), \quad (10)$$

*where $\mathsf{S}$ is defined in Eq. (9). In the local first order regime, ignore the $o(\eta)$ term and define*

$$\mathsf{A}(\eta) \triangleq \begin{bmatrix} (1+\beta_1)I + \eta(1-\beta_1)\mathsf{S} & -\beta_1 I \\ I & 0 \end{bmatrix}.$$

*Then the frozen linearized dynamics converges exponentially to 0 if and only if $\rho(\mathsf{A}(\eta)) < 1$.*

Theorem 4.1 establishes that stability is determined by the spectral radius of the augmentation matrix $\mathsf{A}(\eta)$. However, evaluating the spectral radius of a step-dependent block matrix is analytically difficult. To provide a more practical condition, we examine the behavior of the system as the stepsize $\eta$ approaches zero. In this limit, the stability of the discrete Adam updates is governed by the spectral properties of the continuous-time operator $\mathsf{S}$.

**Theorem 4.2.** *Work in the frozen regime of Theorem 4.1 and ignore the $o(\eta)$ term. If $\mathsf{S}$ is Hurwitz, then there exists $\eta_0 > 0$ such that $\rho(\mathsf{A}(\eta)) < 1, \forall \eta \in (0, \eta_0)$.*

This result bridges the gap between the optimizer's discrete recurrence and the underlying operator-theoretic properties of the TD updates. Collectively, Theorems 4.1 and 4.2 establish that a sufficient condition for the exponential decay of the frozen linearized TD error is simply that $\mathsf{S}$ is Hurwitz (i.e., its eigenvalues lie strictly in the left half of the complex plane) and the stepsize $\eta$ is sufficiently small.

*Remark* 4.3. For a real matrix $\mathsf{S}$, Hurwitzness is equivalent to $\Re(\lambda) < 0$ for every $\lambda \in \mathrm{spec}(\mathsf{S})$. Furthermore, if $\Re(\bar\lambda) = \max_{\lambda \in \mathrm{spec}(\mathsf{S})} \Re(\lambda) = 0$, then $|r_1(\eta)|^2 = 1 + O(\eta^2)$, and if an eigenvalue $\bar\lambda$ attaining the maximum also satisfies $\Im(\bar\lambda) = 0$ then $\bar\lambda = 0$ and the characteristic polynomial admits the root $r = 1$ for all $\eta$, so $\rho(\mathsf{A}(\eta)) \geq 1$.

Detailed proofs of these theorems, along with supporting lemmas regarding gradient factorization and momentum recursion, are provided in Appendix B.2 and B.3. Notably, in supervised learning (SL), the bootstrapped feedback term $\gamma \mathsf{K}(X', X)$ is absent. Consequently, $\mathsf{S} = -\mathsf{K}(X, X)$. This term is negative semidefinite due to the inherent properties of $\mathsf{K}(X, X)$. Thus, the dynamics lack positive feedback and typically satisfy $\rho(\mathsf{A}(\eta)) \leq 1$, thereby explaining why collapse is much less common in SL settings.

**Empirical verification.** To connect the operator-level criteria above to observable training behavior, we monitor the dominant eigenmode $\bar{\lambda} \in \operatorname{spec}(\mathsf{S})$ in the late stage where the "frozen terminal" approximation is most accurate, and juxtapose its evolution with the critic loss[3] and task return. Figure 1 illustrates a tight correspondence between the predicted Schur stability of the augmented dynamics (Theorem 4.1) and the learning curves: when the spectrum of $\mathsf{S}$ remains in a regime compatible with contractive $\rho(\mathsf{A}(\eta))$, [4] training exhibits bounded critic loss and sustained performance. In contrast, once the leading mode drifts toward nonnegative real part, the error-propagation feedback ceases to be contractive and the loss rapidly amplifies, accompanied by a sharp degradation in return. Moreover, the observed transition concentrates near the marginal boundary described in Remark: when $\Re(\bar{\lambda}) \approx 0$ with $\Im(\bar{\lambda}) \approx 0$, the augmented recurrence develops a unit root and exactly a unit root when $\bar{\lambda} = 0$, so the iterates no longer decay and training tends to stagnate in a degenerate, near-constant regime rather than recovering[5].

Notably, while previous works such as Kumar et al. (2022) and Yue et al. (2023) derive their insights primarily from the dynamics of SGD, they overlook the complex internal state dynamics of adaptive optimizers like Adam. *Adam is not just 'fast SGD'.* Its momentum and variance adaptation mechanisms create a completely different dynamical system (a second-order difference equation system) compared to SGD. Our work is the first to derive stability conditions specifically for this Adam-dominated regime, which explains why our spectral condition differs from theirs.

## 5. Adam with Orthogonality Correction

### 5.1. When is the TD dynamics operator Hurwitz?

We have shown that the frozen linearized TD-error dynamics is exponentially stable for sufficiently small stepsize once the TD dynamics operator $S$ is Hurwitz. The goal

---

[3]Critic loss of TD3+BC is equal to its TD loss.

[4]In particular, when $\mathsf{S}$ is Hurwitz so that a sufficiently small stepsize would guarantee $\rho(\mathsf{A}(\eta)) < 1$ by Theorem 4.2.

[5]More detailed experimental settings and derivations regarding Figure 1 can be found in Appendix B.5, and we also provide a theoretical case study for online RL in Appendix B.6.

here is therefore to give a compact sufficient condition for Hurwitzness that cleanly separates two effects controlled by different knobs: (i) a bootstrapped scale term, primarily controlled by input normalization/clipping and spectral norm constraints, and (ii) a feature-conditioning term, controlled by near-orthogonality together with input normalization. For simplicity, we denote $\Phi = D^{1/2}Z(X)$, $\Phi_* = D^{1/2}Z(X')$ so that $S = \gamma \Phi_*^\top \Phi - \Phi^\top \Phi$. We begin with a compact sufficient condition for Hurwitzness:

**Proposition 5.1.** *If*

$$\gamma \|\Phi\|_2 \|\Phi_*\|_2 + \|\Phi^\top \Phi - I\|_2 < 1, \qquad (11)$$

*then $S$ is Hurwitz.*

Proposition 5.1 can be obtained by the conclusion of Proposition C.3. To further extend this to the under-parameterized regime, we refer to Proposition C.4, which still guarantees convergence under a slightly weaker condition. The formal bridge is proved in Appendix C.5: $R(\omega)$ controls $\varepsilon(\omega) = \|\Psi(\omega)^\top \Psi(\omega) - I\|_2$, then Corollary C.13 controls $\|\Phi^\top \Phi - I\|_2$, and Corollary C.14 closes the Hurwitz condition. This proposition is a conservative sufficient condition rather than a necessary characterization. In practical TD3/SAC-style critics, stop-gradient target networks and Polyak averaging attenuate the bootstrap term through the refined operator $\mathsf{S}_\alpha = \alpha\gamma\mathsf{K}(X', X) - \mathsf{K}(X, X)$, as detailed in Appendix B.5. When $M > P$, the column-Gram condition is replaced by the row-Gram complement in Proposition C.4, which controls all nonzero modes of $S$ and rules out divergent modes. Moreover, it offers an intuitive recipe for stability by balancing two competing forces: $\gamma \|\Phi\|_2 \|\Phi_*\|_2$ quantifies the bootstrapped amplification through greedy targets, while $\|\Phi^\top \Phi - I\|_2$ quantifies how far the Jacobian features deviate from a near-orthonormal geometry. The remaining question is *how to control two terms in practice*.

**Controlling the bootstrapped scale term $\gamma \|\Phi\|_2 \|\Phi_*\|_2$.** To make Proposition 5.1 operational, Appendix Lemma C.5 shows that, under standard input normalization/clipping and spectral norm constraints/normalization, the operator norms of the Adam-whitened Jacobian feature matrices are uniformly bounded $\|\Phi\|_2, \|\Phi_*\|_2 \leq \sqrt{M}G$, where $M \triangleq |X|$ and $G$ is the explicit constant defined in Lemma C.5. Therefore, we have $\gamma \|\Phi\|_2 \|\Phi_*\|_2 \leq \gamma MG^2$. Thus, to control $\gamma \|\Phi\|_2 \|\Phi_*\|_2$, it suffices to keep the per-sample Jacobian magnitude bounded via *input normalization* and *spectral norm constraints* on the critic network.

**Controlling the feature-conditioning term $\|\Phi^\top \Phi - I\|_2$.** The second term in Proposition 5.1 requires that the Gram matrix of whitened Jacobian features be close to identity. Appendix Proposition C.7 explains how to control this Gram deviation by separating a parameter contribution from a data contribution. Specifically, assume that at the frozen iterate

the whitened Jacobian feature matrix admits a dictionary-type factorization $\Phi = \Psi U$, which is analyzed in the Appendix C. Then Proposition C.7 yields:

$$\|\Phi^\top \Phi - I\|_2 \leq \left(1 + \delta + (M-1)\rho\right)\varepsilon + \delta + (M-1)\rho, \quad (12)$$

where $\varepsilon \triangleq \|\Psi^\top \Psi - I\|_2$ is a parameter-only near-isometry / orthogonality quantity, while $\delta$ and $\rho$ quantify code normalization and overlap (how close $\|u_i\|_2^2$ is to 1 and how large $|u_i^\top u_j|$ can be for $i \neq j$). This decomposition is the bridge we need: enforcing parameter orthogonality in the optimizer targets $\varepsilon$ directly, while input normalization/clipping also helps keep the induced per-sample codes on a comparable scale, which supports small $\delta$ and limits the data-induced contribution to the Gram deviation[6]. In summary, it suffices to control the parameter near-isometry term $\varepsilon = \|\Psi^\top \Psi - I\|_2$ and the intput normalization, in order to constrain $\|\Phi^\top \Phi - I\|_2$.

**End-to-end sufficient condition and takeaway.** In this subsection, Proposition 5.1 reduces stability of the frozen linearized TD dynamics to Hurwitzness of the TD operator $S$. Appendix Lemma C.5 shows that input normalization/clipping and spectral norm constraints bound the bootstrapped scale term $\gamma\|\Phi\|_2\|\Phi_*\|_2$, while Appendix Proposition C.7 reduces the conditioning term $\|\Phi^\top \Phi - I\|_2$ to the parameter orthogonality distortion $\varepsilon = \|\Psi^\top \Psi - I\|_2$. By Theorem 4.2, these controls suffice to ensure exponential decay of the frozen linearized TD error for sufficiently small stepsize. The first two are ubiquitous in deep RL practice (Fujimoto & Gu, 2021; Yu et al., 2020; Miyato et al., 2018), whereas the next subsection focuses on how to control $\|\Psi^\top \Psi - I\|_2$ *directly in the optimizer* without contaminating Adam's moment statistics.

### 5.2. Enforcing parameter orthogonality in the optimizer

Section 5.1 shows that normalization/clipping and spectral constraints mainly control the scale term in the Hurwitz sufficient condition, whereas the term $\varepsilon = \|\Psi^\top \Psi - I\|_2$ requires separate control. This quantity quantifies how far the parameter-induced subspace geometry deviates from an isometry and therefore captures the parameter-side contribution to the conditioning requirement. Since $\Psi$ is not explicitly represented in a general deep network, we cannot constrain $\varepsilon$ directly. Instead, we act on the matrix-shaped weight blocks that implement the network's linear maps and enforce blockwise near-orthogonality on these blocks.

In a general deep network, however, $\Psi$ is only an implicit object induced by the frozen linearization, so $\varepsilon$ cannot be constrained directly. Our strategy is therefore to control a tractable surrogate at the level of the actual model param-

eters: the matrix-shaped weight blocks that implement the network's linear maps. This is the sense in which we use the phrase *parameter orthogonality* below, namely, blockwise near-orthogonality of selected weight matrices as a proxy for improving the geometry of the induced dictionary $\Psi$. Concretely, let $\omega$ denote the full parameter vector. We select a collection of constrained blocks, indexed by $b \in \mathcal{B}$, and reshape them into matrices $\{W_b\}_{b\in\mathcal{B}}$. Following standard soft orthogonality regularization (Brock et al., 2017; Bansal et al., 2018), we penalize deviations from semi-orthogonality by $\bar{R}(\omega) \triangleq \sum_{b\in\mathcal{B}} R(W_b)$, where[7] for a generic block matrix $W \in \mathbb{R}^{r\times c}$,

$$R(W) = \begin{cases} \frac{1}{4}\|WW^\top - I_r\|_F^2, & r < c, \\ \frac{1}{4}\|W^\top W - I_c\|_F^2, & r \geq c, \end{cases} \quad (13)$$

and denote its gradient by $r_t \triangleq \nabla_{\omega_t} \bar{R}(\omega_t)$. Appendix C.5 makes the surrogate relation precise in the same frozen regime used in the stability analysis. Under the local continuation and baseline-distortion assumptions stated there, the induced dictionary distortion satisfies

$$\varepsilon(\omega) \triangleq \|\Psi(\omega)^\top \Psi(\omega) - I\|_2 \leq \varepsilon_0 + c_1\sqrt{\bar{R}(\omega)} + c_2\bar{R}(\omega),$$

for some local constants $c_1, c_2 > 0$. Consequently, reducing the blockwise orthogonality defect of the actual weight matrices reduces the parameter-side geometry term $\varepsilon$ that enters the Hurwitz sufficient condition in Sec. 5.1. Put differently, normalization/clipping and spectral constraints control the scale term, while blockwise weight orthogonality provides a practical handle on the geometry term.

This leaves one implementation question: *how should this orthogonality control be imposed when the base optimizer is Adam?* The next paragraph explains why directly adding $\bar{R}(\omega)$ to the task loss is not the right mechanism under Adam, and why the orthogonality correction should instead be enforced as a decoupled optimizer-side drift. In other words, AdamO is a decoupled optimizer-side correction rather than a standard regularizer added to the loss.

**Why enforcing orthogonality in the Adam optimizer?** A standard approach is to add an auxiliary penalty,

$$\widetilde{L}_t(\omega) = L_t(\omega) + \lambda\bar{R}(\omega), \quad (14)$$

and run Adam on $\nabla\widetilde{L}_t = g_t + \lambda r_t$, where $g_t = \nabla L_t(\omega_t)$ is the task gradient. However, under Adam the second-moment recursion uses elementwise squares,

$$v_{t+1} = \beta_2 v_t + (1 - \beta_2)(g_t + \lambda r_t)^{\odot 2}, \quad (15)$$

so $\lambda r_t$ is injected into Adam's moment path. This moment contamination is the main design reason for optimizer-level decoupling, rather than a secondary implementation detail.

---

[6]The constraint of $\delta$ and $\rho$ is supported by normalization in the frozen regime, which is discussed in the Appendix C.

[7]When multiple layers are constrained, $\bar{R}(\omega)$ is the sum of these terms over the selected blocks.

*Sufficiency.* Orthogonality can be enforced in the optimizer because $R$ depends only on the current parameters. In particular, for a matrix block $W \in \mathbb{R}^{r \times c}$, the gradient of the regularization Eq. (13) admits a closed form:

$$\nabla_W R(W) = \begin{cases} (WW^\top - I_r)W, & r < c, \\ W(W^\top W - I_c), & r \geq c. \end{cases} \quad (16)$$

Therefore, $r_t$ can be evaluated deterministically at $\omega_t$ and applied as an additional drift without changing the stochastic task-gradient stream $g_t$ that Adam is meant to adapt to.

*Necessity under Adam.* If $\lambda r_t$ is included in the gradient stream, then Eq. (15) records it into $(m_t, v_t)$, creating a history- and coordinate-dependent effective regularization strength and potentially distorting future task updates even after $r_t$ becomes small[8] (Loshchilov & Hutter, 2019). Therefore, we decouple orthogonality from Adam: Adam only sees $g_t$, while orthogonality is enforced by an extra drift update acting on the current parameters. Concretely, AdamO keeps Adam's moments driven only by task gradients and applies a budgeted orthogonality correction after the Adam direction has been formed. A formal statement of this rationale is given in Proposition D.1.

---

**Algorithm 1** AdamO: Adam with Orthogonality Correction

---

1: **Init** $\omega_0, m_0 = v_0 = 0, \eta, \kappa, \tau$ and Adam params
2: **for** $t = 0, 1, \dots$ **do**
3:    $g_t \leftarrow \nabla L_t(\omega_t)$
4:    Compute standard Adam update $u_t$
5:    Compute $\delta_t$ by applying (17)–(19) layer-wise
6:    $\omega_{t+1} \leftarrow \omega_t - \eta(u_t + \delta_t)$
7: **end for**

---

**AdamO: Adam optimizer with orthogonality correction.**
In TD learning, $g_t = \nabla L_t(\omega_t)$ is the task gradient and $u_t$ denotes the standard Adam update direction computed only from $g_t$. We define an orthogonality correction for each constrained parameter matrix. For clarity, we describe the map for a single constrained matrix $W$. In practice, it is applied independently to each constrained matrix, with $g_t, u_t, r_t$ understood as the corresponding layer-wise restrictions. First, we compute a scale-normalized reference step $\delta_{t,0}$ that matches the local Adam scale:

$$\delta_{t,0} = \kappa \frac{\|u_t\|_F}{\|r_t\|_F + \varepsilon_r} r_t. \quad (17)$$

This is applied layer-wise to ensure scale invariance. To prevent the correction from hijacking task progress, we enforce a local budget constraint

$$\langle g_t, \delta_t \rangle_F \geq -\tau\big(\langle g_t, u_t \rangle_F\big)_+, \qquad \tau \in [0, 1). \quad (18)$$

---

[8]cf. the decoupled weight decay in AdamW.

Among all scalings of $\delta_{t,0}$ along its ray, AdamO takes the largest feasible one, yielding the closed form:

$$\delta_t = \begin{cases} \delta_{t,0}, & \text{if } \langle g_t, \delta_{t,0} \rangle_F \geq T_t, \\ \frac{T_t}{\langle g_t, \delta_{t,0} \rangle_F} \delta_{t,0}, & \text{otherwise,} \end{cases} \quad (19)$$

where $T_t \triangleq -\tau\big(\langle g_t, u_t \rangle_F\big)_+$. Finally, AdamO performs

$$\omega_{t+1} = \omega_t - \eta(u_t + \delta_t), \quad (20)$$

where $\delta_t$ collects the per-layer corrections. The pseudocode is given in Algorithm 1.

**Theoretical analysis of orthogonality correction module.**
To enforce orthogonality while preserving Adam update behavior, AdamO applies orthogonality as a decoupled correction within the optimizer instead of mixing an orthogonality penalty into the task gradient. If one runs Adam on a penalized gradient that includes the orthogonality term, the orthogonality signal enters the moment estimates and can distort future task steps even after the orth gradient vanishes. This is exactly the moment of the contamination issue in Proposition D.1, and it motivates keeping the Adam moments driven only by the task gradient.

AdamO then makes the orthogonality correction explicitly safe for task optimization. The closed form scaling satisfies the per-block task progress budget exactly in Lemma D.2. The correction is also ratio clipped by construction, so its magnitude is bounded by the Adam update magnitude, as shown in Lemma D.3 through $\|\Delta_t\| \leq \kappa\|u_t\|$. In the conflict-free regime with $\tau = 0$, the budget enforces $\langle g_t, \Delta_t \rangle \geq 0$, so the orthogonality correction never cancels first-order task descent, and under the smoothness stepsize condition in Eq. (90), the next step task loss under AdamO is noninferior to Adam, as stated in Theorem D.5. When $\tau > 0$, AdamO allows a controlled amount of misalignment to strengthen orthogonality, and Theorem D.5 provides an explicit upper bound on the worst-case single-step degradation that depends on $\tau$ and on curvature through $\mu$, the stepsize $\eta$, and the correction scale $\kappa$.

Finally, the continuous time Hamiltonian view explains why the orthogonality module does not break the Adam stability structure. Proposition D.6 shows that Adam admits a dissipative Hamiltonian whenever $\beta_1 \geq \beta_2/4$. AdamO changes only the parameter dynamics by adding the orthogonality drift, so the Hamiltonian derivative gains exactly one additional inner product term, and Theorem D.8 shows that monotone decrease is preserved for $\tau = 0$ and becomes a controlled differential inequality for $\tau > 0$. Overall, the orthogonality module does not introduce uncontrolled growth in the task dynamics, and its effect is explicitly bounded and regulated by $\kappa$ and $\tau$. Specifically, we establish the theoretical admissible ranges for $\kappa$ in Appendix D.5.

*Table 1.* Comparison between standard **Adam** and our **AdamO** on D4RL benchmarks. **Bold** indicates the higher score. Subscripts denote the relative percentage improvement (↑) or degradation (↓) of AdamO over Adam. Quantitative analysis across the 10k samples Mujoco Locomotion suite and AntMaze domain where 'm' and 'med' denote medium quality and 'mr' represents medium replay datasets while 'me' stands for medium expert and 'div' indicates diverse data distributions.

| Task Name | TD3+BC | | IQL | | ReBRAC | | ACTIVE | | PARS | | SQOG | |
|---|---|---|---|---|---|---|---|---|---|---|---|---|
| | Adam | AdamO | Adam | AdamO | Adam | AdamO | Adam | AdamO | Adam | AdamO | Adam | AdamO |
| AntMaze-umaze | 72.2 | $\mathbf{92.5}_{\uparrow 28\%}$ | 80.5 | $\mathbf{83.8}_{\uparrow 4\%}$ | 94.3 | $\mathbf{96.5}_{\uparrow 2\%}$ | 92.4 | $\mathbf{95.1}_{\uparrow 3\%}$ | **97.3** | $93.5_{\downarrow 4\%}$ | 89.6 | $\mathbf{93.1}_{\uparrow 4\%}$ |
| AntMaze-umaze-div | 47.0 | $\mathbf{82.2}_{\uparrow 75\%}$ | 55.8 | $\mathbf{68.2}_{\uparrow 22\%}$ | 87.2 | $\mathbf{91.5}_{\uparrow 5\%}$ | 75.9 | $\mathbf{78.2}_{\uparrow 3\%}$ | **93.2** | $92.4_{\downarrow 1\%}$ | 72.8 | $\mathbf{87.1}_{\uparrow 20\%}$ |
| AntMaze-med-play | 0.3 | $\mathbf{28.5}_{\uparrow \infty}$ | **70.4** | $70.1_{\downarrow 0.4\%}$ | 85.2 | $\mathbf{89.4}_{\uparrow 5\%}$ | 73.7 | $\mathbf{86.5}_{\uparrow 17\%}$ | 91.5 | $\mathbf{92.8}_{\uparrow 1\%}$ | 60.9 | $\mathbf{65.4}_{\uparrow 7\%}$ |
| AntMaze-med-div | 0.2 | $\mathbf{16.4}_{\uparrow \infty}$ | 66.9 | $\mathbf{75.5}_{\uparrow 13\%}$ | 79.8 | $\mathbf{85.1}_{\uparrow 7\%}$ | 77.8 | $\mathbf{82.4}_{\uparrow 6\%}$ | 87.1 | $\mathbf{90.6}_{\uparrow 4\%}$ | 65.8 | $\mathbf{81.2}_{\uparrow 23\%}$ |
| AntMaze-large-play | 0.0 | $\mathbf{13.7}_{\uparrow \infty}$ | 38.5 | $\mathbf{41.8}_{\uparrow 9\%}$ | 55.4 | $\mathbf{61.8}_{\uparrow 12\%}$ | 50.9 | $\mathbf{56.5}_{\uparrow 11\%}$ | 46.6 | $\mathbf{52.9}_{\uparrow 14\%}$ | 52.6 | $\mathbf{57.4}_{\uparrow 9\%}$ |
| AntMaze-large-div | 0.0 | $\mathbf{16.5}_{\uparrow \infty}$ | 32.4 | $\mathbf{36.1}_{\uparrow 11\%}$ | 50.5 | $\mathbf{56.2}_{\uparrow 11\%}$ | 46.6 | $\mathbf{61.8}_{\uparrow 33\%}$ | 50.8 | $\mathbf{56.5}_{\uparrow 11\%}$ | 47.5 | $\mathbf{52.8}_{\uparrow 11\%}$ |
| AntMaze-ultra-div | 0.0 | $\mathbf{2.4}_{\uparrow \infty}$ | 19.0 | $\mathbf{31.1}_{\uparrow 64\%}$ | 5.7 | $\mathbf{9.4}_{\uparrow 65\%}$ | 10.0 | $\mathbf{19.8}_{\uparrow 98\%}$ | 42.1 | $\mathbf{48.6}_{\uparrow 15\%}$ | 0.0 | $\mathbf{4.2}_{\uparrow \infty}$ |
| AntMaze-ultra-play | 0.0 | $\mathbf{1.8}_{\uparrow \infty}$ | 21.0 | $\mathbf{29.9}_{\uparrow 42\%}$ | 20.6 | $\mathbf{26.8}_{\uparrow 30\%}$ | **13.2** | $11.5_{\downarrow 13\%}$ | 55.9 | $\mathbf{62.4}_{\uparrow 12\%}$ | 0.0 | $\mathbf{2.1}_{\uparrow \infty}$ |
| *AntMaze Avg.* | 15.0 | $\mathbf{31.8}_{\uparrow 112\%}$ | 48.1 | $\mathbf{54.6}_{\uparrow 14\%}$ | 59.8 | $\mathbf{64.6}_{\uparrow 8\%}$ | 55.1 | $\mathbf{61.5}_{\uparrow 12\%}$ | 70.6 | $\mathbf{73.7}_{\uparrow 4\%}$ | 48.7 | $\mathbf{55.4}_{\uparrow 14\%}$ |
| HalfCheetah-m | 35.9 | $\mathbf{53.5}_{\uparrow 49\%}$ | 29.7 | $\mathbf{35.2}_{\uparrow 19\%}$ | 42.8 | $\mathbf{49.4}_{\uparrow 15\%}$ | 42.9 | $\mathbf{48.5}_{\uparrow 13\%}$ | 44.9 | $\mathbf{51.2}_{\uparrow 14\%}$ | 36.5 | $\mathbf{41.8}_{\uparrow 15\%}$ |
| HalfCheetah-mr | 39.1 | $\mathbf{46.2}_{\uparrow 18\%}$ | 32.7 | $\mathbf{38.4}_{\uparrow 17\%}$ | 40.7 | $\mathbf{47.1}_{\uparrow 16\%}$ | 34.9 | $\mathbf{40.6}_{\uparrow 16\%}$ | 45.4 | $\mathbf{50.8}_{\uparrow 12\%}$ | 30.2 | $\mathbf{35.5}_{\uparrow 18\%}$ |
| HalfCheetah-me | 33.5 | $\mathbf{60.1}_{\uparrow 79\%}$ | 48.1 | $\mathbf{54.6}_{\uparrow 14\%}$ | 63.9 | $\mathbf{71.5}_{\uparrow 12\%}$ | 55.7 | $\mathbf{62.3}_{\uparrow 12\%}$ | 75.7 | $\mathbf{82.9}_{\uparrow 10\%}$ | 58.9 | $\mathbf{65.4}_{\uparrow 11\%}$ |
| Hopper-m | 40.7 | $\mathbf{75.5}_{\uparrow 86\%}$ | 38.9 | $\mathbf{45.1}_{\uparrow 16\%}$ | 78.6 | $\mathbf{86.2}_{\uparrow 10\%}$ | 26.2 | $\mathbf{31.8}_{\uparrow 21\%}$ | 73.7 | $\mathbf{80.4}_{\uparrow 9\%}$ | 45.8 | $\mathbf{51.3}_{\uparrow 12\%}$ |
| Hopper-mr | 21.3 | $\mathbf{55.8}_{\uparrow 162\%}$ | 46.6 | $\mathbf{52.9}_{\uparrow 14\%}$ | 64.2 | $\mathbf{70.8}_{\uparrow 10\%}$ | 62.5 | $\mathbf{68.7}_{\uparrow 10\%}$ | 67.5 | $\mathbf{74.1}_{\uparrow 10\%}$ | 61.4 | $\mathbf{67.2}_{\uparrow 9\%}$ |
| Hopper-me | 32.6 | $\mathbf{84.2}_{\uparrow 158\%}$ | 66.5 | $\mathbf{73.2}_{\uparrow 10\%}$ | 78.5 | $\mathbf{85.9}_{\uparrow 9\%}$ | 62.7 | $\mathbf{69.1}_{\uparrow 10\%}$ | 75.5 | $\mathbf{81.6}_{\uparrow 8\%}$ | 72.9 | $\mathbf{78.5}_{\uparrow 8\%}$ |
| Walker2d-m | 21.2 | $\mathbf{68.4}_{\uparrow 223\%}$ | 54.9 | $\mathbf{61.5}_{\uparrow 12\%}$ | 62.2 | $\mathbf{69.4}_{\uparrow 12\%}$ | 53.6 | $\mathbf{59.2}_{\uparrow 10\%}$ | 68.0 | $\mathbf{74.5}_{\uparrow 10\%}$ | 49.8 | $\mathbf{55.6}_{\uparrow 12\%}$ |
| Walker2d-mr | 19.3 | $\mathbf{52.5}_{\uparrow 172\%}$ | 51.4 | $\mathbf{57.8}_{\uparrow 12\%}$ | 69.4 | $\mathbf{76.1}_{\uparrow 10\%}$ | 45.8 | $\mathbf{51.4}_{\uparrow 12\%}$ | 63.4 | $\mathbf{69.2}_{\uparrow 9\%}$ | 58.5 | $\mathbf{64.3}_{\uparrow 10\%}$ |
| Walker2d-me | 22.4 | $\mathbf{98.5}_{\uparrow 340\%}$ | 57.3 | $\mathbf{63.7}_{\uparrow 11\%}$ | 74.0 | $\mathbf{80.8}_{\uparrow 9\%}$ | 58.6 | $\mathbf{64.9}_{\uparrow 11\%}$ | 81.6 | $\mathbf{88.3}_{\uparrow 8\%}$ | 68.7 | $\mathbf{74.9}_{\uparrow 9\%}$ |
| *Locomotion Avg.* | 29.6 | $\mathbf{66.1}_{\uparrow 123\%}$ | 47.3 | $\mathbf{53.6}_{\uparrow 13\%}$ | 63.8 | $\mathbf{70.8}_{\uparrow 11\%}$ | 49.2 | $\mathbf{55.2}_{\uparrow 12\%}$ | 66.2 | $\mathbf{72.6}_{\uparrow 10\%}$ | 53.6 | $\mathbf{59.4}_{\uparrow 11\%}$ |
| Pen-human | −4.1 | $\mathbf{83.1}_{\uparrow \infty}$ | 75.1 | $\mathbf{99.8}_{\uparrow 33\%}$ | **105.4** | $102.5_{\downarrow 3\%}$ | 106.2 | $\mathbf{109.8}_{\uparrow 3\%}$ | 88.1 | $\mathbf{94.5}_{\uparrow 7\%}$ | **77.0** | $75.9_{\downarrow 1\%}$ |
| Pen-cloned | 5.6 | $\mathbf{82.4}_{\uparrow 1371\%}$ | 46.5 | $\mathbf{82.2}_{\uparrow 77\%}$ | **98.5** | $92.4_{\downarrow 6\%}$ | 96.5 | $\mathbf{100.2}_{\uparrow 4\%}$ | **107.1** | $105.8_{\downarrow 1\%}$ | 73.6 | $\mathbf{87.4}_{\uparrow 19\%}$ |
| Door-human | −0.3 | $\mathbf{0.2}_{\uparrow \infty}$ | 3.5 | $\mathbf{4.8}_{\uparrow 37\%}$ | −0.1 | $\mathbf{0.1}_{\uparrow \infty}$ | 0.0 | $\mathbf{0.2}_{\uparrow \infty}$ | 0.1 | $\mathbf{0.3}_{\uparrow 200\%}$ | −0.1 | $\mathbf{0.1}_{\uparrow \infty}$ |
| Door-cloned | −0.3 | $\mathbf{0.1}_{\uparrow \infty}$ | 3.3 | $\mathbf{4.5}_{\uparrow 36\%}$ | 0.1 | $\mathbf{0.5}_{\uparrow 400\%}$ | 0.1 | $\mathbf{0.4}_{\uparrow 300\%}$ | 0.1 | $\mathbf{0.3}_{\uparrow 200\%}$ | 0.1 | $\mathbf{0.5}_{\uparrow 400\%}$ |
| Hammer-human | **1.1** | $0.5_{\downarrow 55\%}$ | 1.9 | $\mathbf{3.1}_{\uparrow 63\%}$ | 0.3 | $\mathbf{0.8}_{\uparrow 167\%}$ | 0.3 | $\mathbf{0.9}_{\uparrow 200\%}$ | 0.3 | $\mathbf{0.8}_{\uparrow 167\%}$ | 0.3 | $\mathbf{0.8}_{\uparrow 167\%}$ |
| Hammer-cloned | 0.3 | $\mathbf{0.4}_{\uparrow 33\%}$ | 1.7 | $\mathbf{3.2}_{\uparrow 88\%}$ | 5.4 | $\mathbf{7.5}_{\uparrow 39\%}$ | 5.9 | $\mathbf{7.8}_{\uparrow 32\%}$ | 4.6 | $\mathbf{7.0}_{\uparrow 52\%}$ | 5.1 | $\mathbf{6.8}_{\uparrow 33\%}$ |
| Relocate-human | −0.3 | $\mathbf{0.2}_{\uparrow \infty}$ | 0.1 | $\mathbf{0.5}_{\uparrow 400\%}$ | 0.2 | $\mathbf{0.6}_{\uparrow 200\%}$ | 0.2 | $\mathbf{0.7}_{\uparrow 250\%}$ | 0.1 | $\mathbf{0.7}_{\uparrow 600\%}$ | 0.2 | $\mathbf{0.6}_{\uparrow 200\%}$ |
| Relocate-cloned | **0.1** | $0.0_{\downarrow 100\%}$ | 0.0 | 0.0 | 0.0 | $\mathbf{0.4}_{\uparrow \infty}$ | **0.2** | $0.1_{\downarrow 50\%}$ | **0.1** | $0.0_{\downarrow 100\%}$ | −0.2 | $\mathbf{0.6}_{\uparrow \infty}$ |
| *Adroit Avg.* | 0.3 | $\mathbf{20.9}_{\uparrow 6867\%}$ | 16.5 | $\mathbf{24.8}_{\uparrow 50\%}$ | **26.2** | $25.6_{\downarrow 2\%}$ | 26.2 | $\mathbf{27.5}_{\uparrow 5\%}$ | 25.1 | $\mathbf{26.2}_{\uparrow 4\%}$ | 19.5 | $\mathbf{21.6}_{\uparrow 11\%}$ |

# 6. Experiment

## 6.1. Experimental Setup

**Baselines.** To demonstrate the versatility of our approach, we incorporate the proposed optimizer into a comprehensive suite of offline RL methods. Our evaluation spans diverse paradigms, including policy-constraint and regularization methods (TD3+BC, Fujimoto & Gu, 2021; Re-BRAC, Tarasov et al., 2023), and in-sample learning (IQL, Kostrikov et al., 2022; ACTIVE, Chen et al., 2025). Furthermore, we assess performance on algorithms designed for robustness against distribution shift (PARS, Kim et al., 2025; SQOG, Yao et al., 2025). Additionally, we compare our approach against general-purpose first-order optimizers, including SGD (Robbins & Monro, 1951), Adam (Kingma & Ba, 2017), and AdamW (Loshchilov & Hutter, 2019). We also compare against stability-focused techniques: periodic state *Resetting* (Asadi et al., 2023), the adaptive TRAC (Muppidi et al., 2024), Kron (Castanyer et al., 2025), and

cautious momentum methods such as C-AdamW (Liang et al., 2025b). In all actor-critic experiments, these optimizers are applied exclusively to the critic networks to strictly isolate their influence on value stability.

**Domains and Datasets.** Our evaluation suite covers numerous tasks across distinct domains, encompassing locomotion, manipulation, and high-dimensional control. We utilize standard D4RL datasets (Fu et al., 2020) for *AntMaze*, *Adroit*, *MuJoCo*, and *FrankaKitchen*. This selection ensures coverage of diverse data qualities and horizon lengths. We additionally report model-based offline RL baselines (MOPO (Yu et al., 2020) and MOBILE (Sun et al., 2023)), newer benchmark results on OGBench (Park et al., 2025) and NeoRL2 (Gao et al., 2025), and related control experiments in Appendix A.4.6. To facilitate cross-domain aggregation, we report the standard D4RL normalized score: $\text{Score} = 100 \times \frac{R - R_{\text{random}}}{R_{\text{expert}} - R_{\text{random}}}$, where denotes the learned policy's return (Fu et al., 2020).

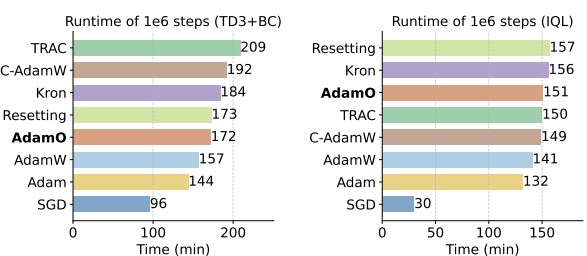

*Figure 2.* Comparison of computational overhead and wall-clock runtime across different optimizers.

## 7. Main Results

**Computational Efficiency** In Fig. 2, AdamO incurs only a minor computational overhead compared to Adam (e.g., 172 vs. 144 minutes on TD3+BC) while remaining faster than or comparable to advanced optimizers like TRAC and Kron. This confirms that AdamO strikes a favorable balance, delivering SOTA performance with manageable costs.

**Performance Analysis** Table 1 details the quantitative comparison between Adam and AdamO across six offline RL algorithms on D4RL benchmarks. The results indicate that AdamO consistently outperforms the standard Adam optimizer across the vast majority of tasks. We observe robust improvements in the MuJoCo Locomotion suite and Adroit domain, where AdamO achieves higher scores on diverse dataset qualities and tasks relative to the baselines. The advantages of AdamO are most evident in the challenging AntMaze domain where sparse rewards often hinder learning. While baseline algorithms using Adam frequently fail to learn meaningful policies on large and ultra maps, AdamO successfully recovers performance and achieves significant score increases. For instance, TD3+BC with AdamO improves the average AntMaze score by over 100 percent compared to the baseline. This demonstrates the capability of our optimizer to handle complex optimization landscapes better than standard methods.

**Appendix Reference** Due to space constraints, we provide more extended benchmark descriptions, implementation and baselines details in Appendix A.1–A.3. This supplementary section also contains experimental details and complete results comparing different optimizers and algorithms. Furthermore, we refer readers to Appendix A.4 for additional results on generalization, hyperparameter robustness, and computational efficiency, along with a direct verification of the theoretical stability conditions. Appendix A.4.6 further includes model-based offline RL baselines, OGBench/NeoRL2 benchmark results, SGD-O compatibility, Monte Carlo critic-calibration evidence, and a TD3 ablation showing that conservative policy constraints remain complementary to AdamO.

## 8. Conclusion

This work establishes an optimizer-centric framework for offline reinforcement learning, identifying that value collapse is not merely an architectural flaw but a spectral phenomenon of the self-excitation operator. By introducing AdamO, we demonstrate that this instability can be actively suppressed through a principled, decoupled orthogonality correction. However, a fundamental tension remains. While infinitesimal step sizes ($\eta \to 0$) theoretically permit robust stability control via a large orthogonality budget, they incur prohibitive computational costs in practice. Consequently, our approach remains bound by the inescapable trade-off between spectral stability and training efficiency.

We invite the community to look beyond architectural heuristics and address this optimization bottleneck. Uncovering the mechanism is merely the "morning light" of the Way, the pursuit of a complete solution continues.

## Impact Statement

This paper aims to advance the reliability of offline reinforcement learning by providing an optimizer-centric characterization of temporal-difference error collapse and proposing *AdamO* to improve critic stability via a decoupled orthogonality correction. Improved stability can have positive societal impact by making it more practical to learn from previously collected datasets, potentially reducing the need for risky online exploration and costly trial-and-error in safety-sensitive settings such as robotics, industrial control, or data-driven decision systems. At the same time, making offline RL training more stable may lower the barrier to deploying RL-based policies in high-stakes environments where distribution shift, reward misspecification, or biased/low-quality logged data can lead to harmful outcomes.

## Acknowledgements

This research was supported in part by the National Natural Science Foundation of China under Grants 62572496 and 62432004, the Fundamental and Interdisciplinary Disciplines Breakthrough Plan of the Ministry of Education of China under Grant No. JYB2025XDXM122, the Guangdong Natural Science Foundation under Grant 2026A1515011265, the Shenzhen Science and Technology Program under Grant JCYJ20250604175500001, the Young Elite Scientist Sponsorship Program by CAST under Contract ZB2025-218, a grant from the Guoqiang Institute, Tsinghua University, the Shenzhen Science and Technology Program under Grant No. SYSRD20250529113401002, and in part by the Smartchip Research Project of the Chinese Institute of Electronics under Grant No. 2024-02.

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

# A. Experimental Setup

This section specifies the implementation and evaluation settings needed to replicate our results.

## A.1. Benchmarks

Our empirical study covers four standard offline RL testbeds, namely MuJoCo, Adroit, FrankaKitchen, and AntMaze, following common practice in the literature. We describe each benchmark below.

**MuJoCo.** We use the standard MuJoCo locomotion benchmarks from D4RL, including `Hopper`, `HalfCheetah`, and `Walker2d`. These continuous control tasks provide dense rewards that encourage forward progress while maintaining stability. We report results on the common offline dataset settings `medium`, `medium-replay`, and `medium-expert`, which cover different behavior quality and policy mixture levels.

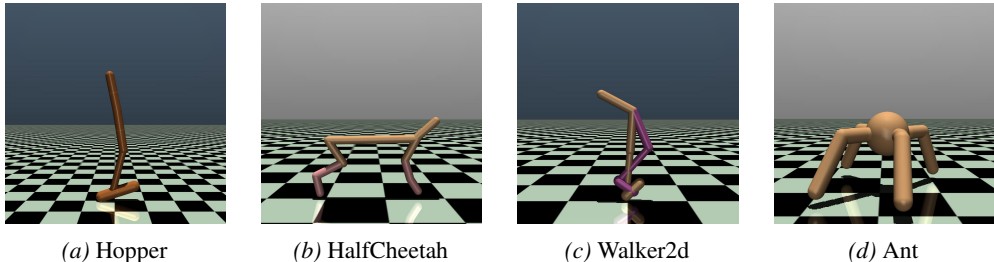

*(a)* Hopper      *(b)* HalfCheetah      *(c)* Walker2d      *(d)* Ant

*Figure 3.* MuJoCo locomotion environments used in our offline RL benchmark.

**Adroit.** We evaluate dexterous manipulation with the Adroit suite, which requires fine grained contact rich control. Following D4RL, we include four tasks, `pen`, `hammer`, `door`, and `relocate`. We use the standard dataset variants `human`, `cloned`, and `expert`, which represent limited demonstrations, rollouts from an imitation policy mixed with demonstrations, and trajectories produced by a stronger policy.

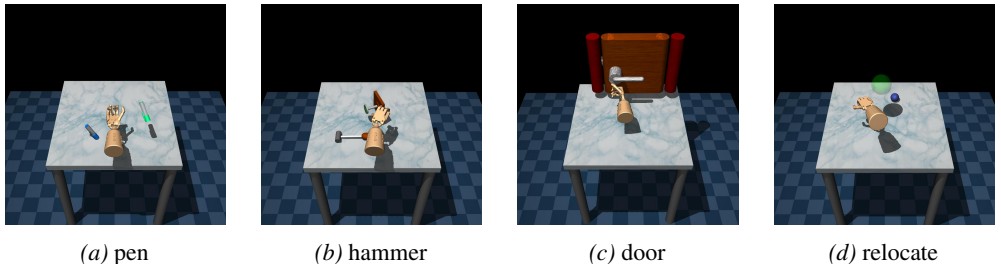

*(a)* pen      *(b)* hammer      *(c)* door      *(d)* relocate

*Figure 4.* Adroit dexterous manipulation tasks in D4RL.

**AntMaze.** We use AntMaze to test sparse reward goal reaching in mazes with the MuJoCo ant. The reward is given at the goal and is zero elsewhere, so offline learning must rely on credit assignment and good use of the dataset support. We include multiple maze scales and variants, and follow the D4RL collection protocol that uses waypoint-guided navigation to generate the offline data.

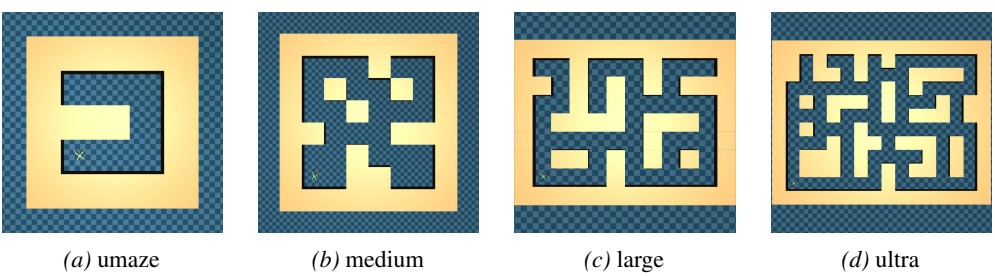

*(a)* umaze      *(b)* medium      *(c)* large      *(d)* ultra

*Figure 5.* AntMaze layouts used for offline evaluation.

**FrankaKitchen.** We consider FrankaKitchen, a long-horizon manipulation domain where a Franka arm must coordinate multiple interactions in a kitchen scene to reach a target configuration. We follow the D4RL datasets `complete`, `partial`, and `mixed`, which differ in how directly the logged trajectories align with the evaluation objective and how much composition from subtrajectories is required.

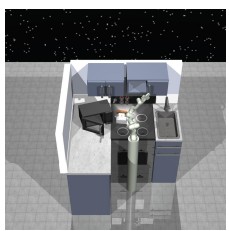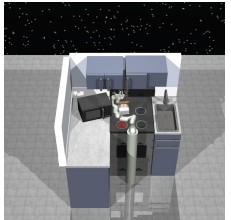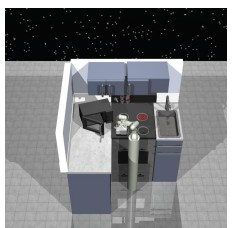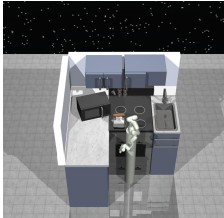

*Figure 6.* FrankaKitchen datasets and an example scene used in our benchmark.

### A.2. Implementation Details

We implement our framework using PyTorch 2.5.1. Our codebase is built upon the open-source libraries CORL: https://github.com/tinkoff-ai/CORL (Apache-2.0 License) and OfflineRL-Kit: https://github.com/yihaosun1124/OfflineRL-Kit (MIT License). All experiments were conducted on a server running Ubuntu 20.04.2 LTS, equipped with four NVIDIA GeForce RTX 3090 GPUs.

**Network Architecture.** Our method uses consistent network structures across benchmarks. The policy is parameterized as a 2-layer feedforward neural network with 256 hidden units, utilizing ReLU activation functions and Tanh Gaussian outputs. The discriminator follows a similar architecture: a 2-layer feedforward network with 256 hidden units and ReLU activations. For the more complex Antmaze tasks, the discriminator depth is increased to 3 layers.

**Optimization.** We train our models using the Adam and AdamO optimizers. For both optimizers, we use standard momentum parameters $\beta_1 = 0.9$ and $\beta_2 = 0.999$, a numerical stability term $\epsilon = 1 \times 10^{-4}$, and no weight decay. The learning rates are distinct across components: the critic network is trained with a learning rate of $1 \times 10^{-4}$, while all other components utilize a learning rate of $3 \times 10^{-4}$.

**Hyperparameters.** Task-specific hyperparameters are adjusted according to the dataset characteristics. Specifically, for the Locomotion (10k samples) and Adroit (human, cloned) datasets, we set the regularization coefficient $\kappa = 1$ and the temperature $\tau = 0.05$. Conversely, for the Kitchen, Antmaze, and Adroit (expert) tasks, we adopt $\kappa = 1 \times 10^{-4}$ and $\tau = 0$.

**Evaluation Metrics.** All experiments are conducted over 3-5 random seeds. To facilitate cross-domain aggregation, we report the standard D4RL normalized score:

$$\text{Score} = 100 \times \frac{R - R_{\text{random}}}{R_{\text{expert}} - R_{\text{random}}}, \tag{21}$$

where $R$ denotes the learned policy's return, while $R_{\text{random}}$ and $R_{\text{expert}}$ represent the reference scores for random and expert policies, respectively (Fu et al., 2020).

### A.3. Baseline Details

In this section, we provide detailed implementation references for the baseline methods. To ensure reproducibility, we utilize official or widely recognized community implementations.

**Offline RL Algorithms.** To maintain a strictly controlled comparison across standard offline RL paradigms, we utilize the unified implementations provided by the CORL library (Tarasov et al., 2022) available at https://github.com/tinkoff-ai/CORL. This codebase is used for the following methods:

- *TD3+BC* (Fujimoto & Gu, 2021): A minimalist baseline combining TD3 with a behavior cloning regularization term.

- *IQL* (Kostrikov et al., 2022): An in-sample learning approach utilizing expectile regression to avoid querying out-of-distribution actions.

- *ReBRAC* (Tarasov et al., 2023): An improved version of the minimalist approach that revisits network architecture and normalization choices.

For recent state-of-the-art (SOTA) methods, we utilize their official repositories:

- *ACTIVE* (Chen et al., 2025): A recent in-sample method employing actor-critic temperature adjustment. Source: `https://openreview.net/attachment?id=qiluFujVc8&name=supplementary_material`

- *PARS* (Kim et al., 2025): A method focused on robustness via reward scaling and penalizing infeasible actions. Source: `https://github.com/LGAI-Research/pars`

- *SQOG* (Yao et al., 2025): An approach utilizing convex hulls and smooth Bellman operators for OOD generalization. Source: `https://github.com/yqpqry/SQOG`

**Regularization Modules.** We compare against standard architectural and algorithmic regularization techniques:

- *DR3* (Kumar et al., 2022): Explicit regularization of the value function rank.

- *Normalization*: We evaluate standard normalization layers including Batch Normalization (BN), Layer Normalization (LN), Weight Normalization (WN), and Spectral Normalization (SN), implemented using standard *PyTorch* modules.

**Optimizer Baselines.** In this paper, we benchmark against diverse optimizers.

- *Standard Optimizers*: We use the official PyTorch implementations for *SGD* (Robbins & Monro, 1951), *Adam* (Kingma & Ba, 2017), and *AdamW* (Loshchilov & Hutter, 2019). Source: `https://github.com/pytorch/pytorch`

- *AdamC* (Liang et al., 2025b): Also known as C-AdamW, employing cautious momentum. Source: `https://github.com/kyleliang919/C-Optim`

- *Kron* (Castanyer et al., 2025): A parameter-efficient optimizer utilizing Kronecker-factored gradient preconditioners. Source: `https://github.com/roger-creus/stable-deep-rl-at-scale`

- *TRAC* (Muppidi et al., 2024): An adaptive optimizer for managing non-stationarity. Source: `https://github.com/redsnic/torch_erf`

- *Resetting* (Asadi et al., 2023): A technique involving the periodic resetting of the parameters of the optimizer. Pseudo-code provided in: `https://openreview.net/pdf?id=AnFUgNC3Yc`

### A.4. Main Experimental Analysis

We further provide a comprehensive empirical evaluation to validate our theoretical framework. Our analysis is structured to peel back the layers of AdamO's performance, moving from broad generalization capabilities to fine-grained mechanistic verification and practical robustness. Specifically, we center our evaluation around five core inquiries: Notably, we ensure comprehensive coverage of both Q-learning and SARSA paradigms. We employ TD3+BC and IQL as their respective representatives, applying both algorithms across all verification stages to ensure rigorous consistency.

- *Q1: Generalization.* Can AdamO consistently outperform baselines across diverse tasks and algorithmic backbones, verifying its universality (Appendix A.4.1)?

- *Q2: Comparative Advantage.* How does AdamO position itself against strictly strictly architectural (e.g., LayerNorm) or objective-based (e.g., regularization) remedies (Appendix A.4.2)?

- *Q3: Sensitivity & Robustness.* Is AdamO resilient to hyperparameter variations ($\kappa, \tau$) and batch-size scaling, making it a reliable choice for practitioners (Appendix A.4.3)?

- *Q4: Cost-Benefit Analysis.* Does AdamO incur a manageable computational overhead relative to the stability gains it provides (Appendix A.4.4)?

- *Q5: Mechanistic Verification.* Does the optimizer **empirically enforce** the spectral constraints (Hurwitz condition) on the TD operator as theoretically derived, thereby actively preventing collapse (Appendix A.4.5)?

### A.4.1. GENERALIZATION ANALYSIS (Q1)

We evaluate the consistent performance benefits of AdamO by integrating it into fundamentally different algorithmic backbones across a broad spectrum of environments. As illustrated in Figure 7, AdamO provides a robust performance boost regardless of the underlying reinforcement learning paradigm. Specifically, we observe significant improvements in both IQL and TD3+BC. These two methods represent SARSA-style in-sample learning and Q-learning style policy constraints, respectively. The radar charts reveal that AdamO consistently expands the performance frontier across diverse D4RL domains, including *MuJoCo locomotion*, *AntMaze* navigation, and *Adroit* dexterous manipulation. For instance, AdamO achieves dominant scores in challenging tasks like *Pen-Human* and *AntMaze-Large* where standard Adam often fails to learn meaningful policies. This robust plug-and-play nature allows AdamO to suppress value instability at the optimization level across fundamentally different temporal-difference structures. Detailed percentage improvements across an even wider array of state-of-the-art algorithms are provided in Table 1.

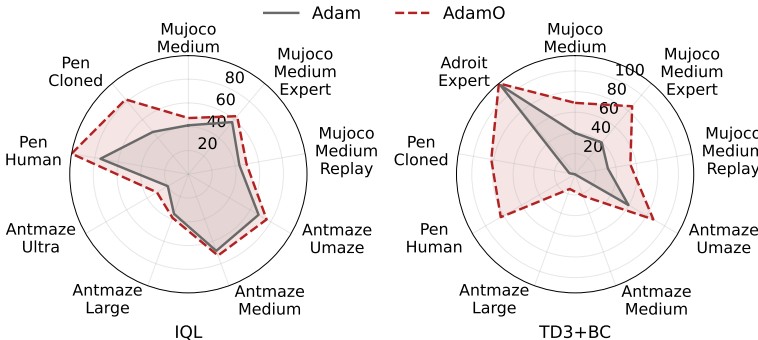

*Figure 7.* Radar charts visualizing the aggregated normalized scores across diverse D4RL domains.

### A.4.2. COMPARATIVE ADVANTAGE (Q2)

We evaluate the performance of our proposed AdamO optimizer against a robust set of architectural interventions and optimization baselines to determine the efficacy of an optimizer-level fix for value collapse.

**Comparison with Normalization and Regularization.** As demonstrated in Table 3, AdamO consistently outperforms or matches traditional function-space stabilizers. While techniques such as Layer Normalization (+LN) and Spectral Normalization (+SN) provide significant stability gains over the vanilla "Base" backbones, their performance often varies depending on the task and algorithm. For instance, while +SN achieves a high score of 95.9 on Walker2d-medium-expert with TD3+BC, AdamO reaches a superior 98.5. More importantly, AdamO exhibits higher stability across fundamentally different paradigms, such as in the AntMaze domain, where it maintains the highest average scores for both TD3+BC (41.6) and IQL (54.6). Compared to explicit rank regularization like +DR3, which requires additional computational overhead, AdamO achieves better results across almost all Locomotion and AntMaze tasks by directly suppressing instability within the optimization dynamics. To further isolate optimizer-level decoupling from ordinary orthogonality regularization (Brock et al., 2017; Bansal et al., 2018), Table 2 compares AdamO with two naive orthogonality variants: adding $r_t$ to the TD loss and adding $r_t$ directly on top of Adam.

**Comparison with Stability-Focused Optimizers.** Table 5 and Table 4 provide a detailed comparison between AdamO and other advanced optimizers. Standard first-order methods like SGD and Adam often struggle with value divergence in challenging offline environments, particularly in the AntMaze-large and Adroit-human tasks. Specialized optimizers like TRAC, Kron, and C-AdamW improve performance by managing non-stationarity or curvature, yet they remain susceptible to bootstrapping-induced collapse. AdamO significantly surpasses these baselines, achieving an average score of 66.1 in

*Table 2.* Naive orthogonality baselines with TD3+BC. Naive orthogonality helps over Adam in several collapse-prone tasks, but remains consistently below AdamO, supporting that the gain comes from decoupled optimizer design rather than orthogonality alone.

| Task | Adam | Adam + loss penalty $r_t$ | Adam + naive $r_t$ | AdamO |
|------|------|---------------------------|--------------------|-------|
| AntMaze-umaze-div | 47.0 | 61.3 | 60.2 | **82.2** |
| AntMaze-med-div | 0.2 | 1.1 | 8.7 | **16.4** |
| Walker2d-me | 22.4 | 18.7 | 40.8 | **98.5** |
| Pen-human | -4.1 | -1.4 | 24.8 | **83.1** |
| Pen-cloned | 5.6 | 9.7 | 26.3 | **82.4** |

Locomotion and 93.4 on the Pen subset using the TD3+BC backbone, compared to the next best optimizer, Kron, which scores 58.0 and 67.6 respectively. The stability benefits are further highlighted by the reduced standard deviations across multiple random seeds. In the challenging AntMaze-umaze task, AdamO achieves $92.5 \pm 1.2$, demonstrating much tighter convergence than the baseline Adam ($72.2 \pm 13.1$). By decoupling the orthogonality correction from the task-gradient descent, AdamO effectively mitigates the "deadly triad" instabilities that persist even when using other state-of-the-art optimizers. These results suggest that addressing value collapse at the optimizer level provides a more robust and principled solution than relying on heuristic architectural adjustments or generic curvature approximations.

A.4.3. SENSITIVITY AND ROBUSTNESS (Q3)

We investigate the sensitivity of AdamO to its core hyperparameters, including the orthogonality coefficient $\kappa$ and the conflict budget $\tau$. Specifically, we analyze how these parameters balance the trade-off between enforcing geometric stability and preserving the optimization trajectory of the base algorithm.

**Ablation Study on Conflict Budget $\tau$.** Figure 9 illustrates the learning dynamics on the Pen-Cloned and Pen-Human tasks under different budget constraints. We observe that a small positive budget $\tau = 0.05$ enables the optimizer to correct the spectral geometry even when the current noisy gradient opposes the orthogonality direction. The strict conflict-free mode $\tau = 0$ prevents degradation of the immediate task loss but may be too conservative to arrest collapse in highly unstable early training phases. Conversely, an unconstrained correction with $\tau = 100$ allows the orthogonality drift to dominate the update and destroys the task learning signal. This confirms that a modest controlled budget provides the best balance for challenging domains.

**Sensitivity to Orthogonality Coefficient $\kappa$.** Figure 8 presents a sweep of $\kappa$ values across multiple domains for both TD3+BC and IQL backbones. The results exhibit a distinct pattern consistent with our theoretical analysis in Appendix D.5. When $\kappa$ is too small (e.g. $< 10^{-5}$ in unstable tasks) the optimizer behaves like standard Adam and fails to prevent value collapse in challenging environments such as Adroit-human or sparse-data Locomotion. As $\kappa$ increases, performance improves until it reaches a critical threshold. Beyond this point, performance degrades because large corrections violate the local approximation bounds required for Adam's moment statistics to remain valid under finite step sizes.

**Task-Dependent Configuration.** Our hyperparameter selection follows a principled logic based on dataset stability. For domains that are empirically prone to collapse such as Locomotion-10k and Adroit-human we utilize a stronger correction $\kappa = 1$ and $\tau = 0.05$. This aggressive setting is necessary to actively suppress the rapid error amplification characteristic of these tasks. In contrast we adopt conservative settings $\kappa = 10^{-4}$ and $\tau = 0$ for inherently stable tasks like Kitchen and AntMaze. In these regimes the priority is to preserve the adaptive statistics of Adam since the spectral radius is naturally well-behaved. This adaptation ensures AdamO is effective across the full spectrum of offline RL challenges.

*Table 3.* Normalized average scores on Benchmarks. We compare our method against two backbone algorithms and their variants with various regularization techniques. The "Base" column corresponds to the vanilla TD3+BC or IQL score, respectively.

| Task Name | Base | +LN | +BN | +WN | +SN | +DR3 | AdamO (Ours) |
|---|---|---|---|---|---|---|---|
| *Backbone: TD3+BC* | | | | | | | |
| *MuJoCo Locomotion (10k samples)* | | | | | | | |
| HalfCheetah-medium | 35.9 | 50.6 | 44.1 | 38.8 | 52.3 | 47.0 | **53.5** |
| HalfCheetah-medium-replay | 39.1 | 45.2 | 42.5 | 40.1 | 44.9 | 43.2 | **46.2** |
| HalfCheetah-medium-expert | 33.5 | 56.4 | 44.9 | 40.2 | 55.0 | 47.8 | **60.1** |
| Hopper-medium | 40.7 | 67.3 | 55.6 | 44.9 | 71.7 | 64.2 | **75.5** |
| Hopper-medium-replay | 21.3 | 47.4 | 36.5 | 24.7 | 46.2 | 41.9 | **55.8** |
| Hopper-medium-expert | 32.6 | 80.6 | 46.9 | 37.8 | 75.8 | 64.4 | **84.2** |
| Walker2d-medium | 21.2 | 60.8 | 36.3 | 28.7 | 59.3 | 50.4 | **68.4** |
| Walker2d-medium-replay | 19.3 | 44.9 | 32.4 | 26.4 | 49.5 | 39.5 | **52.5** |
| Walker2d-medium-expert | 22.4 | 84.1 | 46.1 | 34.0 | 95.9 | 69.5 | **98.5** |
| Ant-medium | 61.7 | 78.2 | 69.5 | 63.4 | 75.2 | 74.0 | **79.1** |
| Ant-medium-replay | 49.3 | 62.7 | 54.7 | 52.3 | 59.4 | 59.2 | **62.8** |
| Ant-medium-expert | 72.0 | 101.3 | **103.6** | 78.3 | 83.6 | 68.7 | 100.9 |
| *AntMaze* | | | | | | | |
| AntMaze-Umaze | 72.2 | 89.5 | 80.1 | 75.1 | 91.8 | 84.8 | **92.5** |
| AntMaze-Umaze-diverse | 47.0 | 78.1 | 64.2 | 52.3 | 73.5 | 69.9 | **82.2** |
| AntMaze-medium-play | 0.3 | 25.4 | 14.0 | 6.0 | 24.8 | 20.0 | **28.5** |
| AntMaze-medium-diverse | 0.2 | 12.7 | 5.5 | 1.8 | 13.6 | 12.6 | **16.4** |
| AntMaze-Large-Play | 0.0 | 12.9 | 5.5 | 2.1 | 10.6 | 9.2 | **13.7** |
| AntMaze-Large-diverse | 0.0 | 14.1 | 2.3 | **16.5** | 12.8 | 10.5 | **16.5** |
| *Adroit* | | | | | | | |
| Pen-human | -4.1 | 69.3 | 29.3 | 12.0 | 69.5 | 49.7 | **83.1** |
| Pen-cloned | 5.6 | 76.1 | 38.6 | 18.5 | 65.6 | 58.0 | **82.4** |
| Pen-expert | 109.7 | 114.1 | 111.4 | 110.5 | 114.3 | 113.1 | **114.8** |
| Relocate-expert | **106.2** | 105.7 | 105.9 | 106.1 | 105.7 | 105.8 | 105.6 |
| Hammer-expert | 125.0 | 126.6 | 125.7 | 125.4 | 126.3 | 126.0 | **126.7** |
| Door-expert | 109.4 | 120.2 | 110.2 | 108.9 | **121.7** | 116.4 | 112.3 |
| *Backbone: IQL* | | | | | | | |
| *MuJoCo Locomotion (10k samples)* | | | | | | | |
| HalfCheetah-medium | 29.7 | 25.3 | 31.6 | 31.1 | 27.1 | 30.6 | **35.2** |
| HalfCheetah-medium-replay | 32.7 | **42.0** | 16.0 | 32.2 | 28.7 | 24.4 | 38.4 |
| HalfCheetah-medium-expert | 48.1 | 49.9 | 51.1 | 25.7 | 64.3 | **65.1** | 54.6 |
| Hopper-medium | 38.9 | 30.6 | **59.3** | 17.2 | 34.8 | 40.0 | 45.1 |
| Hopper-medium-replay | 46.6 | 42.3 | 40.2 | 30.2 | 46.5 | 41.9 | **52.9** |
| Hopper-medium-expert | 66.5 | **74.7** | 42.5 | 41.6 | 73.7 | 50.1 | 73.2 |
| Walker2d-medium | 54.9 | 46.4 | 32.1 | 35.3 | **65.6** | 33.5 | 61.5 |
| Walker2d-medium-replay | 51.4 | **59.3** | 38.2 | 39.6 | 41.5 | 44.9 | 57.8 |
| Walker2d-medium-expert | 57.3 | 52.4 | 59.4 | 36.5 | 59.5 | 44.5 | **63.7** |
| *AntMaze* | | | | | | | |
| AntMaze-umaze | 80.5 | 70.9 | 78.2 | 65.2 | **85.2** | 76.5 | 83.8 |
| AntMaze-umaze-diverse | 55.8 | 56.7 | 61.7 | 46.8 | 51.5 | 44.6 | **68.2** |
| AntMaze-medium-play | 70.4 | 64.6 | 65.1 | 63.4 | **75.3** | 47.4 | 70.1 |
| AntMaze-medium-diverse | 66.9 | 67.6 | 71.5 | 67.6 | **76.3** | 49.9 | 75.5 |
| AntMaze-large-play | 38.5 | 36.9 | 34.2 | **42.6** | 38.5 | 39.6 | 41.8 |
| AntMaze-large-diverse | 32.4 | 26.7 | 24.2 | 27.0 | 24.9 | 32.7 | **36.1** |
| AntMaze-ultra-diverse | 19.0 | 25.4 | 35.0 | 29.9 | **36.9** | 24.3 | 31.1 |
| AntMaze-ultra-play | 21.0 | **31.9** | 31.3 | 27.5 | 23.3 | 28.5 | 29.9 |
| *Adroit* | | | | | | | |
| Pen-human | 75.1 | 97.7 | 78.6 | 90.2 | 95.6 | 83.0 | **99.8** |
| Pen-cloned | 46.5 | 74.9 | 55.4 | 51.1 | 53.2 | 80.7 | **82.2** |
| Door-human | 3.5 | **5.4** | 3.7 | 3.1 | 3.4 | 3.7 | 4.8 |
| Door-cloned | 3.3 | 4.4 | 2.9 | 3.7 | **6.4** | 4.2 | 4.5 |
| Hammer-human | 1.9 | **4.3** | 2.4 | 2.2 | 2.4 | 4.1 | 3.1 |
| Hammer-cloned | 1.7 | 2.3 | 1.6 | 2.3 | **3.3** | 3.0 | 3.2 |
| Relocate-human | 0.1 | **0.7** | 0.1 | 0.3 | 0.5 | 0.6 | 0.5 |
| Relocate-cloned | 0.0 | 0.0 | 0.0 | 0.0 | 0.0 | 0.0 | 0.0 |

*Table 4.* Performance comparison on benchmarks using IQL. We report the mean scores $\pm$ standard deviation for individual tasks, and the arithmetic mean for averages. **AdamO (Ours)** shows superior performance and improved stability across domains.

| Task Name | SGD | Adam | AdamW | C-AdamW | Resetting | TRAC | Kron | AdamO (Ours) |
|---|---|---|---|---|---|---|---|---|
| | | | | *Mujoco Locomotion (10k samples)* | | | | |
| halfcheetah-m | $22.6 \pm 5.4$ | $29.7 \pm 3.1$ | $29.0 \pm 3.5$ | $30.6 \pm 2.8$ | $33.4 \pm 2.2$ | $32.8 \pm 3.0$ | $34.9 \pm 2.5$ | $\mathbf{35.2 \pm 1.8}$ |
| halfcheetah-mr | $23.6 \pm 6.1$ | $32.7 \pm 4.2$ | $34.1 \pm 4.0$ | $36.5 \pm 3.5$ | $34.8 \pm 3.9$ | $\mathbf{39.0 \pm 4.1}$ | $35.5 \pm 3.3$ | $38.4 \pm 2.9$ |
| halfcheetah-me | $37.7 \pm 8.2$ | $48.1 \pm 5.3$ | $45.9 \pm 6.1$ | $49.5 \pm 5.0$ | $49.3 \pm 4.8$ | $\mathbf{54.9 \pm 5.2}$ | $53.5 \pm 4.5$ | $54.6 \pm 3.6$ |
| hopper-m | $33.2 \pm 7.5$ | $38.9 \pm 4.1$ | $42.0 \pm 4.5$ | $39.3 \pm 5.2$ | $39.6 \pm 4.0$ | $44.8 \pm 4.8$ | $42.0 \pm 3.9$ | $\mathbf{45.1 \pm 3.2}$ |
| hopper-mr | $37.8 \pm 8.0$ | $46.6 \pm 6.8$ | $46.6 \pm 6.5$ | $48.6 \pm 5.1$ | $48.9 \pm 5.2$ | $51.0 \pm 6.0$ | $50.3 \pm 5.1$ | $\mathbf{52.9 \pm 4.0}$ |
| hopper-me | $54.9 \pm 10.2$ | $66.5 \pm 8.4$ | $65.7 \pm 9.1$ | $70.3 \pm 7.5$ | $67.0 \pm 8.0$ | $69.6 \pm 7.2$ | $69.3 \pm 7.5$ | $\mathbf{73.2 \pm 6.1}$ |
| walker2d-m | $44.4 \pm 9.5$ | $54.9 \pm 6.2$ | $54.5 \pm 6.8$ | $58.1 \pm 5.8$ | $58.4 \pm 5.1$ | $57.1 \pm 6.5$ | $58.4 \pm 5.9$ | $\mathbf{61.5 \pm 4.8}$ |
| walker2d-mr | $40.9 \pm 8.8$ | $51.4 \pm 7.0$ | $51.4 \pm 7.2$ | $53.5 \pm 6.1$ | $55.2 \pm 5.8$ | $55.3 \pm 6.3$ | $\mathbf{57.8 \pm 5.2}$ | $57.8 \pm 4.9$ |
| walker2d-me | $44.4 \pm 9.1$ | $57.3 \pm 8.5$ | $60.7 \pm 7.9$ | $57.4 \pm 7.2$ | $63.0 \pm 6.5$ | $62.6 \pm 6.8$ | $\mathbf{64.7 \pm 6.0}$ | $63.7 \pm 5.2$ |
| *Locomotion Avg.* | 37.7 | 47.3 | 47.8 | 49.3 | 50.0 | 51.9 | 51.8 | **53.6** |
| | | | | *AntMaze* | | | | |
| antmaze-umaze | $66.2 \pm 15.4$ | $80.5 \pm 9.2$ | $82.9 \pm 8.5$ | $79.8 \pm 10.1$ | $80.4 \pm 8.8$ | $81.5 \pm 7.5$ | $\mathbf{84.4 \pm 6.1}$ | $83.8 \pm 5.7$ |
| antmaze-umaze-di | $52.9 \pm 12.8$ | $55.8 \pm 10.5$ | $55.3 \pm 11.2$ | $56.5 \pm 9.8$ | $58.4 \pm 9.5$ | $63.1 \pm 8.2$ | $65.5 \pm 7.8$ | $\mathbf{68.2 \pm 6.5}$ |
| antmaze-med-play | $63.0 \pm 14.2$ | $70.4 \pm 11.5$ | $69.9 \pm 12.1$ | $\mathbf{70.5 \pm 10.8}$ | $70.5 \pm 9.9$ | $69.7 \pm 10.2$ | $69.8 \pm 9.5$ | $70.1 \pm 8.2$ |
| antmaze-med-div | $55.3 \pm 13.5$ | $66.9 \pm 9.8$ | $71.3 \pm 9.2$ | $69.1 \pm 10.5$ | $72.3 \pm 8.5$ | $73.5 \pm 7.9$ | $71.2 \pm 8.8$ | $\mathbf{75.5 \pm 7.2}$ |
| antmaze-large-play | $30.0 \pm 10.5$ | $38.5 \pm 8.2$ | $38.7 \pm 8.5$ | $39.1 \pm 7.8$ | $40.5 \pm 7.2$ | $39.3 \pm 7.5$ | $40.5 \pm 6.8$ | $\mathbf{41.8 \pm 5.9}$ |
| antmaze-large-div | $26.6 \pm 9.8$ | $32.4 \pm 7.5$ | $32.6 \pm 7.2$ | $33.4 \pm 7.0$ | $34.0 \pm 6.5$ | $34.4 \pm 6.8$ | $34.8 \pm 6.2$ | $\mathbf{36.1 \pm 5.5}$ |
| antmaze-ultra-div | $15.4 \pm 8.5$ | $19.0 \pm 6.2$ | $20.2 \pm 6.5$ | $20.4 \pm 5.8$ | $27.6 \pm 5.5$ | $26.4 \pm 5.2$ | $28.6 \pm 4.8$ | $\mathbf{31.1 \pm 4.1}$ |
| antmaze-ultra-play | $14.9 \pm 8.1$ | $21.0 \pm 6.5$ | $18.4 \pm 7.1$ | $23.2 \pm 6.2$ | $26.1 \pm 5.8$ | $23.1 \pm 6.0$ | $25.6 \pm 5.4$ | $\mathbf{29.9 \pm 4.5}$ |
| *AntMaze Avg.* | 40.6 | 48.1 | 48.7 | 49.0 | 51.2 | 51.4 | 52.5 | **54.6** |
| | | | | *Adroit* | | | | |
| pen-human | $57.1 \pm 25.4$ | $75.1 \pm 18.2$ | $83.7 \pm 12.5$ | $94.4 \pm 5.8$ | $83.6 \pm 15.1$ | $95.2 \pm 4.2$ | $94.6 \pm 4.5$ | $\mathbf{99.8 \pm 0.3}$ |
| pen-cloned | $30.5 \pm 15.2$ | $46.5 \pm 12.8$ | $57.2 \pm 11.5$ | $61.9 \pm 10.2$ | $63.8 \pm 9.5$ | $66.1 \pm 8.8$ | $70.9 \pm 8.1$ | $\mathbf{82.2 \pm 5.4}$ |
| door-human | $3.3 \pm 2.5$ | $3.5 \pm 2.8$ | $4.0 \pm 2.1$ | $4.5 \pm 2.5$ | $4.2 \pm 2.2$ | $\mathbf{4.8 \pm 2.9}$ | $4.1 \pm 2.0$ | $\mathbf{4.8 \pm 1.5}$ |
| door-cloned | $3.0 \pm 2.1$ | $3.3 \pm 2.4$ | $4.2 \pm 2.5$ | $3.8 \pm 2.2$ | $3.8 \pm 2.0$ | $4.3 \pm 2.6$ | $4.2 \pm 2.1$ | $\mathbf{4.5 \pm 1.8}$ |
| hammer-human | $1.3 \pm 1.2$ | $1.9 \pm 1.5$ | $1.6 \pm 1.1$ | $2.8 \pm 2.1$ | $2.0 \pm 1.8$ | $2.6 \pm 2.0$ | $2.9 \pm 1.9$ | $\mathbf{3.1 \pm 1.6}$ |
| hammer-cloned | $1.1 \pm 0.9$ | $1.7 \pm 1.2$ | $1.8 \pm 1.4$ | $1.8 \pm 1.2$ | $2.5 \pm 1.5$ | $2.6 \pm 1.8$ | $3.1 \pm 2.0$ | $\mathbf{3.2 \pm 1.5}$ |
| relocate-human | $0.0 \pm 0.0$ | $0.1 \pm 0.1$ | $0.2 \pm 0.2$ | $0.1 \pm 0.1$ | $0.3 \pm 0.2$ | $\mathbf{0.6 \pm 0.4}$ | $0.2 \pm 0.2$ | $0.5 \pm 0.3$ |
| relocate-cloned | $0.0 \pm 0.0$ | $0.0 \pm 0.0$ | $0.1 \pm 0.1$ | $0.1 \pm 0.1$ | $0.1 \pm 0.1$ | $\mathbf{0.2 \pm 0.2}$ | $\mathbf{0.2 \pm 0.2}$ | $0.0 \pm 0.0$ |
| *Adroit Avg.* | 12.0 | 16.5 | 19.1 | 21.2 | 20.0 | 22.1 | 22.5 | **24.8** |
| | | | | *Kitchen* | | | | |
| kitchen-complete | $50.5 \pm 4.3$ | $62.6 \pm 9.5$ | $58.5 \pm 7.5$ | $56.4 \pm 6.3$ | $60.6 \pm 2.4$ | $59.6 \pm 2.4$ | $61.9 \pm 1.5$ | $\mathbf{64.8 \pm 8.7}$ |
| kitchen-mixed | $52.6 \pm 6.3$ | $54.9 \pm 7.3$ | $57.1 \pm 1.2$ | $58.4 \pm 9.6$ | $55.9 \pm 8.4$ | $61.1 \pm 2.9$ | $60.7 \pm 2.6$ | $\mathbf{62.3 \pm 2.6}$ |
| kitchen-partial | $42.6 \pm 3.7$ | $48.7 \pm 5.7$ | $50.3 \pm 4.8$ | $52.5 \pm 3.6$ | $56.8 \pm 6.4$ | $58.0 \pm 2.2$ | $47.9 \pm 3.6$ | $\mathbf{61.7 \pm 4.3}$ |
| Kitchen Avg. | 48.6 | 55.4 | 55.3 | 55.8 | 57.8 | 59.6 | 56.8 | **62.9** |

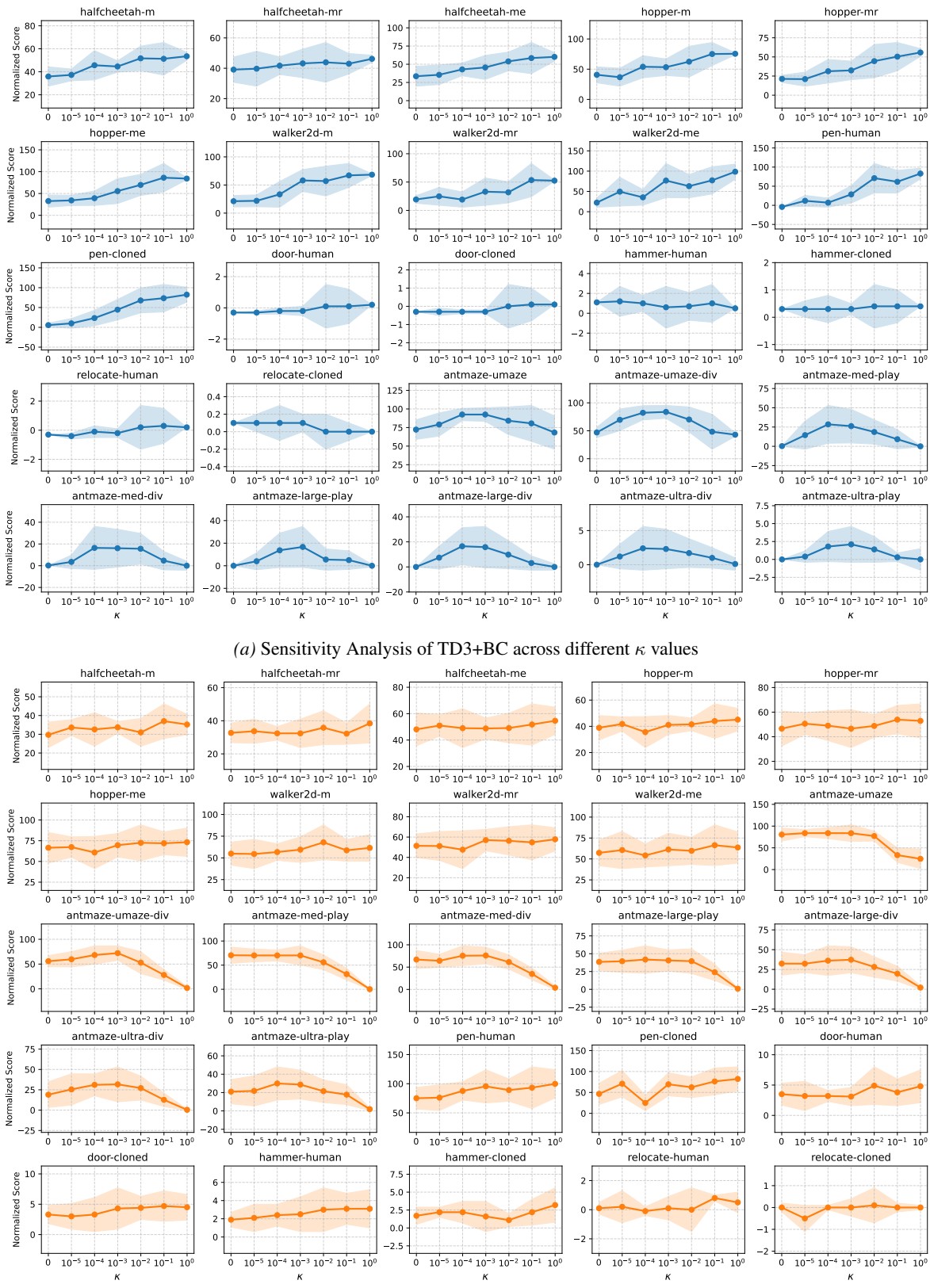

*(a)* Sensitivity Analysis of TD3+BC across different $\kappa$ values

*(b)* Sensitivity Analysis of IQL across different $\kappa$ values

*Figure 8.* Sensitivity Analysis of various algorithms across different $\kappa$ values.

*Table 5.* Performance comparison on benchmarks using TD3+BC. We report the mean scores $\pm$ standard deviation for individual tasks. **AdamO (Ours)** shows not only superior performance but also improved stability across domains.

| Task Name | SGD | Adam | AdamW | C-AdamW | Resetting | TRAC | Kron | AdamO (Ours) |
|---|---|---|---|---|---|---|---|---|
| | | | | *AntMaze* | | | | |
| AntMaze-umaze | $65.5 \pm 11.5$ | $72.2 \pm 13.1$ | $79.4 \pm 2.5$ | $88.5 \pm 6.8$ | $78.5 \pm 1.9$ | $85.1 \pm 4.5$ | $89.2 \pm 2.0$ | $\mathbf{92.5 \pm 1.2}$ |
| AntMaze-umaze-div | $42.3 \pm 9.5$ | $47.0 \pm 9.3$ | $51.7 \pm 4.8$ | $59.8 \pm 6.2$ | $58.5 \pm 4.9$ | $68.4 \pm 3.9$ | $75.5 \pm 5.7$ | $\mathbf{82.2 \pm 3.5}$ |
| AntMaze-med-play | $0.2 \pm 0.1$ | $0.3 \pm 0.3$ | $0.3 \pm 0.3$ | $0.4 \pm 1.4$ | $5.6 \pm 2.1$ | $8.5 \pm 1.5$ | $9.2 \pm 2.5$ | $\mathbf{28.5 \pm 2.1}$ |
| AntMaze-med-div | $0.2 \pm 0.1$ | $0.2 \pm 0.2$ | $0.3 \pm 0.3$ | $0.4 \pm 1.4$ | $8.5 \pm 0.5$ | $5.2 \pm 1.0$ | $6.8 \pm 1.2$ | $\mathbf{16.4 \pm 4.5}$ |
| AntMaze-large-play | $0.0 \pm 0.0$ | $0.0 \pm 0.0$ | $0.0 \pm 0.0$ | $0.0 \pm 0.0$ | $1.5 \pm 0.3$ | $2.4 \pm 0.5$ | $7.5 \pm 1.2$ | $\mathbf{13.7 \pm 8.2}$ |
| AntMaze-large-div | $0.0 \pm 0.0$ | $0.0 \pm 0.0$ | $0.0 \pm 0.0$ | $0.0 \pm 0.0$ | $9.2 \pm 6.8$ | $8.5 \pm 0.2$ | $4.1 \pm 1.8$ | $\mathbf{16.5 \pm 5.4}$ |
| *AntMaze Avg.* | 18.0 | 20.0 | 22.0 | 24.8 | 27.0 | 29.7 | 32.1 | **41.6** |
| | | | | *Mujoco Locomotion (10k samples)* | | | | |
| HalfCheetah-m | $24.1 \pm 0.3$ | $35.9 \pm 8.4$ | $28.5 \pm 0.3$ | $31.4 \pm 0.2$ | $35.1 \pm 0.6$ | $42.5 \pm 1.7$ | $49.2 \pm 0.9$ | $\mathbf{53.5 \pm 3.5}$ |
| HalfCheetah-mr | $26.5 \pm 1.1$ | $39.1 \pm 8.3$ | $31.5 \pm 0.3$ | $34.2 \pm 0.3$ | $34.8 \pm 0.3$ | $38.5 \pm 1.5$ | $42.1 \pm 1.4$ | $\mathbf{46.2 \pm 2.4}$ |
| HalfCheetah-me | $21.2 \pm 2.7$ | $33.5 \pm 13.6$ | $25.8 \pm 0.6$ | $29.5 \pm 0.6$ | $35.6 \pm 3.1$ | $45.2 \pm 10.5$ | $52.8 \pm 3.8$ | $\mathbf{60.1 \pm 5.9}$ |
| Hopper-m | $28.5 \pm 5.1$ | $40.7 \pm 13.2$ | $34.5 \pm 4.5$ | $41.2 \pm 2.9$ | $45.6 \pm 0.2$ | $58.2 \pm 11.2$ | $68.9 \pm 1.4$ | $\mathbf{75.5 \pm 1.8}$ |
| Hopper-mr | $10.2 \pm 2.5$ | $21.3 \pm 4.7$ | $12.8 \pm 6.1$ | $15.6 \pm 8.2$ | $25.8 \pm 4.9$ | $35.6 \pm 2.9$ | $48.2 \pm 1.0$ | $\mathbf{55.8 \pm 5.1}$ |
| Hopper-me | $19.8 \pm 7.5$ | $32.6 \pm 13.9$ | $24.9 \pm 12.5$ | $28.5 \pm 8.9$ | $42.5 \pm 2.5$ | $55.8 \pm 13.1$ | $70.5 \pm 11.5$ | $\mathbf{84.2 \pm 2.5}$ |
| Walker2d-m | $10.1 \pm 3.5$ | $21.2 \pm 10.1$ | $12.5 \pm 2.8$ | $15.2 \pm 4.1$ | $30.5 \pm 2.5$ | $45.2 \pm 0.9$ | $58.9 \pm 1.9$ | $\mathbf{68.4 \pm 1.6}$ |
| Walker2d-mr | $8.5 \pm 4.2$ | $19.3 \pm 6.6$ | $10.2 \pm 9.2$ | $11.8 \pm 3.9$ | $22.5 \pm 2.0$ | $32.5 \pm 1.4$ | $45.6 \pm 3.1$ | $\mathbf{52.5 \pm 1.1}$ |
| Walker2d-me | $11.2 \pm 2.1$ | $22.4 \pm 11.2$ | $14.5 \pm 0.5$ | $18.2 \pm 1.1$ | $40.5 \pm 0.6$ | $65.2 \pm 0.5$ | $85.6 \pm 0.9$ | $\mathbf{98.5 \pm 2.7}$ |
| *Locomotion Avg.* | 17.8 | 29.6 | 21.7 | 25.1 | 34.8 | 46.5 | 58.0 | **66.1** |
| | | | | *Adroit* | | | | |
| Pen-human | $-4.5 \pm 0.3$ | $-4.1 \pm 0.2$ | $-3.8 \pm 3.4$ | $-2.5 \pm 2.2$ | $22.5 \pm 9.2$ | $45.2 \pm 7.1$ | $48.5 \pm 4.1$ | $\mathbf{83.1 \pm 6.5}$ |
| Pen-cloned | $5.1 \pm 4.6$ | $5.6 \pm 5.0$ | $6.2 \pm 5.6$ | $7.9 \pm 7.1$ | $24.5 \pm 6.5$ | $42.5 \pm 1.8$ | $52.8 \pm 2.2$ | $\mathbf{82.4 \pm 5.2}$ |
| Pen-expert | $102.5 \pm 7.3$ | $109.7 \pm 7.3$ | $115.2 \pm 21.3$ | $\mathbf{125.8 \pm 1.9}$ | $111.2 \pm 2.3$ | $113.0 \pm 9.2$ | $101.5 \pm 6.3$ | $114.8 \pm 2.1$ |
| *Pen Avg.* | 34.4 | 37.1 | 39.2 | 43.7 | 52.7 | 66.9 | 67.6 | **93.4** |

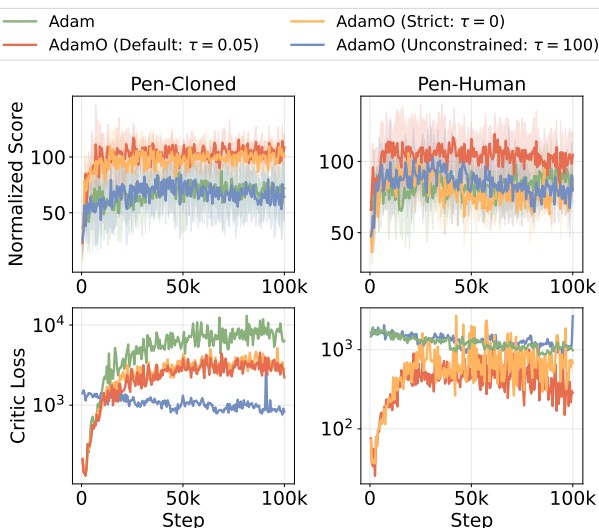

*Figure 9.* Ablation study on the conflict budget $\tau$. A moderate budget successfully prevents collapse while maintaining high performance.

### A.4.4. Cost-Benefit Analysis (Q4)

We assess the computational overhead of AdamO in terms of both memory footprint and wall-clock runtime. Regarding spatial efficiency, AdamO maintains a GPU memory usage of 4.4GB, which is identical to vanilla Adam and standard normalization techniques such as LN, BN, WN, and SN. Among all tested methods, only the explicit feature regularization (+DR3) incurs a higher memory cost of 4.8GB. Temporally, as shown in Table 6, AdamO requires 172 minutes for 1 million steps on the TD3+BC backbone. While this is a slight increase compared to the 144 minutes required by vanilla Adam, AdamO remains more efficient than other stability-oriented optimizers, including Resetting (173 min), Kron (184 min), C-AdamW (192 min), and TRAC (209 min). These results indicate that AdamO provides a favorable balance between improved critic stability and manageable computational costs, making it highly practical for large-scale offline RL training.

*Table 6.* Comparison of GPU memory usage and training runtime (TD3+BC) for 1 million steps.

| Method / Optimizer | GPU Memory (GB) | Runtime (min) |
|---|---|---|
| Base | 4.4 | 144 |
| +LN | 4.4 | 145 |
| +BN | 4.4 | 145 |
| +WN | 4.4 | 142 |
| +SN | 4.4 | 147 |
| +DR3 | 4.8 | 196 |
| SGD | 4.4 | 96 |
| Adam | 4.4 | 144 |
| AdamW | 4.4 | 157 |
| C-AdamW | 4.4 | 192 |
| Resetting | 4.4 | 173 |
| TRAC | 4.4 | 209 |
| Kron | 4.4 | 184 |
| **AdamO (Ours)** | **4.4** | **172** |

### A.4.5. Mechanistic Verification (Q5)

We investigate whether AdamO effectively maintains the Hurwitz condition of the TD operator $S$ in practice, thereby preventing value-function collapse.

**Global Stability across Tasks.** We first evaluate the training stability of the critic network across 20 D4RL benchmark tasks using the IQL algorithm. As illustrated in Figure 10, the baseline optimizer, Adam (blue curves), exhibits significant instability in numerous environments. Specifically, in challenging domains such as *AntMaze* and *Adroit* (e.g., `pen-cloned`, `door-human`), Adam frequently suffers from loss spikes and divergence, indicating a failure in value function approximation. In sharp contrast, AdamO (red curves) demonstrates superior stability, maintaining consistently lower critic losses throughout the training process. This empirical evidence suggests that AdamO effectively mitigates the optimization instability inherent in offline RL settings.

**Spectral Analysis and Hurwitz Condition.** To uncover the underlying mechanism of this stability, we conduct a fine-grained spectral analysis on the TD3+BC algorithm. Figure 11 visualizes the training dynamics across four representative tasks, tracking four metrics: normalized score, critic loss, and the maximum real $\text{Re}(\bar{\lambda})$ and imaginary $\text{Im}(\bar{\lambda})$ parts of the eigenvalues of the operator $S$. The results reveal a decisive correlation between spectral properties and performance.

We observe that value collapse in Adam (manifested by exploding critic loss and plummeting scores) coincides with the real part of the eigenvalues becoming significantly positive, e.g., reaching $10^{16}$ magnitude in *pen-human*. This clearly violates the Hurwitz condition, driving the linear system towards divergence. Conversely, AdamO successfully constrains the real part of the eigenvalues near zero ($\text{Re}(\lambda) \approx 0$) throughout training. By actively suppressing the positive growth of the real spectrum, AdamO maintains the TD operator within the Hurwitz stability region. Furthermore, AdamO also exhibits smaller imaginary components, indicating reduced oscillatory behavior in the error dynamics. These findings explicitly answer Q5:

***AdamO prevents collapse not by chance, but by effectively enforcing the Hurwitz condition of the TD operator in practice.***

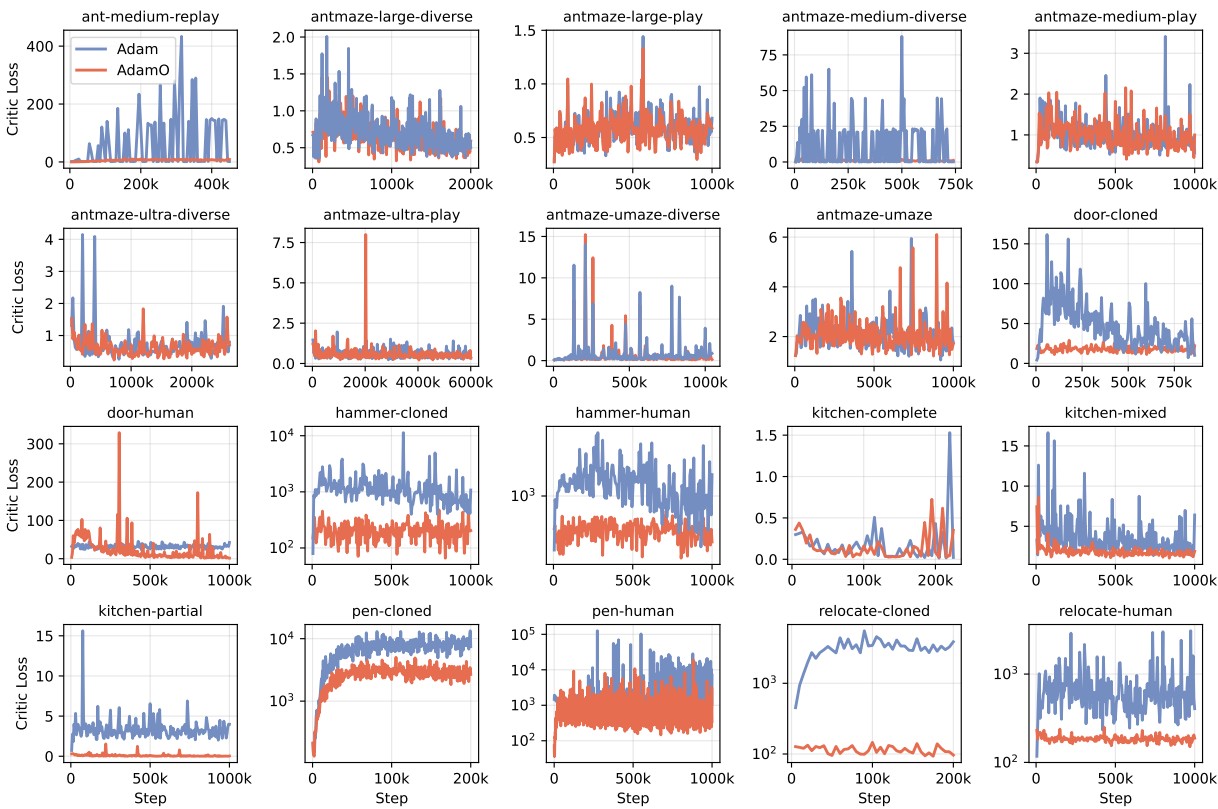

*Figure 10.* Critic loss across different tasks.

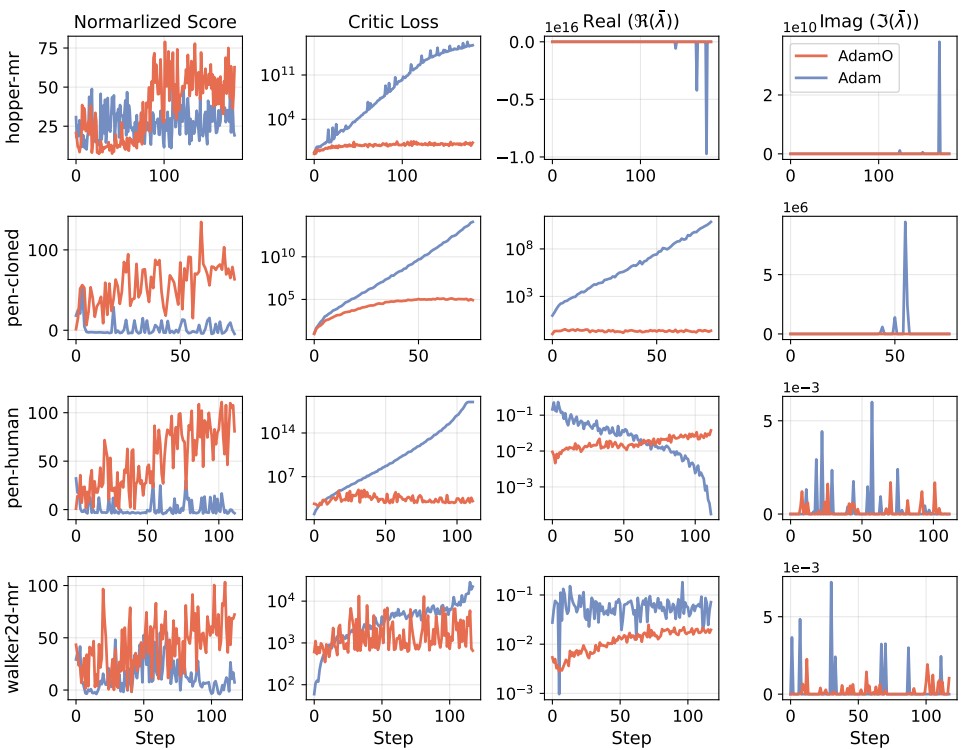

*Figure 11.* Training dynamics across four representative tasks.

### A.4.6. EXTENDED BENCHMARK AND CONTROL RESULTS

We include additional final experiments requested by reviewers to test whether AdamO remains useful beyond the main D4RL/model-free setting: model-based offline RL, newer OGBench and NeoRL2 benchmarks, optimizer compatibility, Monte Carlo critic calibration, and the role of conservative policy constraints.

**Model-Based Offline RL.** Table 7 replaces the critic optimizer with AdamO in MOPO (Yu et al., 2020) and MOBILE (Sun et al., 2023) using the OfflineRL-Kit implementation (Sun, 2023). The consistent gains show that AdamO also improves model-based offline RL methods that suffer from value-estimation instability.

*Table 7.* Model-based offline RL results on Adroit tasks. Replacing Adam with AdamO consistently improves MOPO and MOBILE.

| Adroit Task | MOPO–Adam | MOPO–AdamO | MOBILE–Adam | MOBILE–AdamO |
|---|---|---|---|---|
| pen-human | 9.6 | **42.3** | 25.7 | **49.4** |
| door-human | -0.3 | **-0.1** | -0.1 | **0.0** |
| hammer-human | 0.2 | **0.4** | 0.4 | **0.6** |
| pen-cloned | 50.8 | **64.2** | 64.7 | **72.5** |
| door-cloned | 12.9 | **15.0** | 19.5 | **23.2** |
| hammer-cloned | 0.4 | **0.7** | **1.1** | 1.0 |

**Newer Benchmarks.** Tables 8 and 9 extend the evidence beyond D4RL to NeoRL2 (Gao et al., 2025) and OGBench (Park et al., 2025). These results support that AdamO's critic-side stabilization remains beneficial under newer benchmark distributions and long-horizon/stitching-style tasks.

*Table 8.* NeoRL2 10k-subset results with TD3+BC. Each task is subsampled to 10,000 transitions when the original split is larger than 10k; Fusion is omitted because its full split contains only 2,000 transitions.

| Method | Pipeline | DMSD | Random-Friction Hopper | Rocket Recovery | Safety HalfCheetah |
|---|---|---|---|---|---|
| Adam | 6 | 15 | 13 | 51 | 60 |
| AdamO | **23** | **30** | **24** | **61** | **66** |

*Table 9.* Mean success rates (%) on representative OGBench tasks using the official OGBench environments/datasets and singletask variants. AdamO improves over Adam for both MOPO and MOBILE.

| OGBench Task | MOPO–Adam | MOPO–AdamO | MOBILE–Adam | MOBILE–AdamO |
|---|---|---|---|---|
| PointMaze-Large-Navigate | 31 | **37** | 42 | **48** |
| PointMaze-Large-Stitch | 9 | **17** | 24 | **31** |
| AntMaze-Large-Navigate | 16 | **29** | 33 | **44** |
| AntMaze-Large-Stitch | 7 | **16** | 15 | **25** |
| AntMaze-Large-Explore | 2 | **4** | 6 | **8** |
| HumanoidMaze-Large-Navigate | 4 | **13** | 11 | **20** |
| HumanoidMaze-Large-Stitch | 1 | **5** | 5 | **11** |

**Compatibility Beyond Adam.** Table 10 adds SGD-O, where the same orthogonality correction is applied to SGD. SGD-O improves substantially over SGD, indicating that the correction module is compatible with other first-order optimizers, while AdamO remains strongest because its design is matched to Adam's adaptive moments.

**Critic Calibration and Conservative Constraints.** Table 11 directly compares critic estimates with Monte Carlo returns rather than TD error alone. Adam's critic estimates can drift orders of magnitude away from realized returns, while AdamO keeps the gap small on the same collapse-prone tasks.

Table 12 further shows that AdamO and conservative offline RL constraints are complementary: TD3+AdamO stabilizes the critic and improves TD3, but without BC it still underperforms TD3+BC+AdamO because policy-support constraints address OOD exploitation rather than optimizer-side critic instability.

*Table 10.* Performance comparison on D4RL benchmarks using TD3+BC. We report mean normalized scores and include SGD-O to test whether the proposed correction generalizes beyond Adam.

| Task | SGD | SGD-O | Adam | AdamO |
|------|-----|-------|------|-------|
| AntMaze-umaze | 65.5 | 86.0 | 72.2 | **92.5** |
| AntMaze-umaze-div | 42.3 | 75.4 | 47.0 | **82.2** |
| AntMaze-med-play | 0.2 | 20.3 | 0.3 | **28.5** |
| AntMaze-med-div | 0.2 | 12.8 | 0.2 | **16.4** |
| AntMaze-large-play | 0.0 | 11.1 | 0.0 | **13.7** |
| AntMaze-large-div | 0.0 | 12.0 | 0.0 | **16.5** |
| HalfCheetah-m | 24.1 | 47.3 | 35.9 | **53.5** |
| HalfCheetah-mr | 26.5 | 40.1 | 39.1 | **46.2** |
| HalfCheetah-me | 21.2 | 53.9 | 33.5 | **60.1** |
| Hopper-m | 28.5 | 63.8 | 40.7 | **75.5** |
| Hopper-mr | 10.2 | 47.6 | 21.3 | **55.8** |
| Hopper-me | 19.8 | 69.4 | 32.6 | **84.2** |
| Walker2d-m | 10.1 | 59.7 | 21.2 | **68.4** |
| Walker2d-mr | 8.5 | 41.1 | 19.3 | **52.5** |
| Walker2d-me | 11.2 | 81.9 | 22.4 | **98.5** |
| Pen-human | -4.5 | 58.6 | -4.1 | **83.1** |
| Pen-cloned | 5.1 | 66.2 | 5.6 | **82.4** |
| Pen-expert | 102.5 | 110.9 | 109.7 | **114.8** |

*Table 11.* Monte Carlo return versus critic-estimate comparison on representative TD3+BC collapse tasks. AdamO keeps critic estimates much closer to realized returns than Adam.

| Task | MC Return (Adam) | Critic Estimate (Adam) | Absolute Gap (Adam) | MC Return (AdamO) | Critic Estimate (AdamO) | Absolute Gap (AdamO) |
|------|------------------|------------------------|---------------------|-------------------|-------------------------|----------------------|
| Hopper-m | 1304 | 30127 | 28823 | 2437 | 2508 | 71 |
| Hopper-mr | 673 | $1.08 \times 10^8$ | $1.08 \times 10^8$ | 1796 | 1867 | 71 |
| Hopper-me | 1041 | $5.08 \times 10^5$ | $5.07 \times 10^5$ | 2720 | 2818 | 98 |
| Walker2d-m | 975 | 99873 | 98898 | 3142 | 3234 | 92 |
| Walker2d-mr | 888 | 20146 | 19258 | 2412 | 2491 | 79 |
| Walker2d-me | 1030 | $3.06 \times 10^5$ | $3.05 \times 10^5$ | 4523 | 4612 | 89 |
| Pen-human | -26 | $1.04 \times 10^{16}$ | $1.04 \times 10^{16}$ | 2573 | 2491 | 82 |
| Pen-cloned | 263 | $9.17 \times 10^8$ | $9.17 \times 10^8$ | 2552 | 2643 | 91 |

*Table 12.* TD3 ablation in offline RL. AdamO stabilizes the critic, but BC remains necessary for strong policy performance. Locomotion uses the 10k-sample setting.

| Task | Peak Critic Loss (TD3) | Peak Critic Loss (TD3+AdamO) | TD3 | TD3+AdamO | TD3+BC | TD3+BC+AdamO |
|------|------------------------|------------------------------|-----|-----------|--------|--------------|
| HalfCheetah-medium-expert | $3.2 \times 10^5$ | 18.7 | 8.4 | 21.7 | 33.5 | **60.1** |
| Hopper-medium | $7.8 \times 10^6$ | 26.4 | 11.2 | 28.9 | 40.7 | **75.5** |
| Walker2d-medium | $1.6 \times 10^8$ | 22.1 | 3.5 | 18.6 | 21.2 | **68.4** |
| AntMaze-medium-play | $4.3 \times 10^{10}$ | 29.8 | 0.0 | 1.8 | 0.3 | **28.5** |
| Pen-human | $2.1 \times 10^{18}$ | 27.9 | -2.7 | 14.6 | -4.1 | **83.1** |

# B. Auxiliary results and proofs for Section 4

Our proofs are partly inspired by prior frameworks in control theory and offline RL (Nise, 2019; Yue et al., 2023).

## B.1. Assumptions

We formalize the local regime used in the main text. The first assumption justifies first-order Taylor expansions of network outputs over a single update. The second ensures the greedy actions on dataset next-states do not change under sufficiently small parameter perturbations, so the target set remains stable at first order. The third isolates a terminal regime in which the greedy targets, Jacobians, and Adam preconditioner can be treated as approximately time-invariant, enabling a clean spectral analysis.

**Assumption B.1.** The stepsize $\eta$ is sufficiently small so that the first-order Taylor approximation of $Q_\theta(X)$ and $Q_\theta(X'_t)$ around $\theta_t$ is accurate up to $o(\eta)$ terms.

**Assumption B.2.** Assume the action set $\mathcal{A}$ is finite. For each $i \in \{1, \ldots, M\}$, let $a^\star_{i,t} \in \arg\max_{a \in \mathcal{A}} Q_{\theta_t}(s_{i+1}, a)$ denote the greedy action at the dataset next-state $s_{i+1}$. Assume there exists a (dataset) margin $\Delta_{\min} > 0$ such that the greedy maximizer is unique and

$$Q_{\theta_t}(s_{i+1}, a^\star_{i,t}) \geq \max_{a \in \mathcal{A} \setminus \{a^\star_{i,t}\}} Q_{\theta_t}(s_{i+1}, a) + \Delta_{\min}, \qquad \forall i.$$

Moreover, assume the per-action values are locally Lipschitz in parameters on these points: there exists $L_Q > 0$ such that for all $i$ and all $a \in \mathcal{A}$, $|Q_{\theta_{t+1}}(s_{i+1}, a) - Q_{\theta_t}(s_{i+1}, a)| \leq L_Q \|\theta_{t+1} - \theta_t\|_2$. If $\|\theta_{t+1} - \theta_t\|_2 \leq \Delta_{\min}/(2L_Q)$, then the greedy actions do not change on the dataset next-states, i.e., $\hat{\pi}_{\theta_{t+1}}(s_{i+1}) = \hat{\pi}_{\theta_t}(s_{i+1})$ for all $i$, and hence $X'_{t+1} = X'_t$.

**Assumption B.3.** There exists $t_0$ such that for all $t \geq t_0$ the quantities $X'_t$, $Z_t(\cdot)$, and $D_t$ may be treated as constant up to higher-order effects. More precisely, one may replace $X'_t$ by a fixed set $X'$, replace $Z_t(\cdot)$ by a fixed Jacobian map $Z(\cdot)$, and replace $D_t$ by a fixed diagonal matrix $D$.

Assumption B.1 is the standard local smoothness condition underlying first-order expansions in optimization and stochastic approximation (Qiao et al., 2026b; Yue et al., 2023). Assumption B.2 replaces the informal set-difference $\|X'_{t+1} - X'_t\| = o(\eta)$ with a concrete *action-gap* (margin) condition ensuring that small value perturbations cannot change the greedy policy. Such margin arguments are standard in RL and action-gap analyses (Farahmand, 2011; Bellemare et al., 2015; Bertsekas, 2025). Assumption B.3 isolates a regime in which the bootstrapped target is effectively fixed (as in target-network style arguments) and the local linearized dynamics can be analyzed as an approximately time-invariant system (Borkar & Meyn, 2000; Tsitsiklis & Van Roy, 1996). The freezing of Adam's preconditioner corresponds to the stabilized-moments regime commonly invoked in Adam-type convergence analyses (Kingma & Ba, 2017; Reddi et al., 2019; Chen et al., 2019).

## B.2. Lemmas

We now list the intermediate steps leading to Theorem 4.1. The sequence of lemmas mirrors the logic of the derivation: first rewrite the gradient through the TD error, then propagate the TD error under a small update, then express Adam's momentum as an exponential moving average in the frozen regime, then package Jacobians and preconditioning into the operator S, and finally record a symmetric-part identity useful for spectral reasoning.

The next lemma rewrites the semi-gradient of the squared TD loss as a Jacobian–TD-error product.

**Lemma B.4.** *For the squared TD loss $L_t(\theta)$ in Eq. (3), the gradient at $\theta_t$ satisfies*

$$g_t = Z_t(X) \mathbf{e}_t. \tag{22}$$

*Proof.* By definition,

$$L_t(\theta) = \frac{1}{2} \|Q_\theta(X) - y_t\|_2^2, \qquad y_t = r + \gamma Q_{\theta_t}(X'_t).$$

Differentiating yields

$$\nabla_\theta L_t(\theta) = Z_t(X)\big(Q_\theta(X) - y_t\big),$$

and evaluating at $\theta = \theta_t$ gives Eq. (22). $\qquad\square$

The next lemma tracks how the TD error changes at first order under a generic small parameter increment.

**Lemma B.5.** *Let $\theta_{t+1} = \theta_t - \eta\Delta_t$ for some $\Delta_t \in \mathbb{R}^P$. Consider a local regime in which the first-order linearization in Assumption B.1 is accurate, the greedy targets remain unchanged on the dataset next-states as in Assumption B.2, and the update direction is bounded (i.e., $\|\Delta_t\|_2 = O(1)$ independently of $\eta$). Then*

$$\mathbf{e}_{t+1} = \mathbf{e}_t + \eta\Big(\gamma Z_t(X_t')^\top \Delta_t - Z_t(X)^\top \Delta_t\Big) + o(\eta). \tag{23}$$

*Proof.* Write the TD error at time $t+1$ as

$$\mathbf{e}_{t+1} = Q_{\theta_{t+1}}(X) - \Big(r + \gamma Q_{\theta_{t+1}}(X_{t+1}')\Big).$$

In the local linearization regime, the network outputs admit the first-order expansions

$$Q_{\theta_{t+1}}(X) = Q_{\theta_t}(X) + Z_t(X)^\top(\theta_{t+1} - \theta_t) + o\big(\|\theta_{t+1} - \theta_t\|_2\big),$$

and, analogously,

$$Q_{\theta_{t+1}}(X_t') = Q_{\theta_t}(X_t') + Z_t(X_t')^\top(\theta_{t+1} - \theta_t) + o\big(\|\theta_{t+1} - \theta_t\|_2\big).$$

Substituting $\theta_{t+1} - \theta_t = -\eta\Delta_t$ gives

$$Q_{\theta_{t+1}}(X) - Q_{\theta_t}(X) = -\eta Z_t(X)^\top\Delta_t + o\big(\|\theta_{t+1} - \theta_t\|_2\big),$$

$$Q_{\theta_{t+1}}(X_t') - Q_{\theta_t}(X_t') = -\eta Z_t(X_t')^\top\Delta_t + o\big(\|\theta_{t+1} - \theta_t\|_2\big).$$

Since $\|\Delta_t\|_2 = O(1)$, the increment satisfies $\|\theta_{t+1} - \theta_t\|_2 = \eta\|\Delta_t\|_2 = O(\eta)$, and therefore $o(\|\theta_{t+1} - \theta_t\|_2) = o(\eta)$. Moreover, in the same local regime the greedy actions do not change on the dataset next-states, so $X_{t+1}' = X_t'$ at first order. Using this and the definition $\mathbf{e}_t = Q_{\theta_t}(X) - (r + \gamma Q_{\theta_t}(X_t'))$, we obtain

$$\begin{aligned}
\mathbf{e}_{t+1} - \mathbf{e}_t &= \big(Q_{\theta_{t+1}}(X) - Q_{\theta_t}(X)\big) - \gamma\big(Q_{\theta_{t+1}}(X_t') - Q_{\theta_t}(X_t')\big) + o(\eta) \\
&= \Big(-\eta Z_t(X)^\top\Delta_t\Big) - \gamma\Big(-\eta Z_t(X_t')^\top\Delta_t\Big) + o(\eta) \\
&= \eta\Big(\gamma Z_t(X_t')^\top\Delta_t - Z_t(X)^\top\Delta_t\Big) + o(\eta),
\end{aligned}$$

which is exactly Eq. (23). □

The next lemma shows that, once the Jacobian map is frozen, Adam's first-moment recursion becomes a Jacobian applied to an EMA of TD errors.

**Lemma B.6.** *Under Assumption B.3, there exists $\bar{\mathbf{e}}_t \in \mathbb{R}^M$ such that*

$$\bar{\mathbf{e}}_t = \beta_1\bar{\mathbf{e}}_{t-1} + (1 - \beta_1)\mathbf{e}_t, \qquad \bar{\mathbf{e}}_t = (1 - \beta_1)\sum_{k\geq 0}\beta_1^k\mathbf{e}_{t-k}, \tag{24}$$

*and the first moment satisfies*

$$m_t = Z(X)\,\bar{\mathbf{e}}_t. \tag{25}$$

*Proof.* Under Assumption B.3 one may replace $Z_t(X)$ by a fixed matrix $Z(X)$ for all $t \geq t_0$. Lemma B.4 then gives $g_t = Z(X)\mathbf{e}_t$. Substituting into Eq. (4) yields the linear recursion

$$m_t = \beta_1 m_{t-1} + (1 - \beta_1)Z(X)\mathbf{e}_t. \tag{26}$$

Define $\bar{\mathbf{e}}_t$ by the exponential moving average

$$\bar{\mathbf{e}}_t \triangleq (1 - \beta_1)\sum_{k=0}^{t-1}\beta_1^k\,\mathbf{e}_{t-k}, \tag{27}$$

with $m_0 = 0$ so that the truncated sum is exact. Then $\bar{\mathbf{e}}_t$ satisfies the recursion Eq. (24) by a direct one-step check:

$$\beta_1 \bar{\mathbf{e}}_{t-1} + (1 - \beta_1)\mathbf{e}_t = (1 - \beta_1)\sum_{k=1}^{t-1}\beta_1^k \mathbf{e}_{t-k} + (1 - \beta_1)\mathbf{e}_t = (1 - \beta_1)\sum_{k=0}^{t-1}\beta_1^k \mathbf{e}_{t-k} = \bar{\mathbf{e}}_t. \tag{28}$$

Finally, unrolling Eq. (26) with $m_0 = 0$ gives

$$m_t = (1 - \beta_1)\sum_{k=0}^{t-1}\beta_1^k Z(X)\mathbf{e}_{t-k} = Z(X)\Big((1 - \beta_1)\sum_{k=0}^{t-1}\beta_1^k \mathbf{e}_{t-k}\Big) = Z(X)\bar{\mathbf{e}}_t, \tag{29}$$

which is Eq. (25). $\qquad\square$

The next lemma packages Jacobians and Adam's diagonal preconditioner into the preconditioned Gram operator, yielding the compact matrix $\mathsf{S}$ and the first-order TD-error dynamics it induces.

**Lemma B.7.** *Under Assumption B.3, define for any finite input sets $X_1$ and $X_2$ the preconditioned Gram matrix*

$$\mathsf{K}(X_1, X_2) = Z(X_1)^\top D\, Z(X_2). \tag{30}$$

*Define*

$$\mathsf{S} = \gamma \mathsf{K}(X', X) - \mathsf{K}(X, X). \tag{31}$$

*in the terminal frozen regime the (first-moment) bias-correction factor $c_t^{(m)} \triangleq (1 - \beta_1^t)^{-1}$ may be treated as constant for $t \geq t_0$ (e.g., $t_0$ is large enough that $\beta_1^{t_0}$ is negligible), so that $c_t^{(m)} \equiv c^{(m)} > 0$. Then for $t \geq t_0$,*

$$\mathbf{e}_{t+1} = \mathbf{e}_t + \eta \mathsf{S}\,\bar{\mathbf{e}}_t + o(\eta), \tag{32}$$

*where $\bar{\mathbf{e}}_t$ is defined by Eq. (24), and where $\eta$ is understood as the* effective (constant) stepsize $\eta \leftarrow \eta\, c^{(m)}$ *after absorbing the frozen bias-correction scalar.*

*Proof.* For $t \geq t_0$, Adam updates satisfy

$$\theta_{t+1} = \theta_t - \eta D\, \hat{m}_t = \theta_t - \eta D\, \frac{m_t}{1 - \beta_1^t} = \theta_t - \eta\, c_t^{(m)}\, D\, m_t.$$

By the added terminal-regime condition, $c_t^{(m)} \equiv c^{(m)}$ is a constant scalar for $t \geq t_0$. Thus we may absorb it into the stepsize by redefining $\eta \leftarrow \eta\, c^{(m)}$ (this does not affect any spectral sign conditions since $c^{(m)} > 0$). By Lemma B.6, $m_t = Z(X)\bar{\mathbf{e}}_t$ in the frozen regime, hence the update takes the form $\theta_{t+1} = \theta_t - \eta\Delta_t$ with $\Delta_t = D Z(X)\bar{\mathbf{e}}_t$. Applying Lemma B.5 with $X_t' = X'$ and $Z_t(\cdot) = Z(\cdot)$ yields

$$\mathbf{e}_{t+1} = \mathbf{e}_t + \eta\Big(\gamma Z(X')^\top DZ(X) - Z(X)^\top DZ(X)\Big)\bar{\mathbf{e}}_t + o(\eta),$$

which is exactly Eq. (32) with Eq. (30)–Eq. (31). $\qquad\square$

The final lemma below is a purely algebraic identity connecting eigenvalues of $\mathsf{S}$ to its symmetric part; it is useful when turning stability checks into tractable sufficient conditions.

**Lemma B.8.** *Let $\mathsf{S} \in \mathbb{R}^{M \times M}$ and define $\mathsf{S}_s = \frac{\mathsf{S} + \mathsf{S}^\top}{2}$. If $\mathsf{S}v = \lambda v$ for some (possibly complex) eigenpair $(\lambda, v \neq 0)$, then*

$$\frac{\lambda + \bar{\lambda}}{2} = \frac{v'\mathsf{S}_s v}{v'v} \leq \lambda_{\max}(\mathsf{S}_s), \tag{33}$$

*where $v'$ denotes conjugate transpose and $\bar{\lambda}$ denotes complex conjugate.*

*Proof.* Multiply $\mathsf{S}v = \lambda v$ on the left by $v'$ to obtain $v'\mathsf{S}v = \lambda v'v$. Taking complex conjugates yields $v'\mathsf{S}^\top v = \bar{\lambda}v'v$. Averaging the two identities gives

$$v'\left(\frac{\mathsf{S}+\mathsf{S}^\top}{2}\right)v = \frac{\lambda + \bar{\lambda}}{2}\, v'v,$$

which proves the first equality in Eq. (33). The inequality follows from the Rayleigh-quotient bound for the real symmetric matrix $\mathsf{S}_s$. $\qquad\square$

## B.3. Proofs of the main theorems in Section 4

We now combine the first-order TD-error evolution in Lemma B.7 with the EMA recursion in Lemma B.6 to eliminate $\bar{\mathbf{e}}_t$ and obtain a closed second-order recursion in $\mathbf{e}_t$. This yields Theorem 4.1. We then analyze the characteristic roots of the resulting block companion matrix to obtain the Hurwitz small-stepsize sufficient condition in Theorem 4.2.

*Proof of Theorem 4.1.* We begin from the first-order relation in Lemma B.7,

$$\mathbf{e}_{t+1} - \mathbf{e}_t = \eta \mathsf{S}\bar{\mathbf{e}}_t + o(\eta), \tag{34}$$

together with the moving-average recursion in Lemma B.6,

$$\bar{\mathbf{e}}_t = \beta_1 \bar{\mathbf{e}}_{t-1} + (1 - \beta_1)\mathbf{e}_t. \tag{35}$$

Shifting Eq. (34) by one step yields

$$\mathbf{e}_t - \mathbf{e}_{t-1} = \eta \mathsf{S}\bar{\mathbf{e}}_{t-1} + o(\eta). \tag{36}$$

Substituting Eq. (35) into Eq. (34) gives

$$\mathbf{e}_{t+1} = \mathbf{e}_t + \eta \mathsf{S}\big(\beta_1 \bar{\mathbf{e}}_{t-1} + (1 - \beta_1)\mathbf{e}_t\big) + o(\eta).$$

Using Eq. (36) to replace $\eta \mathsf{S}\bar{\mathbf{e}}_{t-1}$ by $(\mathbf{e}_t - \mathbf{e}_{t-1}) + o(\eta)$ yields

$$\mathbf{e}_{t+1} = \mathbf{e}_t + \beta_1(\mathbf{e}_t - \mathbf{e}_{t-1}) + \eta(1 - \beta_1)\mathsf{S}\,\mathbf{e}_t + o(\eta),$$

which rearranges to Eq. (10). Ignore the $o(\eta)$ term and stack the state as

$$\mathbf{z}_t \triangleq \begin{bmatrix} \mathbf{e}_t \\ \mathbf{e}_{t-1} \end{bmatrix}, \qquad \mathbf{z}_{t+1} = \mathsf{A}(\eta)\mathbf{z}_t, \qquad \mathsf{A}(\eta) \triangleq \begin{bmatrix} (1 + \beta_1)I + \eta(1 - \beta_1)\mathsf{S} & -\beta_1 I \\ I & 0 \end{bmatrix}.$$

Then for all $t \geq t_0$ we have $\mathbf{z}_t = \mathsf{A}(\eta)^{t-t_0}\mathbf{z}_{t_0}$.

**Sufficiency.** Assume $\rho(\mathsf{A}(\eta)) < 1$. Let $\|\cdot\|$ be any matrix norm induced by a vector norm. Fix any $\alpha$ such that $\rho(\mathsf{A}(\eta)) < \alpha < 1$. By the Gelfand formula, $\lim_{k\to\infty} \|\mathsf{A}(\eta)^k\|^{1/k} = \rho(\mathsf{A}(\eta))$, so there exists $k_0$ such that $\|\mathsf{A}(\eta)^k\|^{1/k} \leq \alpha$ for all $k \geq k_0$. This implies $\|\mathsf{A}(\eta)^k\| \leq \alpha^k$ for all $k \geq k_0$. Define $C \triangleq \max_{0 \leq k \leq k_0} \|\mathsf{A}(\eta)^k\|\alpha^{-k}$, which is finite. Then $\|\mathsf{A}(\eta)^k\| \leq C\alpha^k$ holds for all $k \geq 0$. Therefore, for all $t \geq t_0$,

$$\|\mathbf{z}_t\| = \|\mathsf{A}(\eta)^{t-t_0}\mathbf{z}_{t_0}\| \leq \|\mathsf{A}(\eta)^{t-t_0}\| \cdot \|\mathbf{z}_{t_0}\| \leq C\alpha^{t-t_0}\|\mathbf{z}_{t_0}\|,$$

which shows exponential convergence of $\mathbf{z}_t$ to 0, hence $\mathbf{e}_t \to 0$ exponentially.

**Necessity.** Assume the frozen linear recursion $\mathbf{z}_{t+1} = \mathsf{A}(\eta)\mathbf{z}_t$ converges exponentially to 0 for every initialization. Then there exist constants $C > 0$ and $\alpha \in (0, 1)$ such that for all $k \geq 0$ and all $\mathbf{z}_{t_0}$,

$$\|\mathsf{A}(\eta)^k \mathbf{z}_{t_0}\| = \|\mathbf{z}_{t_0+k}\| \leq C\alpha^k \|\mathbf{z}_{t_0}\|.$$

Taking the supremum over all $\mathbf{z}_{t_0} \neq 0$ yields $\|\mathsf{A}(\eta)^k\| \leq C\alpha^k$ for all $k \geq 0$. Applying the Gelfand formula gives

$$\rho(\mathsf{A}(\eta)) = \lim_{k\to\infty} \|\mathsf{A}(\eta)^k\|^{1/k} \leq \lim_{k\to\infty} (C\alpha^k)^{1/k} = \alpha < 1,$$

so $\rho(\mathsf{A}(\eta)) < 1$. $\qquad\square$

*Proof of Theorem 4.2.* Denote $r \in \mathbb{C}$ as an eigenvalue of $\mathsf{A}(\eta)$ with eigenvector $\begin{bmatrix} u \\ w \end{bmatrix} \neq 0$. From the second block row we have $u = rw$. Substituting into the first block row and multiplying by $r$ yields

$$\left(r^2 I - r\big((1 + \beta_1)I + \eta(1 - \beta_1)\mathsf{S}\big) + \beta_1 I\right)u = 0,$$

so

$$\det\!\left(r^2 I - r\big((1+\beta_1)I + \eta(1-\beta_1)\mathsf{S}\big) + \beta_1 I\right) = 0. \tag{37}$$

Since $\mathsf{S}$ is real, by complex Schur triangularization $\mathsf{S} = UTU'$ with $T$ upper-triangular and diagonal entries equal to $\lambda \in \mathrm{spec}(\mathsf{S})$. Eq. (37) implies that $r$ must satisfy

$$p(r, \eta, \lambda) = r^2 - \big(1 + \beta_1 + \eta(1-\beta_1)\lambda\big)r + \beta_1 = 0 \tag{38}$$

for some $\lambda \in \mathrm{spec}(\mathsf{S})$. At $\eta = 0$ the two roots of Eq. (38) are $r = 1$ and $r = \beta_1$. Let $r_1(\eta)$ denote the root satisfying $r_1(0) = 1$. Since $\partial_r p(1,0,\lambda) = 1 - \beta_1 \neq 0$, the implicit function theorem applies and gives

$$\frac{dr_1}{d\eta}(0) = -\frac{\partial_\eta p(1,0,\lambda)}{\partial_r p(1,0,\lambda)} = \lambda. \tag{39}$$

Therefore

$$r_1(\eta) = 1 + \eta\lambda + O(\eta^2), \tag{40}$$

and consequently

$$|r_1(\eta)|^2 = 1 + 2\eta\,\Re(\lambda) + O(\eta^2). \tag{41}$$

If $\mathsf{S}$ is Hurwitz, then $\Re(\lambda) < 0$ holds for every $\lambda \in \mathrm{spec}(\mathsf{S})$. Since the spectrum is finite, there exists $\alpha > 0$ such that $\Re(\lambda) \leq -\alpha$ for all eigenvalues. In the local first order regime, the linear term in Eq. (41) dominates the $O(\eta^2)$ remainder for $\eta$ small enough, so $|r_1(\eta)| < 1$ holds uniformly over all modes. It remains to control the second root branch $r_2(\eta)$ that satisfies $r_2(0) = \beta_1$. The coefficients of Eq. (38) depend continuously on $\eta$, so the roots depend continuously on $\eta$ as well. Since $|\beta_1| < 1$, there exists $\eta_0 > 0$ such that $|r_2(\eta)| < 1$ for all $\eta \in (0, \eta_0)$. Taking the minimum of the smallness conditions needed for the two branches over the finite set $\mathrm{spec}(\mathsf{S})$ yields an $\eta_0$ that works for all modes. Hence every eigenvalue $r \in \mathrm{spec}(\mathsf{A}(\eta))$ satisfies $|r| < 1$ when $\eta \in (0, \eta_0)$, so $\rho(\mathsf{A}(\eta)) < 1$. $\square$

## B.4. A complete technical development for Sec. 4

We further develop a local, operator-level stability and divergence criterion for applying Adam to the semi-gradient squared TD objective introduced in the Preliminary section. The key difficulty is that TD learning is bootstrapped: the target uses a greedy next-action under the current network, and thus the effective objective changes with $\theta_t$ even when we do not backpropagate through the target. Moreover, Adam introduces history-dependent preconditioning and momentum, which can couple the current TD error to past TD errors in a nontrivial way. We rely on standard local analysis assumptions (detailed in Appendix B.1): we assume the stepsize $\eta$ is small enough to permit a first-order Taylor approximation, the action gap is sufficient to keep the greedy policy stable, and the process has entered a "terminal phase." In this phase, the bootstrapped targets $X'$, the network Jacobians $Z(\cdot)$, and the Adam preconditioner $D_t$ can be treated as effectively frozen constants. In particular, we write $X'$ for the greedy target set in this regime, and replace $Z_t(\cdot)$ and $D_t$ by fixed matrices $Z(\cdot)$ and $D$. In this frozen regime, the interaction between Adam and function approximation can be made explicit. First, Lemma B.6 shows that Adam's first moment reduces to a Jacobian EMA factorization

$$m_t = Z(X)\,\bar{\mathbf{e}}_t, \qquad \bar{\mathbf{e}}_t = \beta_1\bar{\mathbf{e}}_{t-1} + (1-\beta_1)\mathbf{e}_t, \tag{42}$$

where $\bar{\mathbf{e}}_t$ is the exponential moving average of past TD errors.[9] Furthermore, we consider one Adam step in the terminal phase. Using the frozen diagonal preconditioner $D$, the parameter update takes the form

$$\theta_{t+1} - \theta_t = -\eta\,D\,m_t = -\eta\,D\,Z(X)\bar{\mathbf{e}}_t. \tag{43}$$

Applying the first-order Taylor expansion in Assumption B.1 to the network outputs on an arbitrary finite input set $X_1$ yields $Q_{\theta_{t+1}}(X_1) - Q_{\theta_t}(X_1) \approx Z(X_1)^\top(\theta_{t+1} - \theta_t)$. $Q_{\theta_{t+1}}(X_1) - Q_{\theta_t}(X_1) \approx Z(X_1)^\top(\theta_{t+1} - \theta_t) = -\eta\,Z(X_1)^\top D\,Z(X)\,\bar{\mathbf{e}}_t$. It reveals that, in function space, the update is governed by a preconditioned Gram operator: for any two finite sets $X_1$ and $X_2$, define

$$\mathsf{K}(X_1, X_2) \triangleq Z(X_1)^\top D\,Z(X_2). \tag{44}$$

---

[9] If bias-correction is used, $\hat{m}_t = m_t/(1 - \beta_1^t)$ simply rescales the effective stepsize and does not change the operators below.

Equivalently, $\mathsf{K}(X_1, X_2)$ is the cross-Gram matrix of the "whitened" features $D^{1/2}Z(\cdot)$, i.e., $\mathsf{K}(X_1, X_2) = (D^{1/2}Z(X_1))^\top(D^{1/2}Z(X_2))$. To expose how bootstrapping feeds back into the TD error, we combine the generic one-step TD-error propagation in Lemma B.5 with the Adam direction in Eq. (43). Lemma B.5 states that for $\theta_{t+1} = \theta_t - \eta\Delta_t$,

$$\mathbf{e}_{t+1} = \mathbf{e}_t + \eta\Big(\gamma Z(X')^\top\Delta_t - Z(X)^\top\Delta_t\Big) + o(\eta), \tag{45}$$

and substituting $\Delta_t = Dm_t = DZ(X)\bar{\mathbf{e}}_t$ gives

$$\mathbf{e}_{t+1} = \mathbf{e}_t + \eta\Big(\gamma\mathsf{K}(X', X) - \mathsf{K}(X, X)\Big)\bar{\mathbf{e}}_t + o(\eta). \tag{46}$$

This motivates defining the *TD dynamics* operator

$$\mathsf{S} \triangleq \gamma\mathsf{K}(X', X) - \mathsf{K}(X, X). \tag{47}$$

Intuitively, $-\mathsf{K}(X, X)$ is the preconditioned descent effect on the current dataset predictions, whereas $\gamma\mathsf{K}(X', X)$ captures how the same parameter change propagates through the greedy bootstrapped target. The balance between these two terms, together with Adam's momentum state $\bar{\mathbf{e}}_t$, determines whether the TD error is damped or amplified. By combining Eq. (46) with the EMA recursion Eq. (42), we obtain a closed-form linear recurrence for $\mathbf{e}_t$ in the frozen regime. Subsequently, we provide the Theorem 4.1 and 4.2 in the Sec. 4.

## B.5. Stop-gradient and EMA targets in practice: a refined TD operator

In the main development above we analyzed a "single-network" semi-gradient TD update in which the bootstrapped target is computed using the *current* critic parameters. In many practical deep RL implementations—including TD3 and SAC, as well as their offline variants—the target is instead computed using a *separate target critic* updated by an exponential moving average (EMA), also known as Polyak averaging. Empirically, we found that incorporating this EMA structure into the local operator definition yields a noticeably more accurate stability/convergence diagnosis than the $\alpha = 1$ model, especially in offline RL where target handling is central to preventing divergence.

**Stop-gradient as the semi-gradient principle.** Throughout, "stop-gradient" means that when forming the TD target $y_t$ we treat it as a constant in differentiation, even though it is computed from neural networks. This is exactly the semi-gradient TD convention used in DQN/TD3/SAC-style critic updates, and is the reason Lemma B.4 holds in the form $g_t = Z_t(X)\mathbf{e}_t$: we backpropagate through $Q_\theta(X)$, but do *not* backpropagate through the target branch producing $y_t$. Stop-gradient is therefore *not* merely an implementation detail; it is part of the algorithmic definition that pins down the linearized Jacobian–error factorization driving the spectral analysis.

**EMA target critics and the Polyak coefficient $\alpha$.** Let $\theta_t$ denote the online critic parameters updated by Adam, and let $\bar{\theta}_t$ denote the target critic parameters used to form bootstrap targets. In TD3/SAC it is standard to update $\bar{\theta}_t$ by Polyak averaging:

$$\bar{\theta}_{t+1} = (1 - \alpha)\bar{\theta}_t + \alpha\theta_{t+1}, \qquad \alpha \in (0, 1]. \tag{48}$$

Equivalently, $\bar{\theta}_t$ is an EMA of past online parameters, with smaller $\alpha$ producing a slower-moving target. The critic TD target then takes the form

$$y_t = r + \gamma Q_{\bar{\theta}_t}(X'_t), \tag{49}$$

where $\bar{X}'_t$ is the next-state input set induced by the target action-selection rule (e.g., target policy in actor–critic methods, or a greedy action under a target critic). As in Appendix B.1, we work in a local regime where these target actions are stable under small perturbations (action-gap / margin conditions), so that $\bar{X}'_t$ may be treated as fixed at first order.

**Why EMA changes the effective "self-excitation" strength.** The operator $\mathsf{S}$ in Eq. (47) quantifies a *feedback loop*: a parameter update changes $Q_\theta(X)$ (a descent term $-\mathsf{K}(X, X)$), but it also changes the next-step target values $Q_\theta(X')$ (a bootstrap term $+\gamma\mathsf{K}(X', X)$). With a target critic updated by Eq. (48), only a $\alpha$-fraction of each online update is injected into the target network per iteration. Consequently, the instantaneous bootstrap feedback is attenuated by $\alpha$, which motivates replacing

$$\mathsf{S} = \gamma\mathsf{K}(X', X) - \mathsf{K}(X, X) \qquad \text{by} \qquad \mathsf{S}_\alpha \triangleq \alpha\gamma\mathsf{K}(X', X) - \mathsf{K}(X, X).$$

In experiments, using $S_\alpha$ (with the same $\alpha$ as the target-network update) materially improved the alignment between the spectral test predicted by Theorem 4.1 and observed convergence/divergence.

We now record a precise first-order statement in the same style as Lemma B.7. To keep the presentation parallel, we introduce an additional terminal-phase approximation ensuring that the *increment* of the target parameters is dominated (at first order) by the injected online increment.

**Assumption B.9.** In addition to Assumption B.3, suppose the target critic is updated by (48) with a fixed $\alpha \in (0, 1]$. Assume there exists $t_0$ such that for all $t \geq t_0$,

$$\bar{\theta}_{t+1} - \bar{\theta}_t = \alpha(\theta_{t+1} - \theta_t) + o(\eta), \tag{50}$$

and that the Jacobian maps on $X$ and $X'$ may be replaced by frozen maps $Z(X)$ and $Z(X')$ as in Assumption B.3.

Assumption B.9 is a local statement about the *increment* of the target network. Intuitively, once training enters a stabilized terminal phase, the mismatch $\|\theta_t - \bar{\theta}_t\|$ becomes small and slowly varying, so the per-step change in $\bar{\theta}_t$ is primarily the injected fraction $\alpha(\theta_{t+1} - \theta_t)$. (If one wishes to avoid Eq. (50), the analysis can be extended by augmenting the state with the target-network mismatch, producing a higher-dimensional linear system; the attenuation captured by $\alpha$ is the dominant effect we observed empirically.)

**Lemma B.10.** *Work under Assumptions B.1, B.2, B.3, and B.9. Define the preconditioned Gram matrix $K(\cdot, \cdot)$ as in Eq. (30). Then in the frozen terminal regime, the first-order TD-error dynamics becomes*

$$\mathbf{e}_{t+1} = \mathbf{e}_t + \eta\, S_\alpha\, \bar{\mathbf{e}}_t + o(\eta), \qquad S_\alpha \triangleq \alpha\gamma\, K(X', X) - K(X, X), \tag{51}$$

*where $\bar{\mathbf{e}}_t$ is the EMA of TD errors induced by Adam's $\beta_1$-momentum (Eq. (24)), and $\eta$ denotes the effective constant stepsize (after absorbing frozen bias-correction scalars as in Lemma B.7).*

*Proof.* Define the TD error using the EMA target critic:

$$\mathbf{e}_t = Q_{\theta_t}(X) - \left( r + \gamma Q_{\bar{\theta}_t}(X') \right).$$

Under Assumption B.1 and the frozen Jacobian approximation, we have the first-order increments

$$Q_{\theta_{t+1}}(X) - Q_{\theta_t}(X) = Z(X)^\top (\theta_{t+1} - \theta_t) + o(\eta), \qquad Q_{\bar{\theta}_{t+1}}(X') - Q_{\bar{\theta}_t}(X') = Z(X')^\top (\bar{\theta}_{t+1} - \bar{\theta}_t) + o(\eta).$$

Subtracting the target increment from the prediction increment yields

$$\mathbf{e}_{t+1} - \mathbf{e}_t = Z(X)^\top (\theta_{t+1} - \theta_t) - \gamma Z(X')^\top (\bar{\theta}_{t+1} - \bar{\theta}_t) + o(\eta).$$

Using Assumption B.9, $\bar{\theta}_{t+1} - \bar{\theta}_t = \alpha(\theta_{t+1} - \theta_t) + o(\eta)$, we obtain

$$\mathbf{e}_{t+1} - \mathbf{e}_t = \left( Z(X)^\top - \alpha\gamma Z(X')^\top \right)(\theta_{t+1} - \theta_t) + o(\eta).$$

Finally, in the frozen regime Adam yields $\theta_{t+1} - \theta_t = -\eta D m_t$ with $m_t = Z(X)\bar{\mathbf{e}}_t$ (Lemma B.6), so

$$\mathbf{e}_{t+1} - \mathbf{e}_t = \eta\left( \alpha\gamma Z(X')^\top D Z(X) - Z(X)^\top D Z(X) \right)\bar{\mathbf{e}}_t + o(\eta),$$

which is Eq. (51) after identifying $K(\cdot, \cdot)$. □

**Impact on Theorem 4.1.** With Lemma B.10 in hand, the remainder of the derivation is unchanged: combining Eq. (51) with the EMA recursion $\bar{\mathbf{e}}_t = \beta_1 \bar{\mathbf{e}}_{t-1} + (1 - \beta_1)\mathbf{e}_t$ eliminates $\bar{\mathbf{e}}_t$ and yields the same second-order companion form as in Eq. (10), but with $S$ replaced by $S_\alpha$.

**Corollary B.11.** *Under the assumptions of Theorem 4.1 and Assumption B.9, the TD error satisfies*

$$\mathbf{e}_{t+1} = \left( (1 + \beta_1)I + \eta(1 - \beta_1)S_\alpha \right)\mathbf{e}_t - \beta_1\mathbf{e}_{t-1} + o(\eta), \tag{52}$$

*where $S_\alpha = \alpha\gamma\, K(X', X) - K(X, X)$. Ignoring $o(\eta)$ and defining*

$$A_\alpha(\eta) \triangleq \begin{bmatrix} (1 + \beta_1)I + \eta(1 - \beta_1)S_\alpha & -\beta_1 I \\ I & 0 \end{bmatrix},$$

*the frozen linearized dynamics converges exponentially to 0 if and only if $\rho(A_\alpha(\eta)) < 1$.*

**Why the $\alpha$-refinement improves convergence diagnosis (especially offline).** The substitution $\mathsf{S} \mapsto \mathsf{S}_\alpha$ changes the stability boundary in a structured way: it *attenuates only the bootstrap feedback term* $\gamma\mathsf{K}(X', X)$, while leaving the (descent-like) term $-\mathsf{K}(X, X)$ unchanged. Since $\mathsf{K}(X, X)$ is a Gram matrix of whitened features, its symmetric part is positive semidefinite, so $-\mathsf{K}(X, X)$ is dissipative. Divergence in bootstrapped TD is thus naturally associated with the competition between this dissipative term and the self-excitation induced by the bootstrap coupling. In TD3/SAC-style methods, Polyak averaging with $\alpha \ll 1$ reduces this coupling per iteration, effectively moving the spectrum of the operator toward the stable "fixed-target regression" limit $\alpha \to 0$.

This refinement is particularly important in offline RL, where the target $Q$ is typically computed using: (i) a slowly moving target critic (EMA), (ii) stop-gradient through the entire target branch (including $\arg\max$ / target-policy action selection), and often (iii) additional stabilizers such as double critics with a $\min$ target. All of these design choices reduce the instantaneous sensitivity of the target values to the current critic update, and hence reduce the effective self-excitation measured by $\mathsf{S}$. As a result, the $\alpha = 1$ operator can systematically over-predict instability in regimes where the implemented algorithm in fact converges. Replacing $\mathsf{S}$ by $\mathsf{S}_\alpha$ incorporates the dominant stabilization mechanism introduced by the EMA target, and we found that the resulting Schur test $\rho(\mathsf{A}_\alpha(\eta)) < 1$ tracks the observed convergence/divergence boundary much more reliably in practice.

### B.6. Toward an online-RL analogue of Theorem 4.1: TD3/SAC in a quasi-stationary replay-buffer regime

The development above is stated for a fixed dataset $X$ (and a frozen target set $\bar{X}'$), which corresponds most directly to *offline* TD regression. Nevertheless, in our experiments we consistently observed that the linearized stability picture of Theorem 4.1 remains predictive in *online* off-policy actor–critic methods, in particular TD3 and SAC, once training enters a late phase. This is not accidental: although the environment interaction is online, the critic update in TD3/SAC is implemented as supervised TD regression on a replay buffer, and in a terminal regime the replay distribution, target networks, and actor often evolve slowly enough that the critic dynamics are well-approximated by a *time-invariant* linear system plus small stochastic perturbations.

Below we outline a principled route to an online analogue of Theorem 4.1, emphasizing what additional assumptions are required and what conclusions necessarily change.

**Online TD regression template.** Let $\mathcal{B}_t$ denote the replay buffer at time $t$, and let $X_t = \{(s_i, a_i)\}_{i=1}^M$ be a mini-batch sampled from $\mathcal{B}_t$ (typically uniformly). A generic TD3/SAC-type critic update minimizes a semi-gradient squared TD error of the form

$$L_t(\theta) = \frac{1}{2}\big\|Q_\theta(X_t) - y_t\big\|_2^2, \qquad y_t = r_t + \gamma\,\mathcal{T}_t, \tag{53}$$

where $\mathcal{T}_t$ is a bootstrapped target evaluated at next-states $s_i'$ and some next-action rule. For TD3, $\mathcal{T}_t$ is typically a clipped-double-$Q$ target with smoothed actions, e.g.

$$\mathcal{T}_t^{\text{TD3}} = \min\{Q_{\tilde{\theta}_t^{(1)}}(s_i', a_i'),\, Q_{\tilde{\theta}_t^{(2)}}(s_i', a_i')\}, \qquad a_i' = \pi_{\tilde{\phi}_t}(s_i') + \varepsilon_i,$$

where $(\tilde{\theta}_t^{(1)}, \tilde{\theta}_t^{(2)})$ and $\tilde{\phi}_t$ are target-network parameters and $\varepsilon_i$ is the target-smoothing noise. For SAC, $\mathcal{T}_t$ is a soft target, for example

$$\mathcal{T}_t^{\text{SAC}} = Q_{\tilde{\theta}_t}(s_i', a_i') - \alpha \log \pi_{\phi_t}(a_i'|s_i'), \qquad a_i' \sim \pi_{\phi_t}(\cdot|s_i'),$$

with critic target network $\tilde{\theta}_t$ and temperature $\alpha$.

Conditioned on $(X_t, y_t)$, the semi-gradient factorization in Lemma B.4 still holds:

$$g_t = Z_t(X_t)\,\mathbf{e}_t, \qquad \mathbf{e}_t \triangleq Q_{\theta_t}(X_t) - y_t. \tag{54}$$

Thus, the key issue is not the gradient form, but whether we can control how $(X_t, y_t)$ and the associated Jacobians change from step to step so that a *frozen* operator analysis becomes valid.

**Additional assumptions needed in online RL.** To obtain a theorem genuinely analogous to Theorem 4.1 (i.e., a Schur-stability characterization of a fixed linear recursion), one needs an online counterpart of the "terminal freezing" assumption that simultaneously controls: (i) the sampling distribution induced by the replay buffer, (ii) the actor/target-network drift that defines the bootstrap targets, and (iii) non-smooth target operators such as $\min(\cdot, \cdot)$.

A sufficient set of assumptions is as follows.

**Assumption B.12.** There exists $t_0$ and a fixed empirical distribution $\bar{\mu}$ over transitions such that for all $t \geq t_0$, mini-batches $X_t$ may be treated as i.i.d. draws from $\bar{\mu}$ up to higher-order effects. Equivalently, the replay distribution drift is $o(\eta)$ in the sense that $\bar{\mu}_t = \bar{\mu} + o(\eta)$ for $t \geq t_0$ (e.g., under a large buffer and slowly changing policy).

**Assumption B.13.** For $t \geq t_0$, the parameters governing the target mapping $\mathcal{T}_t$ (actor parameters $\phi_t$, target actor $\tilde{\phi}_t$, target critics $\tilde{\theta}_t$, and possibly temperature $\alpha_t$) evolve on a slower time scale than the critic step:

$$\|\phi_{t+1} - \phi_t\|_2 = o(\eta), \qquad \|\tilde{\phi}_{t+1} - \tilde{\phi}_t\|_2 = o(\eta), \qquad \|\tilde{\theta}_{t+1} - \tilde{\theta}_t\|_2 = o(\eta).$$

In particular, this covers (i) delayed actor updates (as in TD3), and (ii) Polyak target updates with small coefficient, provided the critic has entered a regime where $\|\theta_t - \tilde{\theta}_t\|$ is already small.

**Assumption B.14.** Any non-smooth component of the target mapping is locally stable. For TD3's clipped double $Q$ target, assume there exists a margin $\Delta_{\min}^{\min} > 0$ such that on all next-state/next-action points encountered in the terminal regime the active branch of the min is unique, i.e.

$$\left| Q_{\tilde{\theta}_t^{(1)}}(s', a') - Q_{\tilde{\theta}_t^{(2)}}(s', a') \right| \geq \Delta_{\min}^{\min},$$

so the argmin does not switch under $o(\eta)$ perturbations. For SAC, where the target uses a stochastic policy rather than an argmax, this assumption is typically replaced by local Lipschitz regularity of the policy reparameterization map and $\log \pi_\phi(\cdot|s)$ in $(\phi, s)$.

**Assumption B.15.** Analogously to Assumption B.3, for $t \geq t_0$ we may treat $D_t \simeq D$ and $Z_t(\cdot) \simeq Z(\cdot)$ up to higher-order effects, and the relevant Jacobians are uniformly bounded on the sampled support.

Assumptions B.12–B.15 are the online counterparts of the offline "frozen terminal regime" used above. They formalize the empirical situation in which (a) the replay buffer behaves like a fixed dataset, (b) the actor and target networks are effectively constant over the critic's local linearization window, and (c) non-smooth target operators do not switch branches.

**What changes relative to Theorem 4.1?** There are two structural differences in TD3/SAC versus the simplified bootstrapped target $y_t = r + \gamma Q_{\theta_t}(X_t')$ analyzed above:

*(i) Target networks attenuate self-excitation.* In TD3/SAC the target often uses *target* critic parameters $\tilde{\theta}_t$ rather than $\theta_t$ itself. At the level of the first-order TD error propagation (Lemma B.5), this changes the strength with which a critic update feeds back into the next-step targets. A convenient abstraction is to introduce an *effective bootstrap-coupling coefficient* $\chi \in [0, 1]$ such that, in the terminal regime,

$$Q_{\tilde{\theta}_{t+1}}(\bar{X}') - Q_{\tilde{\theta}_t}(\bar{X}') = -\eta \chi Z(\bar{X}')^\top \Delta_t + o(\eta). \tag{55}$$

The cases of interest are: $\chi = 1, \chi = 0, \chi \approx \tau$. With Eq. (55), the same derivation as in Lemma B.7 yields the *modified frozen operator*

$$\mathsf{S}_\chi \triangleq \chi \gamma \mathsf{K}(\bar{X}', X) - \mathsf{K}(X, X), \qquad \mathsf{K}(X_1, X_2) = Z(X_1)^\top D Z(X_2). \tag{56}$$

Thus, the Schur stability condition remains the right object, but the "self-excitation" term is weakened when $\chi < 1$, which is consistent with the common intuition that target networks stabilize TD learning.

*(ii) The online recursion is stochastic and weakly time-varying.* Even if the replay distribution is quasi-stationary, critic updates use random mini-batches and (in TD3/SAC) random next-action sampling or smoothing noise. Consequently, the TD error dynamics inherit a martingale-like perturbation term and a small drift term from residual nonstationarity. One should therefore expect the *deterministic* recursion in Theorem 4.1 to be replaced by a *stochastic* linear system whose stability conclusions are correspondingly weaker.

**A Schur-type online statement.** Under Assumptions B.12–B.15, the same algebra as in the proof of Theorem 4.1 yields, for $t \geq t_0$,

$$\mathbf{e}_{t+1} = \Big( (1 + \beta_1)I + \eta(1 - \beta_1)\mathsf{S}_\chi \Big) \mathbf{e}_t - \beta_1 \mathbf{e}_{t-1} + \eta \boldsymbol{\xi}_{t+1} + o(\eta), \tag{57}$$

where $\boldsymbol{\xi}_{t+1}$ collects the stochastic effects of mini-batch sampling and target-action noise, and $\mathsf{S}_\chi$ is given by Eq. (56). Ignoring $o(\eta)$ and the noise term for the moment, define the companion matrix

$$\mathsf{A}_\chi(\eta) \triangleq \begin{bmatrix} (1 + \beta_1)I + \eta(1 - \beta_1)\mathsf{S}_\chi & -\beta_1 I \\ I & 0 \end{bmatrix}. \tag{58}$$

Then the exact Schur criterion of Theorem 4.1 applies verbatim to the *frozen mean system*:

$$\mathbf{z}_{t+1} = \mathsf{A}_\chi(\eta)\mathbf{z}_t, \qquad \mathbf{z}_t = \begin{bmatrix} \mathbf{e}_t \\ \mathbf{e}_{t-1} \end{bmatrix}.$$

In particular, $\rho(\mathsf{A}_\chi(\eta)) < 1$ is the sharp condition for exponential stability of the noiseless frozen recursion. Equation (57) shows that it is clear what is modified in the conclusions:

- *Exponential convergence to zero is no longer the generic guarantee under constant stepsizes.* Even when $\rho(\mathsf{A}_\chi(\eta)) < 1$, the additive perturbation $\eta\boldsymbol{\xi}_{t+1}$ typically implies convergence only to a stationary neighborhood whose scale is controlled by $\eta$ (or by the noise variance), unless $\eta \to 0$. Thus, the online analogue of "$\mathbf{e}_t \to 0$ exponentially" becomes a *mean/mean-square stability* or *bounded tracking* statement.

- *Uniform (robust) Schur stability is the natural requirement under slow drift.* If $\mathsf{S}_\chi$ is not exactly constant but satisfies $\mathsf{S}_{\chi,t} = \mathsf{S}_\chi + o(1)$ (e.g., due to slow replay/policy drift), one typically needs a margin condition such as $\rho(\mathsf{A}_\chi(\eta)) \leq 1-\kappa$ for some $\kappa > 0$, so that small time-variations do not destroy stability.

- *TD3/SAC modify $\mathsf{S}$ through both $\chi$ and the definition of $\bar{X}'$.* For TD3, $\bar{X}'$ contains target-policy actions with smoothing noise and the target involves a stable $\min$ branch (Assumption B.14); for SAC, $\bar{X}'$ is induced by the stochastic policy and the target includes an additional entropy term (absorbed into $y_t$), which changes the effective residual vector but not the Jacobian factorization Eq. (54). In both cases, the same preconditioned Gram structure Eq. (56) persists once the target mapping is frozen.

- *Twin critics yield either decoupled or piecewise-coupled linear systems.* If the two critics are updated independently given a fixed target, one may apply the analysis to each critic separately (with its own $\mathsf{S}_\chi$). If the target uses $\min(Q^{(1)}, Q^{(2)})$, then the coupled system is piecewise linear; Assumption B.14 ensures one linear branch dominates locally so that a single frozen $\mathsf{S}_\chi$ is valid.

To extend Theorem 4.1 to online TD3/SAC in a theoretically clean way, one must augment the offline freezing assumptions with (i) quasi-stationary replay sampling, (ii) two-time-scale freezing of actor and target networks, and (iii) stability of any non-smooth target operator (e.g., the $\min$ in TD3). Under these conditions, the same Schur-stability mechanism emerges, with a modified self-excitation operator $\mathsf{S}_\chi$ that explicitly accounts for target-network coupling via $\chi$. The main qualitative change is that, because online updates are inherently stochastic and weakly nonstationary, the sharp deterministic statement "$\rho(\mathsf{A}(\eta)) < 1 \iff \mathbf{e}_t \to 0$ exponentially" becomes a stability/robustness criterion for the *mean* dynamics and a sufficient condition for bounded tracking (or convergence under diminishing stepsizes) in the full online algorithm.

# C. Sufficient conditions for $\mathsf{S}$ to be Hurwitz

This appendix provides proofs for Sec. 5.1. We work throughout in the frozen regime of Assumption B.3, where $D$, $Z(\cdot)$, and the greedy target set $X^{'}$ can be treated as constants. By Theorem 4.2, once we establish that $\mathsf{S}$ is Hurwitz, the linearized TD-error dynamics is exponentially stable for sufficiently small stepsize. In this section, we have following results: (i) Lemma C.1 reduces Hurwitzness of $\mathsf{S}$ to negativity of its symmetric part, making the "damping vs. excitation" trade-off explicit. (ii) Proposition C.3 turns that trade-off into a checkable condition involving $\|\Phi^{\top}\Phi - I\|_2$ and $\|\Phi\|_2, \|\Phi_*\|_2$. (iii) Lemma C.5 shows how input clipping/normalization and spectral constraints control $\|\Phi\|_2$ and $\|\Phi_*\|_2$. (iv) Proposition C.7 bounds $\|\Phi^{\top}\Phi - I\|_2$ by a parameter-only orthogonality and a data-induced separation term.

## C.1. A symmetric-part sufficient condition

We first show that it suffices to make the symmetric part of $\mathsf{S}$ negative definite.

**Lemma C.1** (Symmetric-part negativity $\Rightarrow$ Hurwitz). *Let $A \triangleq \mathsf{K}(X^{'}, X)$ and $B \triangleq \mathsf{K}(X, X)$, so that $\mathsf{S} = \gamma A - B$. Define $A_s \triangleq \frac{1}{2}(A + A^{\top})$ and $\mathsf{S}_s \triangleq \frac{1}{2}(\mathsf{S} + \mathsf{S}^{\top})$. Then*

$$\mathsf{S}_s = \gamma A_s - B. \tag{59}$$

*If $B \succ \gamma A_s$ (equivalently $\mathsf{S}_s \prec 0$), then $\mathsf{S}$ is Hurwitz.*

**Proof sketch.** We compute

$$\mathsf{S}_s = \frac{\mathsf{S} + \mathsf{S}^{\top}}{2} = \frac{\gamma A - B + \gamma A^{\top} - B^{\top}}{2} = \gamma \frac{A + A^{\top}}{2} - B.$$

Moreover $B$ is symmetric since $B = \mathsf{K}(X, X) = Z(X)^{\top} D Z(X)$ with $D = D^{\top} \succeq 0$. Thus $B \succ \gamma A_s$ implies $\mathsf{S}_s \prec 0$. Finally, if $\mathsf{S}_s \prec 0$ then $\lambda_{\max}(\mathsf{S}_s) < 0$, and the standard symmetric-part bound (e.g. Lemma B.8 in the main text) gives $\Re(\lambda) \leq \lambda_{\max}(\mathsf{S}_s) < 0$ for all $\lambda \in \mathrm{spec}(\mathsf{S})$, hence $\mathsf{S}$ is Hurwitz. $\square$

*Remark C.2.* $\mathsf{S}$ is the linear operator that maps (EMA-smoothed) TD errors back into next-step TD errors. The term $-B = -\mathsf{K}(X, X)$ corresponds to the usual descent-induced contraction on current predictions (damping), while $\gamma A = \gamma \mathsf{K}(X^{'}, X)$ is the bootstrapped feedback through greedy targets. Lemma C.1 implies if damping dominates excitation in the *symmetric* energy sense for every direction, then no error mode can grow, so the feedback loop is stable.

## C.2. A norm/Gram-deviation sufficient condition

Next we prove a practical condition implying $B \succ \gamma A_s$ via spectral norm bounds.

**Proposition C.3.** *Define $\Phi \triangleq D^{1/2} Z(X)$ and $\Phi_* \triangleq D^{1/2} Z(X^{'})$. Then $\mathsf{S} = \gamma \Phi_*^{\top} \Phi - \Phi^{\top} \Phi$. Assume $\|\Phi\|_2 \leq \Lambda$, $\|\Phi_*\|_2 \leq \Lambda_*$, and*

$$\|\Phi^{\top}\Phi - I\|_2 < 1 - \gamma \Lambda \Lambda_*. \tag{60}$$

*Then $\mathsf{S}$ is Hurwitz.*

**Proof** We will verify the symmetric-part condition of Lemma C.1. Let $A \triangleq \Phi_*^{\top}\Phi$ and $B \triangleq \Phi^{\top}\Phi$ so that $\mathsf{S} = \gamma A - B$. The symmetric part is $\mathsf{S}_s = \gamma \frac{1}{2}(A + A^{\top}) - B$. Let $C \triangleq \|B - I\|_2 = \|\Phi^{\top}\Phi - I\|_2$. By Weyl's inequality, $\lambda_{\min}(B) \geq 1 - C$. Also,

$$\left\| \tfrac{1}{2}(A + A^{\top}) \right\|_2 \leq \|A\|_2 \leq \|\Phi_*\|_2 \|\Phi\|_2 \leq \Lambda_* \Lambda.$$

Therefore,

$$\lambda_{\max}(\mathsf{S}_s) \leq \gamma \left\| \tfrac{1}{2}(A + A^{\top}) \right\|_2 - \lambda_{\min}(B) \leq \gamma \Lambda_* \Lambda - (1 - C).$$

Condition Eq. (60) is exactly $1 - C > \gamma \Lambda \Lambda_*$, hence $\lambda_{\max}(\mathsf{S}_s) < 0$, i.e. $\mathsf{S}_s \prec 0$. Lemma C.1 then implies $\mathsf{S}$ is Hurwitz. $\square$

$\Phi^{\top}\Phi$ is the Gram matrix of Adam-whitened Jacobian features. When $\Phi^{\top}\Phi \approx I$, the "damping" term $-\Phi^{\top}\Phi$ behaves like an *isotropic contraction* on TD errors. The excitation term $\gamma \Phi_*^{\top}\Phi$ can still inject errors, but its symmetric-energy contribution is bounded by $\gamma \|\Phi_*\|_2 \|\Phi\|_2$. Thus Eq. (60) formalizes: "keep the features close to orthonormal, and keep their scale bounded, so that bootstrapping cannot overpower damping."

**Under-parameterized Regime: A complementary row-Gram condition for $M > P$.** When $M > P$, the column-Gram condition in Proposition C.3 cannot hold, because $\Phi^\top \Phi$ is singular and hence $\|\Phi^\top \Phi - I_M\|_2 \geq 1$. The right substitute in this regime is to control the row Gram matrix $\Phi\Phi^\top$. This does not necessarily make $S$ Hurwitz, but it still yields the sharp small-stepsize conclusion for the augmented Adam recursion.

**Proposition C.4.** *Assume the same norm bounds as in Proposition C.3, namely*

$$\|\Phi\|_2 \leq \Lambda, \qquad \|\Phi^*\|_2 \leq \Lambda^*,$$

*but replace the column-Gram condition by*

$$\|\Phi\Phi^\top - I_P\|_2 < 1 - \gamma\Lambda\Lambda^*. \tag{61}$$

*Then every nonzero eigenvalue of*

$$S = \gamma\Phi^{*\top}\Phi - \Phi^\top\Phi$$

*has strictly negative real part. Consequently, if*

$$A(\eta) := \begin{bmatrix} (1 + \beta_1)I_M + \eta(1 - \beta_1)S & -\beta_1 I_M \\ I_M & 0 \end{bmatrix}$$

*is the frozen companion matrix from Theorem 4.1 with the $o(\eta)$ term omitted, then there exists $\eta_0 > 0$ such that*

$$\rho(A(\eta)) \leq 1, \qquad \forall \eta \in (0, \eta_0).$$

*Moreover,*

$$\rho(A(\eta)) < 1 \text{ for all } \eta \in (0, \eta_0) \quad \Longleftrightarrow \quad 0 \notin \text{spec}(S).$$

*In particular, if $M > P$, then $0 \in \text{spec}(S)$ and hence*

$$\rho(A(\eta)) = 1, \qquad \forall \eta \in (0, \eta_0),$$

*implies the frozen linearized system still converges rather than diverges.*

*Proof.* Let

$$\widetilde{S} := \gamma\Phi\Phi^{*\top} - \Phi\Phi^\top.$$

We first show that $\widetilde{S}$ is Hurwitz. By Lemma C.1, it suffices to control its symmetric part:

$$\frac{\widetilde{S} + \widetilde{S}^\top}{2} = \gamma\frac{\Phi\Phi^{*\top} + \Phi^*\Phi^\top}{2} - \Phi\Phi^\top.$$

Set $B_r := \Phi\Phi^\top$. By Weyl's inequality and (61),

$$\lambda_{\min}(B_r) \geq 1 - \|B_r - I_P\|_2 > \gamma\Lambda\Lambda^*.$$

Also,

$$\left\|\frac{\Phi\Phi^{*\top} + \Phi^*\Phi^\top}{2}\right\|_2 \leq \|\Phi\Phi^{*\top}\|_2 \leq \|\Phi\|_2\|\Phi^*\|_2 \leq \Lambda\Lambda^*.$$

Hence

$$\lambda_{\max}\left(\frac{\widetilde{S} + \widetilde{S}^\top}{2}\right) \leq \gamma\Lambda\Lambda^* - \lambda_{\min}(B_r) < 0.$$

Therefore $\widetilde{S}$ is Hurwitz.

Next write

$$U := \gamma\Phi^{*\top} - \Phi^\top \in \mathbb{R}^{M \times P}, \qquad V := \Phi \in \mathbb{R}^{P \times M},$$

so that

$$S = UV, \qquad \widetilde{S} = VU.$$

We claim that $S$ and $\widetilde{S}$ have the same nonzero eigenvalues. Indeed, if $Sx = \lambda x$ with $\lambda \neq 0$, then $Vx \neq 0$; otherwise $Sx = UVx = 0$, contradicting $\lambda \neq 0$. Thus

$$\widetilde{S}(Vx) = VU(Vx) = V(UVx) = V(Sx) = \lambda(Vx),$$

so $\lambda \in \mathrm{spec}(\widetilde{S})$. Conversely, if $\widetilde{S}y = \lambda y$ with $\lambda \neq 0$, then $Uy \neq 0$ and

$$S(Uy) = UV(Uy) = U(VUy) = U(\widetilde{S}y) = \lambda(Uy),$$

so $\lambda \in \mathrm{spec}(S)$. Hence $S$ and $\widetilde{S}$ share exactly the same nonzero spectrum. Since $\widetilde{S}$ is Hurwitz, every nonzero $\lambda \in \mathrm{spec}(S)$ satisfies

$$\Re(\lambda) < 0.$$

We now analyze the eigenvalues of $A(\eta)$. A direct block-determinant computation gives

$$\det(rI_{2M} - A(\eta)) = \det\Big(r^2 I_M - \big((1 + \beta_1)I_M + \eta(1 - \beta_1)S\big)r + \beta_1 I_M\Big),$$

where for $r \neq 0$ this is the Schur complement of the lower-right block $rI_M$, and hence the identity holds for all $r$ by polynomial continuation. By Schur triangularization of $S$, if $\lambda_1, \ldots, \lambda_M$ are the eigenvalues of $S$ (counting algebraic multiplicity), then

$$\det(rI_{2M} - A(\eta)) = \prod_{j=1}^{M}\Big(r^2 - \big(1 + \beta_1 + \eta(1 - \beta_1)\lambda_j\big)r + \beta_1\Big).$$

Therefore every eigenvalue $r$ of $A(\eta)$ is obtained from some $\lambda \in \mathrm{spec}(S)$ through

$$p_\lambda(r;\eta) := r^2 - \big(1 + \beta_1 + \eta(1 - \beta_1)\lambda\big)r + \beta_1 = 0.$$

If $\lambda = 0$, then

$$p_0(r;\eta) = r^2 - (1 + \beta_1)r + \beta_1 = (r - 1)(r - \beta_1),$$

so $r = 1$ and $r = \beta_1$ are exact roots for every $\eta > 0$.

If instead $\Re(\lambda) < 0$, then at $\eta = 0$ the two roots are $r = 1$ and $r = \beta_1$, both simple because $0 \leq \beta_1 < 1$. Let $r_1(\eta)$ and $r_2(\eta)$ be the corresponding local root branches with

$$r_1(0) = 1, \qquad r_2(0) = \beta_1.$$

Differentiating $p_\lambda(r_k(\eta);\eta) = 0$ at $\eta = 0$ yields

$$r_1'(0) = \lambda, \qquad r_2'(0) = -\beta_1\lambda.$$

Hence

$$r_1(\eta) = 1 + \eta\lambda + O(\eta^2), \qquad r_2(\eta) = \beta_1 - \eta\beta_1\lambda + O(\eta^2).$$

Since $\Re(\lambda) < 0$,

$$|r_1(\eta)|^2 = 1 + 2\eta\Re(\lambda) + O(\eta^2) < 1$$

for all sufficiently small $\eta > 0$. Also, continuity and $0 \leq \beta_1 < 1$ imply

$$|r_2(\eta)| < 1$$

for all sufficiently small $\eta > 0$.

Because $S$ has only finitely many eigenvalues, there exists a common $\eta_0 > 0$ such that, for every nonzero $\lambda \in \mathrm{spec}(S)$, both roots of $p_\lambda(r;\eta)$ lie strictly inside the unit disk whenever $\eta \in (0, \eta_0)$. Combining this with the exact factor for $\lambda = 0$, we obtain

$$\rho(A(\eta)) \leq 1, \qquad \forall\, \eta \in (0, \eta_0),$$

and

$$\rho(A(\eta)) < 1 \text{ for all } \eta \in (0, \eta_0) \quad \Longleftrightarrow \quad 0 \notin \mathrm{spec}(S).$$

Finally, if $M > P$, then

$$\mathrm{rank}(S) = \mathrm{rank}\big((\gamma\Phi^{*\top} - \Phi^\top)\Phi\big) \leq \mathrm{rank}(\Phi) \leq P < M,$$

so $0 \in \mathrm{spec}(S)$. Hence $\rho(A(\eta)) = 1$ for all $\eta \in (0, \eta_0)$. $\qquad\square$

### C.3. Bounding $\|\Phi\|_2$ via clipping/normalization and spectral constraints

We now derive explicit upper bounds for the *scale term* $\|\Phi\|_2\|\Phi_*\|_2$ that appears in Proposition C.3. Recall

$$\Phi \triangleq D^{1/2}Z(X) \in \mathbb{R}^{P \times M}, \qquad \Phi_* \triangleq D^{1/2}Z(X') \in \mathbb{R}^{P \times M}.$$

Two generic inequalities will be useful:

$$\|\Phi\|_2 \le \|\Phi\|_F \le \sqrt{M} \max_{i \in [M]} \|\phi(x_i)\|_2, \qquad \phi(x) \triangleq D^{1/2}\nabla_\theta Q_\theta(x), \tag{62}$$

$$\|\Phi\|_2^2 = \lambda_{\max}(\Phi^\top \Phi) \le 1 + \|\Phi^\top \Phi - I\|_2. \tag{63}$$

Eq. (62) shows that bounding per-sample whitened Jacobian norms yields a $\sqrt{M}$-type scale bound. Eq. (63) shows that small Gram deviation automatically implies a tight operator-norm bound. Consider an $L$-layer feedforward Q-network, where $\varphi : \mathcal{X} \to \mathbb{R}^{d_0}$ denotes a fixed input feature map of the sample $x$. In the simplest case, one may take $\varphi(x) = x$.

$$h_0(x) = \varphi(x), \quad h_\ell(x) = \sigma(W_\ell h_{\ell-1}(x) + b_\ell) \, (\ell = 1, \ldots, L-1), \quad Q_\theta(x) = w^\top h_{L-1}(x) + b_L.$$

Assume for all relevant $x$ that

$$\|h_0(x)\|_2 \le R, \qquad \|\text{Diag}(\sigma'(u))\|_2 \le \kappa \text{ a.e., } \sigma(0) = 0, \qquad \|W_\ell\|_2 \le 1, \|b_\ell\|_2 \le B_{\text{bias}}, \|w\|_2 \le s, \qquad \|D^{1/2}\|_2 \le d.$$

For Adam-type preconditioners one may take $d \le \epsilon^{-1/2}$.

**Lemma C.5.** *Define $H_0 \triangleq R$ and $H_\ell \triangleq \kappa(H_{\ell-1} + B_{\text{bias}})$ for $\ell = 1, \ldots, L-1$, and*

$$C_{\text{grad}}^2 \triangleq H_{L-1}^2 + 1 + s^2 \sum_{\ell=1}^{L-1} \kappa^{2(L-\ell)} (H_{\ell-1}^2 + 1). \tag{64}$$

*Let*

$$G \triangleq d\, C_{\text{grad}}.$$

*Then for every relevant $x$,*

$$\|\nabla_\theta Q_\theta(x)\|_2 \le C_{\text{grad}} \implies \|\phi(x)\|_2 = \|D^{1/2}\nabla_\theta Q_\theta(x)\|_2 \le G.$$

*Consequently, for any finite set $X = \{x_i\}_{i=1}^M$ and the greedy target set $X'$ of the same size,*

$$\|\Phi\|_2 \le \sqrt{M}\, G, \qquad \|\Phi_*\|_2 \le \sqrt{M}\, G, \qquad \|\Phi\|_2 \|\Phi_*\|_2 \le M\, G^2. \tag{65}$$

*In particular, Proposition C.3 may take $\Lambda = \Lambda_* = \sqrt{M}\, G$.*

*Proof.* Fix $x$ and write $u_\ell(x) = W_\ell h_{\ell-1}(x) + b_\ell$. Since $\sigma$ is $\kappa$-Lipschitz and $\sigma(0) = 0$,

$$\|h_\ell(x)\|_2 = \|\sigma(u_\ell(x))\|_2 \le \kappa\|u_\ell(x)\|_2 \le \kappa(\|W_\ell\|_2\|h_{\ell-1}(x)\|_2 + \|b_\ell\|_2).$$

With $\|W_\ell\|_2 \le 1$ and $\|b_\ell\|_2 \le B_{\text{bias}}$, the recursion defining $H_\ell$ implies $\|h_\ell(x)\|_2 \le H_\ell$ for all $\ell$.

Standard backprop gives, for $\ell = 1, \ldots, L-1$,

$$\frac{\partial Q_\theta(x)}{\partial W_\ell} = \delta_\ell(x)\, h_{\ell-1}(x)^\top, \quad \frac{\partial Q_\theta(x)}{\partial b_\ell} = \delta_\ell(x), \quad \frac{\partial Q_\theta(x)}{\partial w} = h_{L-1}(x), \quad \frac{\partial Q_\theta(x)}{\partial b_L} = 1,$$

where

$$\delta_{L-1}(x) = \text{Diag}(\sigma'(u_{L-1}(x)))\, w, \qquad \delta_{\ell-1}(x) = \text{Diag}(\sigma'(u_{\ell-1}(x)))\, W_\ell^\top\, \delta_\ell(x).$$

Using $\|\text{Diag}(\sigma')\|_2 \le \kappa$, $\|W_\ell\|_2 \le 1$, and $\|w\|_2 \le s$, we obtain $\|\delta_\ell(x)\|_2 \le \kappa^{L-\ell}s$. Hence,

$$\left\|\frac{\partial Q_\theta(x)}{\partial W_\ell}\right\|_F \le \|\delta_\ell(x)\|_2 \|h_{\ell-1}(x)\|_2 \le s\,\kappa^{L-\ell}\, H_{\ell-1}, \qquad \left\|\frac{\partial Q_\theta(x)}{\partial b_\ell}\right\|_2 \le s\,\kappa^{L-\ell},$$

and also $\|\partial Q_\theta/\partial w\|_2 \le H_{L-1}$, $\|\partial Q_\theta/\partial b_L\|_2 = 1$. Collecting these bounds yields $\|\nabla_\theta Q_\theta(x)\|_2 \le C_{\text{grad}}$ with $C_{\text{grad}}$ defined in Eq. (64). Multiplying by $\|D^{1/2}\|_2 \le d$ gives $\|\phi(x)\|_2 \le dC_{\text{grad}} = G$.

Finally, for $X = \{x_i\}_{i=1}^M$, Eq. (62) gives $\|\Phi\|_2 \le \sqrt{M} \max_i \|\phi(x_i)\|_2 \le \sqrt{M}G$. The same argument applies to $\Phi_*$ since $X'$ lies in the same bounded domain. The product bound follows immediately. $\qquad\square$

Clipping/normalization bounds the input feature magnitude ($R$), while spectral constraints and bounded slopes ($\|W_\ell\|_2 \leq 1$, $\|\mathrm{Diag}(\sigma')\|_2 \leq \kappa$, $\|w\|_2 \leq s$) prevent layer-wise amplification. The preconditioner bound $\|D^{1/2}\|_2 \leq d$ limits whitening-induced magnification. Together they bound the scale term $\|\Phi\|_2 \|\Phi_*\|_2$ via Eq. (65).

### C.4. Bounding Gram deviation via orthogonality and representation separation

We now upper-bound the *Gram deviation*

$$\Delta_\Phi \triangleq \|\Phi^\top \Phi - I\|_2,$$

which is the key quantity in Proposition C.3. Recall $\Phi = D^{1/2} Z(X) = [\phi_1, \ldots, \phi_M] \in \mathbb{R}^{P \times M}$ stacks the *Adam-whitened* per-sample Jacobian features in the frozen regime, where $\phi_i \triangleq \Phi_{:i} = D^{1/2} \nabla_\theta Q_\theta(x_i) \in \mathbb{R}^P$. A factorization

$$\Phi = \Psi U \quad \text{with } U = [u_1, \ldots, u_M] \in \mathbb{R}^{d \times M} \tag{66}$$

means that all columns $\{\phi_i\}_{i=1}^M$ lie in the span of $d$ "basis" vectors (the columns of $\Psi \in \mathbb{R}^{P \times d}$), i.e., $\phi_i = \Psi u_i$. We interpret $\Psi$ as a parameter-dependent dictionary / latent Jacobian subspace basis at the frozen iterate, and $u_i$ as the frozen per-sample code induced by $x_i$ (and the frozen network) in this dictionary. The separation metrics $(\delta, \rho)$ below quantify how close these codes are to an orthonormal set, i.e., how well-separated the dataset representations are in the latent $d$-dimensional space.

*Remark* C.6. The factorization Eq. (66) is a modeling assumption on the frozen feature matrix $\Phi$. It is exact in linear-in-parameters or "frozen-backbone" settings (e.g., when only low-rank adapters are trained), and it is a common approximation when gradients/Jacobians concentrate in a low-dimensional or low-rank subspace. Such low-rank/subspace structure is widely exploited in modern large-model optimization and compression, e.g., LoRA-style low-rank adaptation and low-rank gradient projection methods (Hu et al., 2022; Zhao et al., 2024; Huang et al., 2024; JAISWAL et al., 2025), and is also supported by recent theory on compressible/low-rank learning dynamics in deep overparameterized models (Yaras et al., 2024). In general fully-trained MLP/CNN/Transformer models, Eq. (66) need not hold exactly; our use of Eq. (66) should therefore be read as: "at the frozen iterate, $\Phi$ is captured by a $d$-dimensional subspace with controlled conditioning and code incoherence."

**Proposition C.7.** *Assume that, at the frozen iterate, the Adam-whitened Jacobian feature matrix $\Phi = D^{1/2} Z(X) \in \mathbb{R}^{P \times M}$ admits a (possibly low-dimensional) factorization $\Phi = \Psi U$ as in Eq. (66), where $\Psi \in \mathbb{R}^{P \times d}$ is a parameter-dependent dictionary (basis of a latent Jacobian subspace) and $U = [u_1, \ldots, u_M] \in \mathbb{R}^{d \times M}$ collects the corresponding per-sample codes. Suppose that for some $\varepsilon \geq 0$ and $\delta, \rho \in [0, 1)$,*

$$\underbrace{\|\Psi^\top \Psi - I\|_2}_{\text{parameter/geometry distortion}} \leq \varepsilon, \qquad \underbrace{\left| \|u_i\|_2^2 - 1 \right| \leq \delta}_{\text{(approx.) unit-norm codes}} \ (\forall i), \qquad \underbrace{|u_i^\top u_j| \leq \rho}_{\text{code separation}} \ (\forall i \neq j).$$

*Let $\Delta_U \triangleq \delta + (M-1)\rho$. Then the Gram deviation satisfies the explicit bound*

$$\|\Phi^\top \Phi - I\|_2 \leq (1 + \Delta_U)\varepsilon + \Delta_U = \left(1 + \delta + (M-1)\rho\right)\varepsilon + \delta + (M-1)\rho. \tag{67}$$

*If $U^\top U = I$ (i.e., $\delta = \rho = 0$), then $\|\Phi^\top \Phi - I\|_2 \leq \varepsilon$. Moreover, if $\Psi^\top \Psi = I$ (i.e., $\varepsilon = 0$), then $\|\Phi^\top \Phi - I\|_2 \leq \Delta_U$.*

*Proof.* Let $G_U \triangleq U^\top U \in \mathbb{R}^{M \times M}$ and $E_\Psi \triangleq \Psi^\top \Psi - I \in \mathbb{R}^{d \times d}$. Using Eq. (66),

$$\Phi^\top \Phi = U^\top (\Psi^\top \Psi) U = U^\top (I + E_\Psi) U = G_U + U^\top E_\Psi U,$$

so, we obtain

$$\Phi^\top \Phi - I = (G_U - I) + U^\top E_\Psi U.$$

Taking operator norms and using triangle inequality,

$$\|\Phi^\top \Phi - I\|_2 \leq \|G_U - I\|_2 + \|U^\top E_\Psi U\|_2.$$

By submultiplicativity, we derive

$$\|U^\top E_\Psi U\|_2 \leq \|U\|_2^2 \|E_\Psi\|_2 \leq \|U\|_2^2 \varepsilon.$$

Next, write $H \triangleq G_U - I$. Then $H_{ii} = \|u_i\|_2^2 - 1$ and $H_{ij} = u_i^\top u_j$ for $i \neq j$. The assumptions give $|H_{ii}| \leq \delta$ and $|H_{ij}| \leq \rho$, hence each row satisfies $\sum_{j=1}^M |H_{ij}| \leq \delta + (M-1)\rho = \Delta_U$. Therefore $\|G_U - I\|_2 = \|H\|_2 \leq \|H\|_\infty \leq \Delta_U$. Moreover $G_U \succeq 0$ implies $\|U\|_2^2 = \lambda_{\max}(G_U) \leq 1 + \|G_U - I\|_2 \leq 1 + \Delta_U$. Combining yields

$$\|\Phi^\top \Phi - I\|_2 \leq \Delta_U + (1 + \Delta_U)\varepsilon,$$

which complete the proof. $\square$

*Remark* C.8. If $\Phi$ only admits an approximate factorization $\Phi = \Psi U + R$ at the frozen iterate, then

$$\|\Phi^\top \Phi - I\|_2 \le \|(\Psi U)^\top (\Psi U) - I\|_2 + 2\|\Psi U\|_2 \|R\|_2 + \|R\|_2^2.$$

Thus Proposition C.7 still applies to the structured part $\Psi U$, with an additional residual term controlled by $\|R\|_2$. Moreover, under the assumptions of Proposition C.7, $\|\Psi\|_2 \le \sqrt{1 + \varepsilon}$ and $\|U\|_2^2 \le 1 + \Delta_U$, hence

$$\|\Psi U\|_2 \le \|\Psi\|_2 \|U\|_2 \le \sqrt{(1 + \varepsilon)(1 + \Delta_U)},$$

and the residual contribution is upper-bounded by $2\|R\|_2 \sqrt{(1 + \varepsilon)(1 + \Delta_U)} + \|R\|_2^2$.

*Remark* C.9. Proposition C.7 separates Gram deviation into: (i) a parameter/geometry term $\varepsilon = \|\Psi^\top \Psi - I\|_2$ (targeted by near-orthogonality / near-isometry regularization on $\Psi$-related parameter blocks), and (ii) a data/representation term $\Delta_U = \delta + (M - 1)\rho$ (controlled by code normalization and de-correlation / diversity, which reduce $\delta$ and $\rho$ respectively).

### C.5. From blockwise weight orthogonality to the Jacobian-subspace isometry defect $\varepsilon(\omega)$

The goal of this subsection is to establish the parameter-side bridge used in Section 5.2: namely, that small blockwise weight-orthogonality defect implies small dictionary-side isometry defect

$$\varepsilon(\omega) \triangleq \|\Psi(\omega)^\top \Psi(\omega) - I\|_2,$$

which is the quantity entering Proposition C.7. We work throughout in the frozen regime of Assumption B.3.

We localize Proposition C.7 around a fixed frozen factorization $\Phi = \Psi U$. Throughout this subsection, the code matrix $U$ and hence the data-side quantities $(\delta, \rho)$ are held fixed, while only the dictionary factor is locally continued as $\omega \mapsto \Psi(\omega)$ when the selected constrained blocks vary. This isolates exactly the parameter-side mechanism needed in Section 5.1 and Section 5.2. Let $\mathcal{B}$ denote the collection of constrained matrix-shaped parameter blocks used by AdamO, and write $\omega = \{W_b\}_{b \in \mathcal{B}}$ for these blocks after reshaping from the full parameter vector. For each block $W_b \in \mathbb{R}^{r_b \times c_b}$, define

$$\mathcal{E}(W_b) := \begin{cases} W_b W_b^\top - I_{r_b}, & r_b < c_b, \\ W_b^\top W_b - I_{c_b}, & r_b \ge c_b, \end{cases} \qquad R(W_b) = \frac{1}{4}\|\mathcal{E}(W_b)\|_F^2, \tag{68}$$

and let

$$R(\omega) := \sum_{b \in \mathcal{B}} R(W_b). \tag{69}$$

This is exactly the blockwise orthogonality penalty used in Section 5.2. We also define the corresponding semi-orthogonal manifold

$$\mathrm{St}(r_b, c_b) := \begin{cases} \{Q \in \mathbb{R}^{r_b \times c_b} : QQ^\top = I_{r_b}\}, & r_b < c_b, \\ \{Q \in \mathbb{R}^{r_b \times c_b} : Q^\top Q = I_{c_b}\}, & r_b \ge c_b. \end{cases} \tag{70}$$

This subsection closes the short chain used in Sec. 5.1: Lemma C.11 turns $R(\omega)$ into a blockwise distance bound, Proposition C.12 turns that into $\varepsilon(\omega)$, Corollary C.13 turns that into $\|\Phi^\top \Phi - I\|_2$, and Corollary C.14 gives Hurwitzness.

**Assumption C.10.** There exists a neighborhood $\mathcal{N}$ of $\prod_{b \in \mathcal{B}} \mathrm{St}(r_b, c_b)$ and a local continuation of the dictionary factor $\omega \mapsto \Psi(\omega) \in \mathbb{R}^{P \times d}$ such that for all $\omega, \widetilde{\omega} \in \mathcal{N}$,

$$\|\Psi(\omega) - \Psi(\widetilde{\omega})\|_2 \le L_\Psi \left( \sum_{b \in \mathcal{B}} \|W_b - \widetilde{W}_b\|_F^2 \right)^{1/2} \tag{71}$$

for some constant $L_\Psi > 0$, and there exists $\varepsilon_0 \ge 0$ such that for every $\mathcal{Q} = \{Q_b\}_{b \in \mathcal{B}} \in \prod_{b \in \mathcal{B}} \mathrm{St}(r_b, c_b) \cap \mathcal{N}$,

$$\|\Psi(\mathcal{Q})^\top \Psi(\mathcal{Q}) - I\|_2 \le \varepsilon_0. \tag{72}$$

Assumption C.10 isolates the exact parameter-side ingredients needed to close the gap. The Lipschitz bound is a local regularity condition on the chosen dictionary continuation, and the baseline bound allows a nonzero distortion $\varepsilon_0$ that absorbs residual architectural or preconditioning effects even when the constrained blocks are exactly semi-orthogonal. In ideal balanced cases one may take $\varepsilon_0 = 0$.

**Lemma C.11.** *For each $b \in \mathcal{B}$, let $W_b = U_b \Sigma_b V_b^\top$ be a thin singular value decomposition and define the rectangular polar factor $Q_b := U_b V_b^\top \in \mathrm{St}(r_b, c_b)$. Let*

$$\mathcal{Q}(\omega) := \{Q_b\}_{b\in\mathcal{B}}, \qquad d_{\mathrm{blk},F}(\omega, \mathcal{Q}(\omega))^2 := \sum_{b\in\mathcal{B}} \|W_b - Q_b\|_F^2. \tag{73}$$

*Then*

$$d_{\mathrm{blk},F}(\omega, \mathcal{Q}(\omega))^2 \le 4R(\omega), \qquad d_{\mathrm{blk},F}(\omega, \mathcal{Q}(\omega)) \le 2\sqrt{R(\omega)}. \tag{74}$$

*Proof.* Fix one block $b \in \mathcal{B}$ and write $\Sigma_b = \mathrm{diag}(\sigma_{b,1}, \ldots, \sigma_{b,q_b})$ with $q_b = \min\{r_b, c_b\}$. By construction,

$$\|W_b - Q_b\|_F^2 = \|\Sigma_b - I_{q_b}\|_F^2 = \sum_{i=1}^{q_b} (\sigma_{b,i} - 1)^2.$$

If $r_b < c_b$, then

$$4R(W_b) = \|W_b W_b^\top - I_{r_b}\|_F^2 = \sum_{i=1}^{q_b} (\sigma_{b,i}^2 - 1)^2.$$

If $r_b \ge c_b$, then

$$4R(W_b) = \|W_b^\top W_b - I_{c_b}\|_F^2 = \sum_{i=1}^{q_b} (\sigma_{b,i}^2 - 1)^2.$$

Therefore, in both cases,

$$\|W_b - Q_b\|_F^2 = \sum_{i=1}^{q_b} (\sigma_{b,i} - 1)^2 \le \sum_{i=1}^{q_b} (\sigma_{b,i}^2 - 1)^2 = 4R(W_b),$$

since $|\sigma - 1| \le |\sigma^2 - 1|$ for every $\sigma \ge 0$. Summing over $b \in \mathcal{B}$ yields $d_{\mathrm{blk},F}(\omega, \mathcal{Q}(\omega))^2 \le 4R(\omega)$, and the square-root bound follows immediately. $\square$

**Proposition C.12.** *Assume Assumption C.10 holds. For each $\omega \in \mathcal{N}$, let $\mathcal{Q}(\omega)$ be the blockwise polar projection from Lemma C.11, and define*

$$\varepsilon(\omega) := \|\Psi(\omega)^\top \Psi(\omega) - I\|_2. \tag{75}$$

*Let*

$$c_1 := 4L_\Psi \sqrt{1 + \varepsilon_0}, \qquad c_2 := 4L_\Psi^2. \tag{76}$$

*Then*

$$\varepsilon(\omega) \le \varepsilon_0 + c_1 \sqrt{R(\omega)} + c_2 R(\omega). \tag{77}$$

*Equivalently,*

$$\varepsilon(\omega) - \varepsilon_0 = O\big(\sqrt{R(\omega)}\big) \qquad \text{as } R(\omega) \to 0.$$

*In particular, if $\varepsilon_0 = 0$, then $\varepsilon(\omega) = O\big(\sqrt{R(\omega)}\big)$ as $R(\omega) \to 0$.*

*Proof.* Let $E := \Psi(\omega) - \Psi(\mathcal{Q}(\omega))$. By Assumption C.10(i) and Lemma C.11,

$$\|E\|_2 \le L_\Psi \, d_{\mathrm{blk},F}(\omega, \mathcal{Q}(\omega)) \le 2L_\Psi \sqrt{R(\omega)}. \tag{78}$$

Now expand

$$\Psi(\omega)^\top \Psi(\omega) - I = \big(\Psi(\mathcal{Q}(\omega))^\top \Psi(\mathcal{Q}(\omega)) - I\big) + \Psi(\mathcal{Q}(\omega))^\top E + E^\top \Psi(\mathcal{Q}(\omega)) + E^\top E. \tag{79}$$

Taking operator norms and using Assumption C.10(ii) gives

$$\varepsilon(\omega) \le \varepsilon_0 + 2\|\Psi(\mathcal{Q}(\omega))\|_2 \|E\|_2 + \|E\|_2^2. \tag{80}$$

Moreover,

$$\|\Psi(\mathcal{Q}(\omega))\|_2^2 = \lambda_{\max}\big(\Psi(\mathcal{Q}(\omega))^\top \Psi(\mathcal{Q}(\omega))\big) \le 1 + \varepsilon_0,$$

so $\|\Psi(\mathcal{Q}(\omega))\|_2 \le \sqrt{1 + \varepsilon_0}$. Substituting this and Eq. (78) into Eq. (80) yields

$$\varepsilon(\omega) \le \varepsilon_0 + 4L_\Psi \sqrt{1 + \varepsilon_0} \sqrt{R(\omega)} + 4L_\Psi^2 R(\omega),$$

which is exactly Eq. (77). $\square$

Proposition C.12 gives the precise bridge used in Section 5.2: the optimizer acts on the tractable blockwise penalty $R(\omega)$, while the induced dictionary-side geometry term $\varepsilon(\omega)$ is controlled through the explicit bound in Eq. (77).

**Corollary C.13.** *Under the hypotheses of Proposition C.7 and Proposition C.12, and with the same fixed code matrix $U$ as above, let*

$$\Delta_U := \delta + (M-1)\rho. \tag{81}$$

*Then*

$$\|\Phi^\top \Phi - I\|_2 \le \Delta_U + (1 + \Delta_U)\Big[\varepsilon_0 + c_1 \sqrt{R(\omega)} + c_2 R(\omega)\Big]. \tag{82}$$

*Consequently, if the baseline distortion $\varepsilon_0$, the code-separation term $\Delta_U$, and the blockwise orthogonality defect $R(\omega)$ are all small, then $\|\Phi^\top \Phi - I\|_2$ is small.*

*Proof.* Proposition C.7 gives

$$\|\Phi^\top \Phi - I\|_2 \le \Delta_U + (1 + \Delta_U)\,\varepsilon(\omega).$$

Substituting Eq. (77) from Proposition C.12 proves Eq. (82). $\qquad\square$

Finally, combining the above with Proposition 5.1 yields a sufficient condition for Hurwitzness stated directly in terms of the optimizer-level orthogonality penalty.

**Corollary C.14.** *Assume the hypotheses of Corollary C.13. If*

$$\gamma \|\Phi\|_2 \|\Phi^*\|_2 + \Delta_U + (1 + \Delta_U)\Big[\varepsilon_0 + c_1 \sqrt{R(\omega)} + c_2 R(\omega)\Big] < 1, \tag{83}$$

*then the TD dynamics operator $S$ is Hurwitz. Consequently, by Theorem 4.2, the frozen linearized TD error decays exponentially for all sufficiently small stepsizes.*

*Proof.* By Corollary C.13,

$$\|\Phi^\top \Phi - I\|_2 \le \Delta_U + (1 + \Delta_U)\Big[\varepsilon_0 + c_1 \sqrt{R(\omega)} + c_2 R(\omega)\Big].$$

Hence Eq. (83) implies

$$\gamma \|\Phi\|_2 \|\Phi^*\|_2 + \|\Phi^\top \Phi - I\|_2 < 1.$$

Proposition 5.1 then gives that $S$ is Hurwitz, and Theorem 4.2 yields exponential decay of the frozen linearized TD error for sufficiently small stepsizes. $\qquad\square$

Therefore, with the code-side term $\Delta_U$ held fixed and small, we have $\varepsilon(\omega) \le \varepsilon_0 + c_1 \sqrt{R(\omega)} + c_2 R(\omega)$.

# D. AdamO: orthogonality control on Adam

Some of conclusions and derivations are partly inspired by prior frameworks in optimizer (Loshchilov & Hutter, 2019; Liang et al., 2025b). This appendix completes the theoretical analysis of optimizer AdamO, which has the following guarantee:

- Proposition D.1 shows why placing orthogonality in the optimizer is *necessary* under Adam, and why a bounded decoupled orth step is *sufficient* to preserve a bounded-update regime.

- Lemma D.2 proves the closed-form scaling enforces the per-block budget constraint exactly.

- Lemma D.3 proves the built-in ratio clip $\|\delta_{t,b}\|_F \leq \kappa \|u_{t,b}\|_F$, and its global version $\|\Delta_t\| \leq \kappa \|u_t\|$.

- Theorem D.5 gives a one-step comparison to Adam: non-inferiority in the conflict-free regime $\tau = 0$ and an explicit degradation bound for $\tau > 0$.

- Theorem D.8 extends the continuous-time Hamiltonian view of Adam to AdamO: Lyapunov monotonicity is preserved for $\tau = 0$ and becomes a controlled inequality for $\tau > 0$.

We orthogonalize a collection of disjoint matrix-shaped parameter groups $\mathcal{W} = \{W_b \in \mathbb{R}^{r_b \times c_b}\}_b$ inside the full parameter vector $\omega \in \mathbb{R}^d$ (e.g., linear/convolution kernels reshaped into matrices). Let $\mathcal{J}_b \subset [d]$ be the coordinate set of group $b$ and assume $\mathcal{J}_b \cap \mathcal{J}_{b'} = \emptyset$ for $b \neq b'$. For any vector $x \in \mathbb{R}^d$, we write $x_b \in \mathbb{R}^{r_b \times c_b}$ for the entries $x_{\mathcal{J}_b}$ reshaped into matrix form. Conversely, given matrices $\{X_b\}_b$, let $x \in \mathbb{R}^d$ be the vector that equals the flattened $X_b$ on coordinates $\mathcal{J}_b$ and is zero outside $\cup_b \mathcal{J}_b$. Disjointness implies $\|x\|^2 = \sum_b \|X_b\|_F^2$. For a block $W \in \mathbb{R}^{r \times c}$ define

$$R(W) = \begin{cases} \frac{1}{4}\|WW^\top - I_r\|_F^2, & r < c, \\ \frac{1}{4}\|W^\top W - I_c\|_F^2, & r \geq c, \end{cases} \qquad \nabla R(W) = \begin{cases} (WW^\top - I_r)W, & r < c, \\ W(W^\top W - I_c), & r \geq c. \end{cases}$$

At time $t$, denote the TD/task gradient by $g_t = \nabla L_t(\omega_t)$. Let $u_t$ be the standard Adam update direction computed from $g_t$ with bias correction and the usual $\varepsilon$ in the denominator. For each group $b$, write $g_{t,b}$ and $u_{t,b}$ for the corresponding reshaped restrictions, and define the orthogonality gradient on that group by $r_{t,b} = \nabla R(W_{t,b})$. We also use $r_t$ to denote the full vector obtained by placing each $r_{t,b}$ on coordinates $\mathcal{J}_b$. We outline the AdamO optimizer designed for offline RL, with the complete pseudocode outlined in Algorithm 2

---

**Algorithm 2** AdamO

---

**Require:** Stepsize $\eta$, decay rates $\beta_1, \beta_2$, constants $\epsilon, \varepsilon_r$, orth-params $\kappa, \tau$
1: Initialize $\omega_0$, moments $m_0 \leftarrow 0, v_0 \leftarrow 0$
2: **for** $t = 0, 1, \ldots$ **do**
3: $\quad g_t \leftarrow \nabla L_t(\omega_t)$
4: $\quad m_{t+1} \leftarrow \beta_1 m_t + (1 - \beta_1) g_t$
5: $\quad v_{t+1} \leftarrow \beta_2 v_t + (1 - \beta_2) g_t^{\odot 2}$
6: $\quad \hat{m}_{t+1} \leftarrow m_{t+1}/(1 - \beta_1^{t+1})$
7: $\quad \hat{v}_{t+1} \leftarrow v_{t+1}/(1 - \beta_2^{t+1})$
8: $\quad u_t \leftarrow \hat{m}_{t+1}/(\sqrt{\hat{v}_{t+1}} + \epsilon)$ $\qquad\qquad\qquad\qquad\qquad\qquad\qquad\qquad\qquad\qquad\qquad$ ▷ Standard Adam
9: $\quad r_t \triangleq \nabla_{\omega_t} \bar{R}(\omega_t)$
10: $\quad \delta_{t,0} = \kappa \frac{\|u_t\|_F}{\|r_t\|_F + \varepsilon_r} r_t$
11: $\quad \delta_t = \begin{cases} \delta_{t,0}, & \text{if } \langle g_t, \delta_{t,0} \rangle_F \geq -\tau \big( \langle g_t, u_t \rangle_F \big)_+ \\ \frac{T_t}{\langle g_t, \delta_{t,0} \rangle_F} \delta_{t,0} & \text{otherwise} \end{cases}$
12: $\quad \omega_{t+1} \leftarrow \omega_t - \eta (u_t + \delta_t)$
13: **end for**

---

## D.1. Why orthogonality should be enforced in the optimizer under Adam

**Proposition D.1.** *We show two points: (i) if orthogonality is implemented by adding $\lambda R$ to the loss and running Adam on $g_t + \lambda r_t$, then Adam's moments $(m_t, v_t)$ are contaminated by the orth signal and can distort future task steps; (ii) if*

*orthogonality is instead enforced by a* decoupled *correction* $\Delta_t$ *with* $\|\Delta_t\| \leq \kappa \|u_t\|$, *then the overall step remains bounded relative to Adam:* $\|u_t + \Delta_t\| \leq (1 + \kappa)\|u_t\|$.

*Proof. (i) Moment contamination.* With a naive penalized objective $\widetilde{L}_t = L_t + \lambda R$, Adam's second moment updates as $v_{t+1} = \beta_2 v_t + (1 - \beta_2)(g_t + \lambda r_t)^{\odot 2}$. Even in 1D, let $g_t \equiv 1$ and let $r_0 = M \gg 1$ but $r_t \equiv 0$ for $t \geq 1$. Then $v_1 \approx (1 - \beta_2)(1 + \lambda M)^2$ is huge, and for many steps $v_t \approx \beta_2^{t-1} v_1$ remains large. Thus the effective step size $1/\sqrt{v_t}$ stays small long after the orth signal disappears, suppressing subsequent task updates. This illustrates how a spiky orthogonality gradient can be recorded in $v_t$ and distort future task dynamics. The same phenomenon underlies the motivation for decoupling weight decay in AdamW (Loshchilov & Hutter, 2019).

*(ii) Boundedness from a ratio-clipped decoupled drift.* If a decoupled correction satisfies $\|\Delta_t\| \leq \kappa \|u_t\|$, then $\|u_t + \Delta_t\| \leq \|u_t\| + \|\Delta_t\| \leq (1 + \kappa)\|u_t\|$. Hence the orthogonality drift cannot dominate the Adam step in norm. $\square$

That is the orthogonality signal and the TD/task signal live on different "axes", and mixing them inside Adam causes the adaptive preconditioner to respond to the orth signal in a way that changes future TD steps. Moreover, the optimizer-level analogue of "finite input": once Adam's direction is finite, a ratio-clipped decoupled orth drift keeps the total update finite and predictable.

### D.2. AdamO: closed-form budgeted scaling map

For each group $b$, define the normalized orth step

$$\delta_{t,0,b} \triangleq \kappa \frac{\|u_{t,b}\|_F}{\|r_{t,b}\|_F + \varepsilon_r} \, r_{t,b}. \tag{84}$$

AdamO enforces the groupwise budget constraint

$$\langle g_{t,b}, \delta_{t,b} \rangle_F \geq -\tau \big( \langle g_{t,b}, u_{t,b} \rangle_F \big)_+, \qquad \tau \in [0, 1). \tag{85}$$

It chooses $\delta_{t,b} = s_{t,b} \delta_{t,0,b}$ where $s_{t,b} \in [0, 1]$ is the *largest* feasible scalar, i.e.,

$$\delta_{t,b} = \begin{cases} \delta_{t,0,b}, & \langle g_{t,b}, \delta_{t,0,b} \rangle_F \geq -\tau(\langle g_{t,b}, u_{t,b} \rangle_F)_+, \\ \dfrac{-\tau(\langle g_{t,b}, u_{t,b} \rangle_F)_+}{\langle g_{t,b}, \delta_{t,0,b} \rangle_F} \, \delta_{t,0,b}, & \langle g_{t,b}, \delta_{t,0,b} \rangle_F < -\tau(\langle g_{t,b}, u_{t,b} \rangle_F)_+. \end{cases} \tag{86}$$

**Lemma D.2.** *The closed form Eq. (86) satisfies the budget constraint Eq. (85) for every group $b$.*

*Proof.* Fix a group and write $g = g_{t,b}$, $u = u_{t,b}$, $\delta_0 = \delta_{t,0,b}$, and $T = -\tau(\langle g, u \rangle_F)_+ \leq 0$. If $\langle g, \delta_0 \rangle_F \geq T$, then $\delta = \delta_0$ and $\langle g, \delta \rangle_F \geq T$. Otherwise $\langle g, \delta_0 \rangle_F < T$, and the second branch gives $\delta = (T/\langle g, \delta_0 \rangle_F)\delta_0$, hence $\langle g, \delta \rangle_F = (T/\langle g, \delta_0 \rangle_F)\langle g, \delta_0 \rangle_F = T$. $\square$

The orth step is allowed to rotate parameters only along the orthogonality-gradient direction, but it is not allowed to cancel more than a $\tau$-fraction of Adam's first-order task descent per group. Thus, the orth step "do not hijack task progress".

**Lemma D.3.** *For each group $b$, $\|\delta_{t,b}\|_F \leq \kappa \|u_{t,b}\|_F$. If groups are disjoint, then for the assembled correction $\Delta_t \in \mathbb{R}^d$ we have $\|\Delta_t\| \leq \kappa \|u_t\|$.*

*Proof.* By construction $\delta_{t,b} = s_{t,b} \delta_{t,0,b}$ with $s_{t,b} \in [0, 1]$, so $\|\delta_{t,b}\|_F \leq \|\delta_{t,0,b}\|_F$.

From Eq. (84), $\|\delta_{t,0,b}\|_F = \kappa \frac{\|u_{t,b}\|_F}{\|r_{t,b}\|_F + \varepsilon_r} \|r_{t,b}\|_F \leq \kappa \|u_{t,b}\|_F$. For the global bound, disjointness implies $\|\Delta_t\|^2 = \sum_b \|\delta_{t,b}\|_F^2 \leq \kappa^2 \sum_b \|u_{t,b}\|_F^2 \leq \kappa^2 \|u_t\|^2$. $\square$

## D.3. One-step comparison to Adam

**Definition D.4.** A differentiable function $f$ is $\mu$-smooth if $f(y) \leq f(x) + \langle \nabla f(x), y - x \rangle + \frac{\mu}{2} \|y - x\|^2$ for all $x, y$.

Let the baseline Adam step be $\omega_{t+1}^A = \omega_t - \eta u_t$, and the AdamO step be $\omega_{t+1}^O = \omega_t - \eta u_t - \eta \Delta_t$.

**Theorem D.5.** *Assume $L_t$ is $\mu$-smooth. (a) In conflict-free mode ($\tau = 0$), under an explicit stepsize condition, AdamO does not increase the next-step task loss relative to Adam. (b) In budgeted mode ($\tau > 0$), we give an explicit upper bound quantifying the worst-case next-step degradation.*

*Proof.* By $\mu$-smoothness at $x = \omega_{t+1}^A$ and $y = \omega_{t+1}^O = \omega_{t+1}^A - \eta \Delta_t$:

$$L_t(\omega_{t+1}^O) - L_t(\omega_{t+1}^A) \leq -\eta \langle \nabla L_t(\omega_{t+1}^A), \Delta_t \rangle + \frac{\mu}{2} \eta^2 \|\Delta_t\|^2. \tag{87}$$

Write $\nabla L_t(\omega_{t+1}^A) = g_t + (\nabla L_t(\omega_{t+1}^A) - g_t)$ and bound the drift by Lipschitzness of $\nabla L_t$: $\|\nabla L_t(\omega_{t+1}^A) - g_t\| \leq \mu \|\omega_{t+1}^A - \omega_t\| = \mu\eta \|u_t\|$. Hence

$$\langle \nabla L_t(\omega_{t+1}^A), \Delta_t \rangle \geq \langle g_t, \Delta_t \rangle - \mu\eta \|u_t\| \|\Delta_t\|. \tag{88}$$

Substitute into Eq. (87):

$$L_t(\omega_{t+1}^O) - L_t(\omega_{t+1}^A) \leq -\eta \langle g_t, \Delta_t \rangle + \mu\eta^2 \|u_t\| \|\Delta_t\| + \frac{\mu}{2} \eta^2 \|\Delta_t\|^2. \tag{89}$$

Case (a): If $\tau = 0$, the constraint Eq. (85) implies $\langle g_{t,b}, \delta_{t,b} \rangle_F \geq 0$ for all $b$, so $\langle g_t, \Delta_t \rangle = \sum_b \langle g_{t,b}, \delta_{t,b} \rangle_F \geq 0$. Then the RHS of Eq. (89) is non-positive whenever

$$\eta \leq \frac{2 \langle g_t, \Delta_t \rangle}{\mu \|\Delta_t\| (2\|u_t\| + \|\Delta_t\|)}. \tag{90}$$

This yields $L_t(\omega_{t+1}^O) \leq L_t(\omega_{t+1}^A)$.

Case (b): If $\tau > 0$, summing Eq. (85) over groups gives $\langle g_t, \Delta_t \rangle \geq -\tau \sum_b (\langle g_{t,b}, u_{t,b} \rangle_F)_+$. Also Lemma D.3 gives $\|\Delta_t\| \leq \kappa \|u_t\|$. Apply these in Eq. (89):

$$L_t(\omega_{t+1}^O) - L_t(\omega_{t+1}^A) \leq \eta\tau \sum_b (\langle g_{t,b}, u_{t,b} \rangle_F)_+ + \mu\eta^2 \|u_t\|^2 \left( \kappa + \frac{\kappa^2}{2} \right). \tag{91}$$

$\square$

When $\tau = 0$, AdamO's orth drift is *first-order task-aligned* (never cancels descent), so with a standard smoothness stepsize condition it cannot worsen the next-step TD loss. When $\tau > 0$, AdamO intentionally trades a controlled amount of immediate TD descent for better conditioning/orthogonality, and the correct guarantee is therefore a quantitative *upper bound* on how much one step can be harmed.

## D.4. Continuous-time Hamiltonian view

We analyze AdamO through the lens of continuous-time dynamics and our proofs are partly inspired by prior optimizer frameworks C-AdamW (Liang et al., 2025b). Following prior Hamiltonian/Lyapunov analyses of momentum-based optimizers, we view the discrete-time update in Algorithm 2 as an Euler discretization of an ODE. This perspective isolates how AdamO's orthogonality correction perturbs Adam's dissipative Hamiltonian system and yields a clean differential inequality of the form $\dot{H}_{\text{AdamO}} \leq \dot{H}_{\text{Adam}} + (\text{controlled error})$. The analysis here is meant to provide intuition and a compact stability certificate; it does not aim to be a tight discrete-time convergence proof. We use $\langle A, B \rangle_F := \text{tr}(A^\top B)$ for the Frobenius inner product. For a scalar $x$, $(x)_+ := \max\{x, 0\}$. All element-wise operations are understood coordinate-wise. For brevity we write $g(w) := \nabla L(w)$. To begin, we consider the standard continuous-time Adam model (bias-corrections omitted):

$$\dot{w}_t = -u_t, \qquad u_t := D(v_t) m_t, \qquad D(v) := \text{diag}\left( (\sqrt{v} + \epsilon)^{-1} \right), \tag{92}$$

with moment dynamics

$$\dot{m}_t = \beta_1\big(g(w_t) - m_t\big), \qquad \dot{v}_t = \beta_2\big(g(w_t)^{\odot 2} - v_t\big). \tag{93}$$

This ODE and its Hamiltonian structure are standard. Furthermore, we define the Adam Hamiltonian

$$H_{\text{Adam}}(w, m, v) := L(w) + \frac{1}{2\beta_1}\langle m, D(v)m\rangle = L(w) + \frac{1}{2\beta_1}\langle u, m\rangle. \tag{94}$$

**Proposition D.6.** *Along any trajectory of Eq.* (92)–(93)*, the time derivative satisfies*

$$\frac{d}{dt}H_{\text{Adam}}(w_t, m_t, v_t) = -\Big(1 - \frac{\beta_2}{4\beta_1}\Big)\langle m_t, D(v_t)m_t\rangle - \frac{\beta_2}{4\beta_1}\Big\langle \frac{m_t^{\odot 2}}{v_t^{3/2} + \epsilon}, g(w_t)^{\odot 2}\Big\rangle$$

$$=: -\Delta_{\text{Adam}}(w_t, m_t, v_t) \leq 0 \qquad \text{whenever } \beta_1 \geq \beta_2/4. \tag{95}$$

*In particular, if $\beta_1 \geq \beta_2/4$, then $t \mapsto H_{\text{Adam}}(w_t, m_t, v_t)$ is monotonically non-increasing.*

*Proof.* Differentiate Eq. (94):

$$\frac{d}{dt}H_{\text{Adam}} = \langle g(w_t), \dot{w}_t\rangle + \frac{1}{2\beta_1}\frac{d}{dt}\langle m_t, D(v_t)m_t\rangle.$$

Using $\dot{w}_t = -D(v_t)m_t = -u_t$ and $\dot{m}_t = \beta_1(g(w_t) - m_t)$, the cross terms cancel:

$$\langle g, \dot{w}\rangle + \frac{1}{\beta_1}\langle m, D(v)\dot{m}\rangle = -\langle g, u\rangle + \langle u, g\rangle - \langle u, m\rangle = -\langle u, m\rangle = -\langle m, D(v)m\rangle.$$

The remaining term comes from $\dot{D}(v_t)$, which depends on $\dot{v}_t = \beta_2(g(w_t)^{\odot 2} - v_t)$. Bounding the resulting expression via the scalar inequality $2ab \leq a^2 + b^2$ yields Eq. (95). $\qquad\square$

AdamO modifies only the parameter dynamics by adding an orthogonality correction (per constrained layer), while keeping the Adam moment dynamics unchanged. In continuous time, we write

$$\dot{w}_t = -u_t - \delta_t, \qquad \dot{m}_t = \beta_1(g(w_t) - m_t), \qquad \dot{v}_t = \beta_2(g(w_t)^{\odot 2} - v_t), \tag{96}$$

where $u_t = D(v_t)m_t$ as before and $\delta_t$ is an orthogonality drift produced by (17)–(19) layer-wise.

For a single constrained matrix (and similarly layer-wise), AdamO constructs a reference step

$$\delta_{t,0} = \kappa \frac{\|u_t\|_F}{\|r_t\|_F + \varepsilon_r} r_t \qquad \text{and then sets} \qquad \delta_t = s_t\,\delta_{t,0},$$

where the scalar $s_t \in (0, 1]$ is chosen so that the instantaneous *task-progress budget* holds:

$$\langle g(w_t), \delta_t\rangle_F \geq -\tau\big(\langle g(w_t), u_t\rangle_F\big)_+, \qquad \tau \in [0, 1). \tag{97}$$

Equivalently, writing $\delta_t = \lambda_t\psi_t$ with $\psi_t := \delta_{t,0}/\|\delta_{t,0}\|_F$ and $\lambda_t := \|\delta_t\|_F$, the budget becomes $\lambda_t\langle g(w_t), \psi_t\rangle_F \geq -\tau(\langle g(w_t), u_t\rangle_F)_+$.

**Lemma D.7.** *Let $g \neq 0$ and $\delta_0$ be given, and define $T := -\tau(\langle g, u\rangle_F)_+$ with $\tau \in [0, 1)$. Among all scalings $\alpha\delta_0$ with $\alpha \geq 0$ that satisfy $\langle g, \alpha\delta_0\rangle_F \geq T$, the piecewise rule Eq. (19) chooses the largest feasible $\alpha$ and guarantees $\langle g, \delta\rangle_F \geq T$.*

*Proof.* If $\langle g, \delta_0\rangle_F \geq T$, choosing $\alpha = 1$ is feasible and maximal. Otherwise $\langle g, \delta_0\rangle_F < T \leq 0$ implies $\langle g, \delta_0\rangle_F < 0$, and the choice $\alpha = T/\langle g, \delta_0\rangle_F \in (0, 1)$ yields equality $\langle g, \alpha\delta_0\rangle_F = T$. Any feasible $\alpha$ must satisfy $\alpha \leq T/\langle g, \delta_0\rangle_F$ when $\langle g, \delta_0\rangle_F < 0$, so this $\alpha$ is maximal. $\qquad\square$

Because $H_{\text{Adam}}$ depends on $w$ only through $L(w)$, perturbing only $\dot{w}_t$ adds a single inner-product term to $\dot{H}$.

**Theorem D.8.** *Consider the perturbed dynamics Eq.* (96) *and the Adam Hamiltonian Eq.* (94)*. Along any trajectory,*

$$\frac{d}{dt}H_{\text{Adam}}(w_t, m_t, v_t) = \frac{d}{dt}H_{\text{Adam}}^{(\text{Adam})}(w_t, m_t, v_t) - \langle g(w_t), \delta_t\rangle_F \tag{98}$$

$$\leq -\Delta_{\text{Adam}}(w_t, m_t, v_t) + \tau\big(\langle g(w_t), u_t\rangle_F\big)_+, \tag{99}$$

*where $\Delta_{\text{Adam}}$ is the nonnegative dissipation term in Eq.* (95)*. In particular:*

- **(Conflict-free case, $\tau = 0$).** If $\beta_1 \geq \beta_2/4$ and $\tau = 0$, then $H_{\text{Adam}}(w_t, m_t, v_t)$ is monotonically non-increasing.

- **(Budgeted case, $\tau > 0$).** If $\beta_1 \geq \beta_2/4$, then any non-monotonicity is controlled by the budget term:

$$H_{\text{Adam}}(t) + \int_0^t \Delta_{\text{Adam}}(s)\, ds \;\leq\; H_{\text{Adam}}(0) + \tau \int_0^t \big(\langle g(w_s), u_s \rangle_F\big)_+ ds. \tag{100}$$

*Proof.* Differentiate $H_{\text{Adam}}(w_t, m_t, v_t) = L(w_t) + \frac{1}{2\beta_1}\langle m_t, D(v_t)m_t \rangle$:

$$\frac{d}{dt} H_{\text{Adam}} = \langle g(w_t), \dot{w}_t \rangle_F + \frac{d}{dt}\Big[\frac{1}{2\beta_1}\langle m_t, D(v_t)m_t \rangle\Big].$$

Under Adam, $\dot{w}_t = -u_t$ and Proposition D.6 gives $\frac{d}{dt} H_{\text{Adam}}^{(\text{Adam})} = -\Delta_{\text{Adam}}$. Under AdamO, $\dot{w}_t = -u_t - \delta_t$, while the $(m_t, v_t)$ dynamics are unchanged. Hence the only change in $\dot{H}$ is the additional term $\langle g(w_t), -\delta_t \rangle_F$, proving Eq. (98). Finally, applying the budget constraint Eq. (97) yields $-\langle g(w_t), \delta_t \rangle_F \leq \tau(\langle g(w_t), u_t \rangle_F)_+$ and thus (99). Integrating Eq. (99) over $[0, t]$ gives Eq. (100). $\qquad\square$

Proposition D.6 shows that Adam admits a dissipative Hamiltonian: the "kinetic energy" term $\frac{1}{2\beta_1}\langle m, D(v)m \rangle$ is drained at rate $\Delta_{\text{Adam}}$ when $\beta_1 \geq \beta_2/4$. Theorem D.8 makes explicit how AdamO's orthogonality drift interfaces with this structure: it contributes a single inner-product term $-\langle g, \delta \rangle_F$ to $\dot{H}$. In the conflict-free regime ($\tau = 0$), the drift can only *help* decrease the Hamiltonian and thus preserves Adam's Hamiltonian stability. In the budgeted regime ($\tau > 0$), AdamO intentionally allows controlled misalignment to prioritize constraint satisfaction; the price is precisely quantified by the additive term $\tau(\langle g, u \rangle_F)_+$ in Eq. (99). Consequently, monotone decrease of $H_{\text{Adam}}$ cannot be guaranteed universally, but any potential violation is explicitly bounded by the total "borrowed" task-descent budget.

### D.5. Choosing $\kappa$

The bounds Eq. (90) and Eq. (91) can be turned into explicit *one-step* admissible ranges for $\kappa$. There is no universal problem-independent constant upper bound on $\kappa$: the admissible range depends on the local smoothness $\mu$, the stepsize $\eta$, and (in conflict-free mode) the alignment between the orth drift and the task gradient. When $\Delta_t \neq 0$, define the (cosine) alignment

$$c_t \;\triangleq\; \frac{\langle g_t, \Delta_t \rangle}{\|g_t\|\,\|\Delta_t\|} \in [-1, 1]. \tag{101}$$

In conflict-free mode ($\tau = 0$), the per-block constraint implies $\langle g_t, \Delta_t \rangle \geq 0$, hence $c_t \geq 0$ whenever $\Delta_t \neq 0$. In budgeted mode ($\tau > 0$), recall the aggregate positive-descent term appearing in Eq. (91):

$$S_t \;\triangleq\; \sum_b \big(\langle g_{t,b}, u_{t,b} \rangle_F\big)_+ \;\geq\; 0. \tag{102}$$

**Corollary D.9.** *Assume $L_t$ is $\mu$-smooth and $\tau = 0$. If $\Delta_t = 0$, AdamO coincides with Adam at step $t$. Otherwise, a sufficient condition for $L_t(\omega_{t+1}^O) \leq L_t(\omega_{t+1}^A)$ is*

$$\eta \;\leq\; \frac{2\,\|g_t\|\,c_t}{\mu\,(2+\kappa)\,\|u_t\|}. \tag{103}$$

*Equivalently, for a fixed stepsize $\eta > 0$, any*

$$0 \leq \kappa \;\leq\; \kappa_{\max}^{(0)}(t) \;\triangleq\; \frac{2\,\|g_t\|\,c_t}{\mu\,\eta\,\|u_t\|} - 2 \tag{104}$$

*is sufficient at step $t$ (provided $\kappa_{\max}^{(0)}(t) > 0$).*

*Proof.* Start from Eq. (90) in Theorem D.5(a):

$$\eta \leq \frac{2\langle g_t, \Delta_t \rangle}{\mu\,\|\Delta_t\|\,(2\|u_t\| + \|\Delta_t\|)}.$$

If $\Delta_t = 0$ the statement is trivial. Otherwise, use the definition Eq. (101), $\langle g_t, \Delta_t \rangle = \|g_t\| \|\Delta_t\| c_t$, to obtain

$$\eta \ \leq \ \frac{2 \|g_t\| c_t}{\mu \left( 2\|u_t\| + \|\Delta_t\| \right)}. \tag{105}$$

Now apply Lemma D.3 to upper bound $\|\Delta_t\| \leq \kappa \|u_t\|$, which yields the sufficient condition

$$\eta \leq \frac{2 \|g_t\| c_t}{\mu \left( 2 + \kappa \right) \|u_t\|},$$

i.e., Eq. (103). Rearranging Eq. (103) for $\kappa$ gives Eq. (104). $\qquad\square$

**Corollary D.10.** *Assume $L_t$ is $\mu$-smooth and $\tau > 0$. Fix any allowable curvature overhead $\varepsilon_t \geq 0$. If $\kappa$ satisfies*

$$\kappa + \frac{\kappa^2}{2} \ \leq \ \frac{\varepsilon_t}{\mu \, \eta^2 \, \|u_t\|^2}, \tag{106}$$

*equivalently*

$$0 \leq \kappa \ \leq \ \kappa_{\max}^{(\varepsilon)}(t) \ \triangleq \ \sqrt{1 + \frac{2\varepsilon_t}{\mu \, \eta^2 \, \|u_t\|^2}} \ - \ 1, \tag{107}$$

*then the one-step degradation obeys the tightened guarantee*

$$L_t(\omega_{t+1}^O) - L_t(\omega_{t+1}^A) \leq \eta\tau S_t + \varepsilon_t. \tag{108}$$

*In particular, choosing $\varepsilon_t = \alpha \, \eta\tau S_t$ for any $\alpha \in (0,1]$ yields the explicit sufficient range*

$$0 \leq \kappa \ \leq \ \sqrt{1 + \frac{2\alpha \, \tau \, S_t}{\mu \, \eta \, \|u_t\|^2}} \ - \ 1. \tag{109}$$

*Proof.* Start from Eq. (91) in Theorem D.5(b):

$$L_t(\omega_{t+1}^O) - L_t(\omega_{t+1}^A) \leq \eta\tau S_t + \mu\eta^2 \|u_t\|^2 \left( \kappa + \frac{\kappa^2}{2} \right).$$

If Eq. (106) holds, then the quadratic curvature term is bounded by $\varepsilon_t$, giving

$$L_t(\omega_{t+1}^O) - L_t(\omega_{t+1}^A) \leq \eta\tau S_t + \varepsilon_t,$$

i.e., Eq. (108). Solving Eq. (106) for $\kappa \geq 0$ yields the closed form Eq. (107). Finally, substituting $\varepsilon_t = \alpha \, \eta\tau S_t$ into Eq. (107) gives Eq. (109). $\qquad\square$

*Remark* D.11. The preceding corollaries turn the one-step guarantees into concrete $\kappa$ ranges. In conflict-free mode ($\tau = 0$), a non-inferiority guarantee from Eq. (90) is ensured whenever $\kappa \leq \kappa_{\max}^{(0)}(t)$ in Eq. (104); this bound becomes tight when the alignment $c_t$ is small, and it shrinks with larger $\mu$ or larger $\eta$. In budgeted mode ($\tau > 0$), Eq. (91) isolates the *curvature-induced* overhead as $\mu\eta^2 \|u_t\|^2(\kappa + \kappa^2/2)$, so choosing $\kappa \leq \kappa_{\max}^{(\varepsilon)}(t)$ in Eq. (107) (or the relative form Eq. (109)) guarantees that the extra harm attributable to $\kappa$ is at most a user-chosen budget. Overall, admissible $\kappa$ must decrease as either the stepsize $\eta$ or the local curvature $\mu$ increases, and it must be particularly conservative when the orth drift has weak task alignment (small $c_t$) or when $\|u_t\|$ is large.

