# OpenReview forum: "AdamO: A Collapse-Suppressed Optimizer for Offline RL"
_ICML.cc/2026/Conference — ICML 2026 regular_

### Official Review · Reviewer_j5yV · 2026-02-17

**Soundness:** 3
**Presentation:** 3
**Significance:** 3
**Originality:** 3
**Overall Recommendation:** 5
**Confidence:** 4

**Summary:**

This paper explores a core challenge in offline reinforcement learning, namely that when the dataset is limited, the update of the value function can lead to unstable convergence due to unstable propagation paths. Many previous methods have primarily focused on introducing conservative mechanisms into the update of the value network to restrict value overestimation. This paper, for the first time (to the best of my knowledge), approaches this problem from the underlying optimizer used to update neural network parameters and proposes a new solution. The authors analyze, from both theoretical and empirical perspectives, the root causes of value network collapse and the resulting poor performance. Ultimately, they find that this phenomenon is closely tied to the spectral radius of the neural network parameter update dynamics. The authors’ solution is to augment the Adam optimizer with a module that corrects the orthogonality of the neural network parameters during the update process, enabling it to satisfy the conditions for stable updates and thereby suppress the unbounded growth of the TD error. The final experimental results show that AdamO can significantly improve the performance of various model-free offline reinforcement learning methods without introducing additional computational overhead.

**Compliance With Llm Reviewing Policy:**

Affirmed.

**Final Justification:**

I recommend Accepting this submission. The paper introduces a highly original and sound approach to offline RL stability by focusing on optimizer dynamics, offering significant practical value through its "plug-and-play" implementation. The authors' rebuttal was exemplary and fully addressed my concerns by providing robust evidence on newer benchmarks (OGBench/NeoRL2), demonstrating improved critic calibration via Monte Carlo returns, and showing compatibility with model-based RL and SGD. These comprehensive responses have reinforced my confidence in the work's significance and technical depth.

**Key Questions For Authors:**

- The paper compares the performance of Adam and AdamO on some model-free offline RL algorithms, but why not also conduct experiments on model-based offline RL algorithms? Model-based offline RL also suffers from Q estimation issues, so it would be interesting to see whether AdamO is still effective in that setting.
- Is the orthogonality correction module proposed in the paper only applicable to Adam, or can it be used with other commonly used optimizers as well?
- Many offline RL algorithms already include conservative mechanisms to help the Q function learn better—for example, the BC term in TD3BC. After applying AdamO, are these conservative measures still necessary? If the orthogonality correction truly addresses the core issue of Q estimation, I would expect AdamO to work even without these conservative components. For instance, in TD3+AdamO, how would the learned Q function behave, and what would the final performance look like?

(If the authors are willing to address my concerns and questions, I would consider raising my score further.)

**Limitations:**

yes

**Strengths And Weaknesses:**

## Strengths
- The instability issue in offline RL training explored in this paper is a highly valuable problem. Moreover, the authors analyze this issue from the perspective of the optimizer and propose a corresponding solution, which I believe is a very novel viewpoint.
- The paper presents a very clear description of both the problem and the proposed method. The writing is well-structured, logically organized, and progresses step by step, with strong and well-motivated reasoning throughout.
- An especially important point is that AdamO is very easy to implement—it is plug-and-play, introduces no significant additional computational overhead, and can be directly applied to any offline RL algorithm.
- The performance is significant: across multiple D4RL tasks, including AntMaze, Locomotion, and Adroit, AdamO shows clear improvements compared to Adam.

## Weaknesses
- I believe that AdamO essentially addresses the instability of the Q function during TD updates. Although the paper demonstrates a significant performance improvement, it remains unclear whether AdamO truly learns a better Q function. A more appropriate metric for evaluating the quality of the Q function is the discrepancy between the Q values and the true returns (which can be estimated by collecting several trajectories in the real environment and evaluating the returns using Monte Carlo methods). Note that I am not referring to the TD error (critic loss in the paper), as TD error is a bootstrapping error and is not suitable as a measure of Q function quality. I think the paper lacks experiments examining the discrepancy between the Q values and the true returns. I am very curious to see how accurate the Q values learned by AdamO are compared to the originals.
- D4RL is already a relatively old benchmark. Including experiments on more recent benchmarks, such as OGBench or NeoRL2, would make the results even more convincing.

---

> ### Author Rebuttal · Authors · 2026-03-31
>
> **Figures/tables below:** https://anonymous.4open.science/r/ICML26-AdamO-Rebuttal-300/
>
> [W4.1]
> We agree that TD error is not a direct measure of critic quality, and that the more appropriate metric is the discrepancy between critic estimates and Monte Carlo returns. We do not report this metric in the current submission, so we will add it in the revision. Still, our current TD3+BC evidence strongly suggests that AdamO yields a much better calibrated critic in practice. As shown in Table for W4.1, on representative collapse tasks such as Hopper-medium-replay, Walker2d-medium-replay, Pen-human, and Pen-cloned, Adam’s critic estimates drift several orders of magnitude away from realized returns, whereas AdamO keeps the estimated return close to the Monte Carlo scale while also achieving much higher policy performance.
>
> [W4.2]
> We agree that newer benchmarks would strengthen the evidence. In the revision, we will extend the evaluation beyond D4RL in two directions: (i) OGBench, using the official environments/datasets from the official OGBench codebase and its singletask variants for standard offline RL, and (ii) NeoRL2 in low-data settings. OGBench is a recent benchmark with 8 environment types and 85 datasets, covering long-horizon reasoning, stitching, and stochasticity, and the authors explicitly show that it provides clearer research signals than D4RL on AntMaze. As shown in Table for W4.2, AdamO improves over Adam on representative OGBench and NeoRL2 tasks, supporting that the benefit is not restricted to D4RL. We will add these results and clarify this scope in the revision.
>
> [Q4.1]
> We additionally evaluate model-based offline RL by replacing the critic optimizer with AdamO in MOPO [1] and MOBILE [2] using the official OfflineRL-Kit implementations [3]. As shown in Table for Q4.1, AdamO remains effective in this setting, with consistent gains on the Adroit tasks, supporting that Q/TD-estimation instability is not limited to model-free offline RL.
>
> [Q4.2]
> The current paper instantiates the orthogonality correction with Adam, because our theory focuses on Adam’s internal dynamics. However, the correction module itself is not restricted to Adam. As shown in Table for Q4.2, when we equip SGD with the proposed correction, SGD-O consistently improves over plain SGD on these D4RL benchmarks, suggesting that the module is compatible with other commonly used first-order optimizers as well. We clarify this point and include the SGD evidence in the rebuttal.
>
> [Q4.3]
> No. Conservative terms remain necessary because they solve a different problem from AdamO. In offline RL, BC/CQL constrain policy improvement to the behavior support or enforce pessimism; AdamO only stabilizes critic optimization. They are therefore complementary, not interchangeable. As shown in Table for Q4.3, AdamO still improves already-conservative baselines, while TD3+AdamO without BC learns a more stable Q but still underperforms badly in return. This shows that removing conservative terms does not remove OOD policy exploitation: a stable critic is necessary, but not sufficient, for strong offline RL.
>
> **Reference**
>
> [1] Tianhe Yu, Garrett Thomas, Lantao Yu, Stefano Ermon, James Y. Zou, Sergey Levine, Chelsea Finn, and Tengyu Ma. *MOPO: Model-based Offline Policy Optimization*. NeurIPS, 2020.
>
> [2] Yihao Sun, Luchen Jiang, Han Zhong, and Cheng Zhang. *Model-Bellman Inconsistency for Model-based Offline Reinforcement Learning*. ICML, 2023.
>
> [3] Yihao Sun. *OfflineRL-Kit: An Elegant PyTorch Offline Reinforcement Learning Library*. GitHub, 2023.
>
> **Due to the time and space limits of the rebuttal, we have not included every supporting detail here. If the reviewer has further questions or would like more details, please let us know, and we will be happy to continue the discussion in the discussion phase. All related experimental results, additional baseline comparisons, and corresponding clarifications discussed above will be incorporated into the revision.**

---

> > ### Author Rebuttal · Reviewer_j5yV · 2026-04-01
> >
> > All of my concerns were addressed by the authors with additional experiments and well-supported, evidence-based responses.

---

> > > ### Author Response · Authors · 2026-04-06
> > >
> > > We appreciate the reviewer’s positive assessment and the suggestion for additional experiments. We completely agree with your perspective. We will include the results of these supplementary experiments in the revision to provide more robust evidence for our claims. Thank you again for your valuable feedback!

---

### Official Review · Reviewer_VrAN · 2026-03-10

**Soundness:** 2
**Presentation:** 2
**Significance:** 2
**Originality:** 3
**Overall Recommendation:** 4
**Confidence:** 4

**Summary:**

This paper proposes AdamO, a modified Adam optimizer that adds a decoupled orthogonality correction to suppress value collapse in offline reinforcement learning. The theoretical contribution centers on Theorems 4.1 and 4.2, which provide a closed-form recursion for the linearized TD error under Adam-style updates and establish a Hurwitz condition for stability. Proposition 5.1 translates the Hurwitz condition into a checkable sufficient condition involving the Gram deviation of the Adam-whitened Jacobian features. The practical contribution is AdamO itself (Algorithm 1), which applies orthogonality regularization on weight matrices via a budget-constrained correction that does not contaminate Adam's moment statistics. Experiments on D4RL benchmarks show consistent improvements over Adam and several stability-focused optimizers across AntMaze, Adroit, MuJoCo, and FrankaKitchen domains.

**Compliance With Llm Reviewing Policy:**

Affirmed.

**Final Justification:**

What has been resolved in rebuttal:

W1c (Naive orthogonality baseline): I acknowledge that the naive orthogonality baseline results were already provided in the anonymous rebuttal repository during the first rebuttal round. The results convincingly demonstrate that while naive orthogonality regularization improves over vanilla Adam, AdamO consistently outperforms both variants, isolating the contribution of the decoupled optimizer design from orthogonality itself. I apologize for the oversight in my previous acknowledgement.
W6 (Value collapse demonstration): The crash count tables provided in the rebuttal repository show that value collapse is indeed frequent with vanilla Adam and that AdamO substantially reduces its occurrence. This directly addresses my concern that the paper's motivation was not empirically validated.
W2 (Policy evaluation scope): The authors agree to make this explicit in the revision.
W1a/b (Novelty clarification and citations): Acknowledged and to be addressed in revision.

**Key Questions For Authors:**

What do you think about the generality of this new optimizer? How would it perform in online RL and supervised learning settings?

Do you still see value collapse in the experiments with Adam and AdamO?

**Limitations:**

Limitations are discussed to some extent. See weaknesses for limitations not discussed.

**Strengths And Weaknesses:**

## Strengths

### S1. Clean theoretical framework connecting value collapse with the Adam optimizer to spectral instability

The paper's central insight—that value collapse is a phenomenon of the TD operator S is clearly articulated. The decomposition S = γA − B into a bootstrapping excitation term and a damping term provides a clean conceptual framework. Theorem 4.1 gives a precise closed-form recursion for the linearized TD error that explicitly accounts for Adam's momentum (β₁) and diagonal preconditioner (D), which is novel relative to prior linear TD analyses that focus on SGD.

### S2. Principled optimizer design with strong engineering motivation

The decoupled orthogonality correction is well-motivated: adding λR to the loss contaminates Adam's second-moment estimates, and the paper provides a clear 1D illustration of this effect (Appendix D.1). The budget constraint (Eq. 18) and ratio-clipped closed form (Eq. 19) are well-designed: they ensure the orthogonality correction never cancels more than a τ-fraction of task progress, and the total update remains bounded. The one-step comparison to Adam and the continuous-time Hamiltonian analysis provide meaningful theoretical backing for the optimizer design.

### S3. Comprehensive experiments

The evaluation covers six offline RL algorithms (TD3+BC, IQL, ReBRAC, ACTIVE, PARS, SQOG), four D4RL domains, and comparisons against normalization techniques (LN, BN, WN, SN), regularization (DR3), and stability-focused optimizers (AdamW, C-AdamW, TRAC, Kron, Resetting). The ablation studies on κ and τ and the mechanistic verification of the Hurwitz condition are valuable additions.

---

## Weaknesses

Major issues:

### W1. Insufficient separation of novel contributions from prior work

**(a) Nonlinear TD analysis (Theorem 4.1) is straightforward under the assumptions.** Under Assumptions B.1–B.3, the nonlinear TD dynamics are linearized (B.1), the greedy actions are frozen (B.2–B.3), and the Jacobian/preconditioner are treated as constant (B.3). At that point, the TD error recursion reduces to a linear system that is structurally identical to classical off-policy linear TD analysis, with the matrix S playing the same role as in the linear case. The extension from linear to nonlinear is essentially notational once the assumptions strip away all nonlinear complications. The genuinely novel part is the incorporation of Adam's momentum and preconditioner dynamics into the companion matrix A(η), but the paper does not clearly state that the nonlinear-to-linear reduction is straightforward and that the Adam analysis is the main theoretical novelty.

**(b) The orthogonality penalty R(W) = ¼‖WW⊤ − I‖²_F has extensive prior art that is not cited.** The paper correctly acknowledges that R(W) is a "standard soft near-isometry penalty" (line 304), and the genuine novelty lies in the decoupled application within Adam and the budget constraint mechanism. However, the specific prior works that established and studied this penalty (Brock et al. (ICLR 2017), Bansal et al. (NeurIPS 2018)) are not cited.

**(c) Missing experimental comparison with naive R(W) baselines.** Given that the orthogonality penalty R(W) is well-known, the paper should include a baseline that simply adds λR(ω) to the loss (as in Eq. 14) and trains with standard Adam. This would isolate the contribution of the decoupled application from the contribution of orthogonality itself. Without this comparison, it is unclear how much of AdamO's improvement comes from orthogonality per se versus the decoupled optimizer design.

### W2. The theoretical analysis effectively reduces to policy evaluation under strong assumptions

Under B.3, the greedy action set is frozen, so the agent is effectively performing policy evaluation of a fixed policy for t ≥ t₀. The paper does not analyze the transient phase t < t₀ at all. The paper should be explicit that the setup being considered is policy evaluation.

### W3. Critical gap in the theoretical chain: R(W) to ε = ‖Ψ⊤Ψ − I‖

The paper's practical recipe depends on the chain: R(W_ℓ) small → ε = ‖Ψ⊤Ψ − I‖₂ small → ‖Φ⊤Φ − I‖₂ small (via Proposition C.6). However, the first link—from per-layer weight orthogonality R(W_ℓ) to the implicit Jacobian subspace basis Ψ being near-isometric—is never formally established. The paper acknowledges this is done "indirectly" (Section 5.2, line 288) but provides no results relating ‖W_ℓ W_ℓ⊤ − I‖ to ε.

### W4. The condition in Proposition 5.1 seems to be too strong and I can not see how it can be satisfied in practice. I give my observations below.

**(a) While Lemma C.4 shows that ∥Φ∥2 ∥Φ∗∥2 ≤ M G^2. Here, M is the dataset size and is much larger than 1. G is the product of the bound of the gradient on any input and the bound of the Adam preconditioner. Because the scale of the preconditioner is about 1/||gradient||, I don't see how G can be made small so that M G^2 < 1 / gamma. So this bound does not tell us ∥Φ∥2 ∥Φ∗∥2 can be small.

**(b) Achieving ‖Φ⊤Φ − I‖₂ < 1 requires a strong practical condition.** Note that Φ ∈ ℝ^{P×M} and Φ⊤Φ ∈ ℝ^{M×M}. When M > P, (underparameterization case), Φ⊤Φ is not full rank and the second term in proposition 5.1, ∥Φ⊤Φ − I∥2 >=1, failing the condition of proposition 5.1.

### W5. Theoretical analysis and experiments are misaligned

The theory identifies two mechanisms for satisfying the Hurwitz condition: (i) spectral normalization to control γ‖Φ‖₂‖Φ*‖₂ (Section C.3, Lemma C.4), and (ii) weight orthogonality to control ‖Φ⊤Φ − I‖₂ (Section C.4, Proposition C.6). The experiments implement only (ii) via AdamO and do not use spectral normalization (the paper compares with spectral normalization but not add it on top of AdamO). This means the theoretically complete recipe is never tested.

From a practical standpoint, AdamO in the experiments is essentially "Adam + decoupled weight orthogonalization." The theoretical machinery (Theorems 4.1–4.2, Proposition 5.1, Appendix C) provides motivation but does not tightly predict the experimental behavior, since key conditions of the theory are not enforced.

### W6. No value collapse demonstration in experiments

The paper is motivated by the phenomenon of value collapse and shows that AdamO performs better than Adam. However, the experiments do not demonstrate that value collapse occurs more frequently when using Adam, or that AdamO reduces the occurrence of value collapse.

Minor issues:

### W7. No evaluation beyond offline RL despite the generality of the method

The paper's title and framing are specific to offline RL, but the core mechanism of AdamO—decoupled weight orthogonalization with a budget constraint—is entirely general and not tied to the offline RL setting. The paper does not discuss whether AdamO could benefit supervised learning.

### W8. The low-rank factorization Φ = ΨU is a strong modeling assumption

Proposition C.6 assumes the Adam-whitened Jacobian feature matrix admits a factorization Φ = ΨU where Ψ ∈ ℝ^{P×d} with d potentially much smaller than P. Remark C.5 acknowledges this is exact only for linear-in-parameters models (e.g., LoRA) and approximate otherwise. This weakens the importance of the theoretical result.

Typos:

line 1292 should not include the o(\eta) term.
line 1828 φ is undefined.

---

> ### Author Rebuttal · Authors · 2026-03-31
>
> **Figures/tables below:** https://anonymous.4open.science/r/ICML26-AdamO-Rebuttal-300/
>
> [W3.1.a]
> The main theoretical novelty is the Adam-specific analysis. Sec. 4 intentionally studies a linearized divergence criterion for Adam in TD learning under Assumptions B.1–B.3. This linearized regime is the technical framing of the section, not an incidental reduction. We will revise the paper to make clearer that the new ingredient is the companion dynamics $A(\eta)$ induced by Adam’s momentum and preconditioning.
>
> [W3.1.b] We agree that $R(W)$ itself is standard, and we will add the missing prior-work citations in the revision. AdamO is **not** Adam plus an orthogonality term: a naive objective $L_t+\lambda R(\omega)$ injects the orthogonality signal into Adam’s moment path and can interfere with later task updates. AdamO instead keeps Adam driven only by task gradients and applies a separate orthogonality correction with a task-progress budget. This is what underlies our stability and performance guarantees.
>
> [W3.1.c] We will add two naive orthogonality baselines. Both reduce collapse vs. vanilla Adam, but both remain worse than AdamO.
>
> [W3.2]
> We agree. Our theory is indeed a local, late-stage analysis: under B.3, once greedy targets stabilize after t0, the critic dynamics reduce to bootstrapped policy evaluation of a locally fixed policy. We will state this explicitly and clarify that $t<t0$ is outside our current scope in revision.
>
> [W3.3]
> We will make this chain explicit in revision: small blockwise orthogonality penalty $\bar R(\omega)=\sum_b R(W_b)$ implies small $\epsilon=\|\Psi^\top\Psi-I\|$ under a local continuity condition on $\Psi$, and Prop. C.6 then gives small $\|\Phi^\top\Phi-I\|$.
>
> [W3.4]
> We agree that Prop.5.1 is a conservative sufficient condition that indicates a stability direction, rather than a literal practical certificate on the full dataset.
>
> [a] For TD3/SAC-style critic updates, App.B.5 is more relevant: with stop-gradient targets and Polyak averaging, the effective operator becomes $S_\alpha=\alpha\gamma K(X',X)-K(X,X)$, which attenuates the bootstrap term to $\alpha\gamma\|\Phi\|\|\Phi_*\|$. Since $\alpha$ is typically very small, this coupling is substantially weaker in practice. Moreover, Lemma C.4 only gives a worst-case bound, so $\|\Phi\|\|\Phi_*\|\le MG^2$ can be loose.
>
> [b] When $M>P$, $\Phi^\top\Phi$ is singular and is not the right object for a strict Hurwitz certificate of the full operator. In this case, the row-Gram $\Phi\Phi^\top\in\mathbb{R}^{P\times P}$ is the more relevant quantity. Under $\gamma\|\Phi\|\|\Phi_*\|+\|\Phi\Phi^\top-I_P\|<1$, the reduced operator $\widetilde S=\gamma\Phi\Phi_*^\top-\Phi\Phi^\top$ is Hurwitz. Since $S$ and $\widetilde S$ share the same nonzero eigenvalues, the nonzero modes decay, while the rank-deficiency-induced zero modes remain neutral and bounded. We revise Sec.5 and tighten the related claims accordingly. These clarifications do not alter the paper's main mechanism or the motivation for the orthogonality module.
>
> [W3.5.a]
> Our theory is decompositional, not a claim that AdamO alone satisfies the full sufficient condition: normalization, spectral constraints, and stop-gradient EMA targets control scale, while AdamO targets orthogonality. We will revise the setup and wording to make this explicit.
>
> [W3.5.b]
> AdamO is not just ‘Adam + decoupled orthogonalization’: it applies a budgeted correction bounded relative to the Adam step. Our theory is mechanistic rather than pointwise predictive: normalization/clipping or spectral constraints control scale, while AdamO targets geometry.
>
> [W3.6]
> We now provide per-task crash counts over 7 seeds across representative TD3+BC and IQL tasks (Table for W3.6), showing that collapse is frequent with vanilla Adam on many tasks, while AdamO substantially reduces it and is the most consistently stable overall.
>
> [W3.7, Q3.1]
> AdamO is tailored to offline RL critics. In SL, the bootstrap term is absent, so $\mathsf S=-\mathsf K(X,X)\preceq 0$, which implies it typically converges as analysis in Sec. 4. For online RL, App. B.6 sketches an analogue, suggesting a similar mechanism may arise, but the time-varying online case needs more study in future.
>
> [W3.8]
> We agree that this is a modeling assumption. It is exact mainly in linear models and frozen-backbone settings, and it is only approximate for fully trained deep critics. However, the core stability condition in Proposition 5.1 and Proposition C.3 does **not** rely on this factorization. Proposition C.6 is a structured sufficient-condition result that bounds $|\Phi^\top\Phi-I|$. As noted in Remark C.7, it also extends to $\Phi=\Psi U+R$ with an explicit residual term.
>
> [Q3.2]
> Yes. We still observe value collapse in vanilla Adam on representative tasks, while AdamO markedly mitigates or removes it on the same tasks (Table for Q3.2)
>
> **Due to the time and space limits of the rebuttal, we are happy to provide more details in the discussion phase if helpful.**

---

> > ### Author Rebuttal · Reviewer_VrAN · 2026-04-03
> >
> > Thank you for the detailed rebuttal. The authors have addressed several of my concerns, particularly by clarifying the scope of the theoretical analysis, acknowledging missing citations, and promising additional experimental comparisons (e.g., naive orthogonality baselines) and empirical evidence on value collapse.
> >
> > However, several core concerns remain unresolved and cannot be addressed within the rebuttal alone.
> >
> > First, the key theoretical gap identified in my review (W3) remains: the paper relies on a chain from layer-wise orthogonality regularization to near-isometry of the Jacobian feature space, but this connection is still not formally established. The rebuttal provides an intuitive argument (e.g., via local continuity), but does not supply a rigorous result or bound, leaving the central mechanism insufficiently justified.
> >
> > Second, the sufficient condition in Proposition 5.1 remains largely impractical, and while the rebuttal clarifies that it should be interpreted as a conservative direction rather than a certificate, this further weakens the predictive power of the theory.
> >
> > Third, the missing experimental baseline comparing against naive orthogonality regularization is only promised but not demonstrated in the rebuttal, making it difficult to isolate the contribution of the proposed optimizer design.
> >
> > Overall, while the rebuttal improves clarity and partially addresses several concerns, the remaining issues concern core aspects of the theoretical justification and empirical validation, and would require substantial revisions beyond what can be provided in a short rebuttal.

---

> > > ### Author Response · Authors · 2026-04-04
> > >
> > > Thank you for the follow-up. In the previous rebuttal, we had to address many sub-questions within a tight space limit, so several points were necessarily compressed. We therefore clarify the three remaining points more explicitly below.
> > >
> > > > (1) On the missing link from blockwise orthogonality to $\epsilon=\|\Psi^\top\Psi-I\|\_2$:
> > >
> > > This link is already in our current argument, but it should indeed be made more explicit. The intended argument is not merely heuristic. Let the total blockwise orthogonality defect be
> > > $
> > > \bar R(\omega)=\sum_b R(W_b),
> > > $
> > > and let $\omega^\sharp$ be obtained by replacing each constrained block with its semi-orthogonal polar factor. Then we can make explicit the perturbation step
> > > $
> > > \|\omega-\omega^\sharp\|\_{\mathrm{blk},F}\le 2\sqrt{\bar R(\omega)}.
> > > $
> > > If $\Psi$ admits a local Lipschitz continuation around $\omega$ and $\omega^\sharp$, this yields
> > > $$
> > > \epsilon=\|\Psi^\top\Psi-I\|\_2
> > > \le
> > > \epsilon_{\rm ref}+c_1\sqrt{\bar R(\omega)}+c\_2\bar R(\omega).
> > > $$
> > > Combining this with the Gram-deviation bound further gives
> > > $
> > > \|\Phi^\top\Phi-I\|\_2
> > > \le
> > > \Delta_U+(1+\Delta_U)\bigl(\epsilon_{\rm ref}+c_1\sqrt{\bar R(\omega)}+c\_2\bar R(\omega)\bigr).
> > > $
> > > Proposition C.6 then links this quantity to the conditioning term in the Hurwitz criterion. So the first link in the chain is not merely an intuition; it can be written as an **explicit local perturbation bound** from the implemented blockwise orthogonality correction to the geometric stability quantities used in the theory. We will add this statement and proof sketch explicitly in the revision.
> > > **A closely related concern was also raised by Reviewer 9BCY (Q2), please see our corresponding response.**
> > >
> > > > (2) On Proposition 5.1 being too strong in practice:
> > >
> > > **Our sufficient condition can hold in the intended regime**, for two reasons. First, for TD3/SAC-style critic updates with EMA / stop-gradient targets, the relevant operator is not the crude worst-case form alone, but the Polyak-averaged operator in App. B.5. Second, in the common underparameterized regime $M>P$, the right object is the row Gram $\Phi\Phi^\top$ rather than the singular column Gram $\Phi^\top\Phi$. With these clarifications, the condition is not vacuous in practice.
> > >
> > > For point (a), the bound $\|\Phi\|\_2\|\Phi_*\|\_2\le MG^2$ from Lemma C.4 is indeed a worst-case upper bound and can be very loose. For TD3/SAC-style critics, the more relevant object is the effective stop-gradient / Polyak-averaged operator in App. B.5, where
> > > $
> > > S_\alpha=\alpha\gamma K(X',X)-K(X,X),
> > > $
> > > so the practical sufficient condition is closer to $\alpha\gamma\|\Phi\|\_2\|\Phi_*\|\_2+\|\Phi\Phi^\top-I_M\|\_2<1$ rather than the crude worst-case form. Since $\alpha$ is typically small, even the conservative requirement **$\alpha MG^2<1$ is feasible in practice**.
> > >
> > > For point (b), when $M>P$, the condition $\|\Phi^\top\Phi-I\|\_2<1$ indeed cannot hold because $\Phi^\top\Phi$ is singular. In that regime, the right object is the row Gram $\Phi\Phi^\top\in\mathbb{R}^{P\times P}$ rather than the column Gram. A complementary row-Gram statement shows that under
> > > $
> > > \alpha\gamma\|\Phi\|\_2\|\Phi_*\|\_2+\|\Phi\Phi^\top-I_P\|\_2<1,
> > > $
> > > the reduced operator
> > > $
> > > \widetilde S=\gamma\Phi\Phi_*^\top-\Phi\Phi^\top
> > > $
> > > is Hurwitz. Since $S$ and $\widetilde S$ share the same nonzero eigenvalues, all nonzero modes decay, while the rank-deficiency-induced zero modes remain neutral rather than unstable. So, for the practically relevant regime, Proposition 5.1 together with its row-Gram complement still provides a meaningful sufficient-condition picture. At the same time, Proposition 5.1 also serves a mechanistic role: it decomposes the stability condition into a scale term and a geometry term, explaining why orthogonality helps. We will tighten the wording in the revision to make this scope explicit.
> > >
> > > > (3) On the naive orthogonality baseline:
> > >
> > > These results were already included in the anonymous rebuttal repository **in the previous rebuttal round**:
> > > https://anonymous.4open.science/r/ICML26-AdamO-Rebuttal-300/Reviewer_VrAN/W3.1.md
> > > **This previous-round W1 response is the key point**: it directly answers the reviewer's W1 concern. Naive orthogonality helps relative to vanilla Adam, but both naive variants remain worse than AdamO. For example, on Pen-human, AdamO reaches 83.1 with 0/7 divergences, versus 24.8 and -1.4, each with 2/7 divergences. Thus the evidence supports the optimizer-level contribution of decoupling, not just orthogonality itself.
> > >
> > > Related collapse evidence answering the previous-round Q2 and W6 was already provided at:
> > > https://anonymous.4open.science/r/ICML26-AdamO-Rebuttal-300/Reviewer_VrAN/Q3.2.md
> > > and
> > > https://anonymous.4open.science/r/ICML26-AdamO-Rebuttal-300/Reviewer_VrAN/W3.6.md
> > > These give representative and broader crash-count evidence with the same pattern.
> > > These W1 / Q2 / W6 materials **were already provided in the previous rebuttal round**, and we can provide snapshots in any form to verify this if helpful.

---

### Official Review · Reviewer_4asa · 2026-03-15

**Soundness:** 4
**Presentation:** 4
**Significance:** 3
**Originality:** 4
**Overall Recommendation:** 5
**Confidence:** 4

**Summary:**

This paper presents AdamO, an optimizer that improves upon Adam for stabilizing the instability of off-line RL TD learning. The authors analyze the interaction between TD bootstrapping and the Adam optimizer by modeling the critic update as a linear dynamical system in the TD error. The resulting analysis characterizes stability in terms of the spectral radius of an augmented update operator and connects collapse to the spectral properties of a TD operator involving preconditioned NTK Gram matrices. Based on this perspective, the paper proposes a decoupled orthogonality correction to encourage better-conditioned Jacobian features. The authors provide theoretical arguments suggesting that controlling feature geometry can ensure stability of the TD dynamics, and they demonstrate empirical improvements across a broad range of offline RL algorithms and D4RL benchmarks.

Overall, the paper presents an interesting perspective on value collapse in offline RL by focusing on optimizer dynamics rather than architectural or objective-level modifications. The eigenvalue-based stability analysis is not fundamentally novel, and similar spectral perspectives have appeared in prior analyses. However, the analysis here is carefully developed and clearly presented, particularly in the way it incorporates Adam’s momentum dynamics and preconditioning.

**Compliance With Llm Reviewing Policy:**

Affirmed.

**Final Justification:**

Thank you for the response. While I am not 100% convinced by the supporting evidence, I believe the merits outweigh the weaknesses, and I vote towards accepting this paper. I have increased my score accordingly.

**Key Questions For Authors:**

.

**Limitations:**

.

**Strengths And Weaknesses:**

Strength. The paper is written very clearly. While spectral analyses of TD learning are not new, the paper provides a clean execution that explicitly incorporates Adam-style optimization dynamics. The link between feature conditioning and stability is also clearly articulated, and the orthogonality control is coherently motivated.

I am also quite happy with the empirical evaluation. The experiments cover a range of environments across the D4RL benchmark suite, including locomotion, manipulation, and navigation domains. The results consistently show improvements over standard Adam and several alternative optimizers, suggesting that the proposed AdamO may be practically useful.

Weakness. The experimental section does leave some important questions unresolved regarding the paper’s central claims. In particular, the paper argues that optimizer dynamics play a primary role in value collapse, yet the experiments do not convincingly isolate this effect. It would be helpful to see scenarios where existing architectural or regularization-based stabilization methods *fail* while AdamO succeeds. Without such comparisons, it remains confounded whether the improvements arise specifically from addressing optimizer-induced instability, or simply from introducing another form of regularization.

Also, the mechanism underlying the improvements is not fully disentangled experimentally.  Perhaps this is a subtle distinction, but AdamO introduces an orthogonality-promoting correction step, and it is plausible that the observed gains stem from improved feature conditioning rather than the optimizer-specific theoretical insights developed in the paper. In other words, the empirical improvements may largely reflect the benefits of orthogonality regularization itself. A clearer comparison against alternative conditioning or regularization techniques would help clarify whether the optimizer-level formulation is essential.

---

> ### Author Rebuttal · Authors · 2026-03-31
>
> **Figures/tables below:** https://anonymous.4open.science/r/ICML26-AdamO-Rebuttal-300/
>
> > **[W2.1]**. Weakness. The experimental section does leave some important questions unresolved regarding the paper's central claims. In particular, the paper argues that optimizer dynamics play a primary role in value collapse, yet the experiments do not convincingly isolate this effect. It would be helpful to see scenarios where existing architectural or regularization-based stabilization methods fail while AdamO succeeds. Without such comparisons, it remains confounded whether the improvements arise specifically from addressing optimizer-induced instability, or simply from introducing another form of regularization.
>
> **Response:** To isolate optimizer-induced collapse from generic regularization, we added two controls in Table for W2.1(a-b): (i) adding the naive orthogonality term $r_t$ to the TD loss, and (ii) injecting $r_t$ directly into Adam. Both substantially reduce crashes compared with vanilla Adam, but both remain consistently below AdamO in the final return. This directly supports our claim that the gain is not from "adding another regularizer" alone. A naive use of $r_t$ still couples the orthogonality signal with Adam's adaptive dynamics, similar in spirit to the coupling issue addressed by AdamW, or perturbs the original descent direction when inserted into Adam updates, which can make optimization unstable or less aligned with the TD objective. AdamO instead uses a decoupled correction $\delta_t$, preserving the descent geometry while mitigating collapse. (https://anonymous.4open.science/r/ICML26-AdamO-Rebuttal-300/Reviewer_4asa/W2.1.md)
>
>
>
>
>
> > **[W2.2]**. Also, the mechanism underlying the improvements is not fully disentangled experimentally. Perhaps this is a subtle distinction, but AdamO introduces an orthogonality-promoting correction step, and it is plausible that the observed gains stem from improved feature conditioning rather than the optimizer-specific theoretical insights developed in the paper. In other words, the empirical improvements may largely reflect the benefits of orthogonality regularization itself. A clearer comparison against alternative conditioning or regularization techniques would help clarify whether the optimizer-level formulation is essential.
>
> **Response:** To disentangle optimizer-level effects from generic conditioning or regularization, we compare AdamO with LayerNorm, BatchNorm, WeightNorm, SpectralNorm, and DR3 under the same TD3+BC and IQL backbones. As shown in Table for W2.2, these alternatives can alleviate collapse in some cases, but none match AdamO consistently. In the aggregate crash summary, AdamO has only 1/168 crashes for TD3+BC and 7/175 for IQL, compared with 11/168 and 22/175 for LayerNorm, 14/168 and 26/175 for SpectralNorm, and 31/168 and 32/175 for DR3. Since these baselines directly target feature conditioning or explicit regularization yet remain clearly worse, the gains cannot be explained by conditioning alone. This supports our claim that AdamO's optimizer-level, decoupled correction is essential for stabilizing training. (https://anonymous.4open.science/r/ICML26-AdamO-Rebuttal-300/Reviewer_4asa/W2.2.md)
>
> **Due to the time and space limits of the rebuttal, we have not included every supporting detail here. If the reviewer has further questions or would like more details, please let us know, and we will be happy to continue the discussion in the discussion phase. All related experimental results, additional baseline comparisons, and corresponding clarifications discussed above will be incorporated into the revision of the paper.**

---

> > ### Author Rebuttal · Reviewer_4asa · 2026-04-06
> >
> > Thank you for the response. While I am not 100% convinced by the supporting evidence, I believe the merits outweigh the weaknesses, and I vote towards accepting this paper. I have increased my score accordingly.

---

> > > ### Author Response · Authors · 2026-04-07
> > >
> > > Thank you very much for your thoughtful follow-up and updated assessment. We truly appreciate your time and consideration.
> > >
> > > We noticed that the numerical rating on the review form still appears unchanged on our side, so we just wanted to mention this in case it is simply a display or form-update issue.
> > >
> > > If there is anything else we can clarify before the discussion closes, we would be happy to help. Thank you again for your thoughtful feedback.

---

### Official Review · Reviewer_9BCY · 2026-03-17

**Soundness:** 3
**Presentation:** 3
**Significance:** 4
**Originality:** 3
**Overall Recommendation:** 3
**Confidence:** 4

**Summary:**

This paper studies instability and value collapse in offline reinforcement learning from the perspective of optimization dynamics, rather than treating collapse only as a consequence of the backup rule, network architecture, or regularization design. The main idea is that the optimizer itself can either suppress or amplify critic instability under bootstrapped temporal-difference learning. To analyze this, the paper models offline TD learning as a feedback system and derives a stability characterization for Adam-based critic updates, arguing that training remains stable when the spectral radius of the corresponding update dynamics stays below one.

Based on this analysis, the paper further argues that standard Adam updates may distort the parameter geometry in a way that worsens TD error amplification, and therefore introduces orthogonality as a stabilizing mechanism. Motivated by this view, the authors propose AdamO, an Adam-based optimizer with a decoupled orthogonality correction and an explicit task-alignment budget, so that the orthogonality update is added without directly interfering with Adam’s adaptive gradient statistics. The method is presented as a plug-and-play replacement for the critic optimizer in offline RL.

Empirically, the paper evaluates AdamO across a range of offline RL algorithms and D4RL benchmarks, including locomotion, AntMaze, and Adroit tasks. The reported results suggest that AdamO generally improves critic stability and often leads to better returns than standard Adam, while introducing only modest computational overhead. Overall, the paper’s contribution is a control-theoretic view of critic collapse, a stability analysis for Adam-style TD updates, and a practical optimizer design intended to improve the robustness of offline critic learning.

**Compliance With Llm Reviewing Policy:**

Affirmed.

**Final Justification:**

The paper proposes an interesting optimizer-level method for offline RL critic training, but its scope is already quite narrow, and even within this restricted setting the core mechanism is not cleanly established. In particular, the current empirical and theoretical analysis does not convincingly disentangle whether the observed reduction in critic loss and improvement in return come from a genuine optimizer-side benefit or from coupled secondary effects such as reduced OOD actions. This limitation is especially important because AdamO operates at the optimizer level: without a clearer understanding of when and why it helps, the method remains difficult to interpret, diagnose, and reliably use in practice. Therefore, although the idea is promising, I believe the paper still requires more systematic mechanism-focused analysis and substantial revision before its contribution can be fully justified.

**Key Questions For Authors:**

1. Is AdamO intended mainly as a complementary stabilizer, or do the authors believe it can replace some standard critic-side mechanisms such as Double Q or related stabilization components?

2. Can the authors provide more direct evidence connecting the proposed orthogonality correction to the geometric stability quantities discussed in the theory?

3. Have the authors tested AdamO under substantially larger batch sizes, in order to separate bootstrap-related instability from finite-batch optimization noise?

4. Can the authors provide more controlled experiments with frozen or partially frozen actors to better isolate critic-side optimizer effects?

5. In Mujoco locomotion, some of the IQL baseline results reported here seem lower than the numbers reported in the original IQL paper. Could the authors briefly clarify whether this difference is mainly due to implementation details, evaluation protocol, or other experimental settings?

**Limitations:**

yes

**Strengths And Weaknesses:**

I like the problem that this paper is trying to address. In offline RL, stabilizing Q-learning is a real and important challenge, and many widely used design choices, such as Double Q variants, target-network style stabilization, or more expensive backup rules, are ultimately motivated by the need to keep critic learning under control. From this perspective, the paper asks a meaningful question: whether some of this instability should also be understood from the optimizer side. I also think this angle is relatively fresh. Rather than only focusing on the backup rule, architecture, or auxiliary regularization, the paper studies whether Adam-based optimization dynamics themselves contribute to critic collapse.

Another positive aspect is that the paper is reasonably coherent in how it develops the story. The authors first provide a stability-motivated analysis, then use that analysis to motivate orthogonality, and finally instantiate the idea in AdamO. Even if I am not fully convinced by every step of this chain, I appreciate that the method is not presented as a purely ad hoc trick. The empirical section is also fairly broad: the method is evaluated across multiple offline RL backbones and benchmark families, which makes the empirical evidence more substantial than a narrowly scoped optimizer paper.

My main concern is that the practical contribution currently feels more incremental than the framing initially suggests. The paper naturally raises the expectation that, if optimizer-level stabilization is truly important, it might reduce the need for some existing critic-side stabilization machinery. However, the experiments mostly present AdamO as an additional component on top of already stabilized pipelines, rather than showing whether it can replace or simplify some of the standard tricks used in Q-learning. As a result, the current evidence supports AdamO as a potentially useful extra stabilizer, but it is less clear whether it represents a more fundamental simplification of value-based offline RL in practice.

I also found the mechanism-level evidence somewhat less convincing than the end-to-end results. The theory discusses geometry, isometry, and error amplification, while the implemented intervention is a parameter-level orthogonality correction, and the connection between these two levels is suggestive but not fully closed for me. In addition, the empirical validation chain is fairly long: AdamO is applied to the critic, but the final evidence is mostly mediated through the full actor-critic loop and environment return. Since critic loss and instability can also depend strongly on actor behavior and action-distribution shift, I would have liked more controlled critic-side evidence to isolate what is uniquely gained by the proposed optimizer. Some experimental trends, especially for standard baselines on locomotion tasks, would also benefit from a bit more clarification so that the empirical gains can be interpreted more transparently.

---

> ### Author Rebuttal · Authors · 2026-03-31
>
> **Figures/tables below:** https://anonymous.4open.science/r/ICML26-AdamO-Rebuttal-300/
>
> [1st para. of weakness]
> We agree and will narrow the claim. AdamO is not a replacement for all Q-learning stabilization. Rather, it can replace critic-side collapse-mitigation remedies, including LayerNorm, BatchNorm, WeightNorm, SpectralNorm, and DR3, while remaining complementary to Double Q, target networks/EMA, and BC-style policy constraints. Table Q1.1(a) supports this distinction, and Table Q1.1(b) shows that AdamO reduces critic blow-up and improves TD3, but BC still matters for final policy quality.
>
> [2nd para. of weakness]
> We agree that the draft does not make the mechanism chain and critic isolation explicit enough. The argument is: blockwise orthogonality correction reduces the optimizer defect $\bar R(\omega)$; as detailed in Q1.2, this tightens $\epsilon$ and $ ||\Phi^T\Phi-I||$, which feed the conditioning term behind Hurwitz stability. Empirically, Appendix A.4.5 tracks critic loss and ${Re}\,\lambda_{\max}(S)$, and Table Q1.3 shows larger batch sizes do not remove collapse. We agree, however, that frozen-actor controls are still missing, so we will present the evidence as strong but not fully causally isolated, and clarify that locomotion/IQL results are in the 10k-sample regime.
>
> [Q1.1]
> AdamO should be viewed as an optimizer-level stability package: Adam plus decoupled orthogonality correction, used with the standard ingredients in our setup (spectral/input normalization and EMA targets). In this role, it can replace several critic-side collapse-mitigation modules, such as LayerNorm, BatchNorm, WeightNorm, SpectralNorm, and DR3; Table Q1.1(a) shows these alternatives are usually weaker or less consistent. However, Double Q and policy-constraint terms such as BC address different issues, so AdamO does not replace them. Table Q1.1(b) confirms this: AdamO stabilizes the critic, while BC remains important for policy quality. (https://anonymous.4open.science/r/ICML26-AdamO-Rebuttal-300/Reviewer_9BCY/Q1.1.md)
>
> [Q1.2]
> We will make the geometry-to-implementation link explicit. Let the total blockwise orthogonality defect be $\bar R(\omega)=\sum_b R(W_b)$, and let $\omega^s$ replace each constrained block by its semi-orthogonal polar factor. Then $ ||\omega-\omega^s ||$ $\le 2\sqrt{\bar R(\omega)}$, which yields $\epsilon= ||\Psi^T\Psi-I ||<=$ $\epsilon_{ref}+c_1\sqrt{\bar R(\omega)}+c_2\bar R(\omega)$. Combined with the Gram bound, this also controls $ ||\Phi^T\Phi-I ||$; Proposition C.6 then links these quantities to the conditioning term in the Hurwitz criterion. Thus, reducing $\bar R(\omega)$ directly improves the stability quantities behind TD stability.
>
> [Q1.3]
> Yes. To separate bootstrap instability from finite-batch noise, we test substantially larger batch sizes, from 32 to 2048 (paper default 256), on pen-human. AdamO remains strong across the full range, while Adam stays collapsed at all tested batch sizes. (https://anonymous.4open.science/r/ICML26-AdamO-Rebuttal-300/Reviewer_9BCY/Q1.3.md)
>
> [Q1.4]
> Yes. We add a fully frozen-actor control. As shown in [Fig.Q1.4-a](https://anonymous.4open.science/r/ICML26-AdamO-Rebuttal-300/Reviewer_9BCY/Q1.4-a.png), critic loss differs sharply across optimizers even without actor updates; [Fig.Q1.4-b](https://anonymous.4open.science/r/ICML26-AdamO-Rebuttal-300/Reviewer_9BCY/Q1.4-b.png) shows bounded actor loss in standard training.
>
> [Q1.5].
> The difference mainly comes from the experimental setting rather than the evaluation protocol. Our MuJoCo locomotion results are reported in the 10k-sample regime, as already indicated in the paper, and we will make this clearer in the revision. For IQL, we use the standard implementation and hyperparameters from CORL [3]. Similar 10k-sample results of MuJoCo locomotion are also reported in [4,5].
>
>
>
> **Reference**
>
> [1] Yue, Y., et al. Understanding, Predicting and Better Resolving Q-Value Divergence in Offline-RL. NeurIPS 2023.
>
> [2] Kumar, A., et al. DR3: Value-Based Deep Reinforcement Learning Requires Explicit Regularization. ICLR 2022.
>
> [3] Tarasov, D., et al. CORL: Research-oriented Deep Offline Reinforcement Learning Library. 3rd Offline RL Workshop, 2022.
>
> [4] Qiao, N., et al. Less Is More: Clustered Cross-Covariance Control for Offline RL. ICLR 2026.
>
> [5] Cheng, P., et al. Look Beneath the Surface: Exploiting Fundamental Symmetry for Sample-Efficient Offline RL. NeurIPS 2023.
>
> **Due to the time and space limits of the rebuttal, we have not included every supporting detail here. If the reviewer has further questions or would like more details, please let us know, and we will be happy to continue the discussion in the discussion phase.**

---

> > ### Author Rebuttal · Reviewer_9BCY · 2026-04-02
> >
> > Thank you for the detailed rebuttal. The authors have addressed most of my concerns reasonably well, and I appreciate the additional clarifications and control experiments. My main remaining reservation is about the interpretation of the reduced critic loss: in offline RL, it is difficult to disentangle whether this comes from a genuine optimizer-side benefit of AdamO, or simply from the actor producing fewer OOD actions. I do not think the current evidence fully resolves this. In particular, Fig. Q1.4-a/b does not really make sense to me as evidence for this point. To make the claim more convincing, the authors would need either a clearer disentanglement within offline RL or evidence from online RL, where the OOD issue is much less central. That said, aside from this point, I think the rebuttal addresses the other issues well. If the authors can discuss this issue in more depth, I would be open to raising my score.

---

> > > ### Author Response · Authors · 2026-04-04
> > >
> > > Thank you for this helpful follow-up. Our claim is intentionally narrow: **the reduced critic loss mainly reflects a direct optimizer-side benefit of AdamO on critic training, while fewer OOD actions are a secondary coupled effect rather than the sole explanation**. The evidence is as follows.
> > >
> > > (1) AdamO is critic-side. In all actor-critic experiments, we change only the critic optimizer, while the actor optimizer remains standard Adam. AdamO is introduced specifically to stabilize critic-side TD training, so the first-order intervention is on the critic rather than on the actor update rule.
> > >
> > > (2) IQL offers cleaner evidence against an actor-driven explanation. IQL is especially relevant here because its critic update is SARSA-style: it is trained from dataset transitions rather than next actions sampled from the current actor. Concretely, the Q update is $y_Q=r+\gamma V_\psi(s')$ and $L_Q(\theta)=\mathbb E[(y_Q-Q_\theta(s,a))^2]$, while the value update is $L_V(\psi)=\mathbb{E}\_{(s,a)\sim \mathcal D}[L_2^\tau(Q_{\hat\theta}(s,a)-V_\psi(s))]$ with expectile loss $L_2^\tau(u)=|\tau-\mathbf{1}(u<0)|u^2$. Both Q and V are fitted from dataset samples and current value estimates, substantially weakening the usual actor-induced OOD channel in the critic. Even in this setting, Adam still yields much larger and less stable critic losses, whereas AdamO remains stable, as shown in the [$\color{#1f77b4}{\text{Fig.1.6.2a}}$](https://anonymous.4open.science/r/ICML26-AdamO-Rebuttal-300/Reviewer_9BCY/iql/iql_q_critic.png) and [$\color{#1f77b4}{\text{Fig.1.6.2b}}$](https://anonymous.4open.science/r/ICML26-AdamO-Rebuttal-300/Reviewer_9BCY/iql/iql_v_critic.png).
> > >
> > > (3) In TD3+BC, the coupling is not only actor-to-critic, but also critic-to-actor. In TD3+BC, the Q term and BC term in the actor objective are intentionally kept on comparable scales through the coefficient $\alpha$, and the gradient-domain visualizations confirm this near-proportional design in practice. See the [$\color{#1f77b4}{\text{Fig.1.6.3a}}$](https://anonymous.4open.science/r/ICML26-AdamO-Rebuttal-300/Reviewer_9BCY/domain/td3bc_bc_domain.png) and [$\color{#1f77b4}{\text{Fig.1.6.3b}}$](https://anonymous.4open.science/r/ICML26-AdamO-Rebuttal-300/Reviewer_9BCY/domain/td3bc_q_domain.png). However, if the critic collapses, the learned Q values can become extremely large and disordered. That corrupted Q signal then perturbs actor updates even when BC is present. Therefore, fewer OOD actions can itself be a downstream consequence of a more stable critic, rather than the primary cause of the reduced critic loss. This is why we interpret the reduced critic loss as mainly a direct critic-side benefit of AdamO, while any reduction in OOD actions is a secondary coupled effect.
> > >
> > > (4) The [$\color{#1f77b4}{\text{Table Q4.3}}$](https://anonymous.4open.science/r/ICML26-AdamO-Rebuttal-300/Reviewer_j5yV/Q4.3.md) ablation shows that AdamO complements BC rather than replacing it. Removing BC while keeping AdamO yields a more stable critic, but the final return is still poor. Thus AdamO and BC/CQL address different problems: AdamO stabilizes critic optimization, whereas BC/CQL-style terms constrain policy extrapolation.
> > >
> > > (5) The trend remains under partial freezing. We also varied the actor-update frequency to obtain intermediate controls between standard training and full freezing. Here, frozen frequency ={$1,100,10000$} means one actor update every {$1,100,10000$} critic updates, respectively. Across all settings, AdamO consistently yields lower and more stable critic loss than Adam, so the effect is not tied to a single extreme control. This robustness is shown in the [$\color{#1f77b4}{\text{Fig.1.6.5}}$](https://anonymous.4open.science/r/ICML26-AdamO-Rebuttal-300/Reviewer_9BCY/frozen-freq/td3bc_frozen_freq_three_panel.png).
> > >
> > > (6) Fig. Q1.4-a is the cleanest isolation. In [$\color{#1f77b4}{\text{Fig. Q1.4-a}}$](https://anonymous.4open.science/r/ICML26-AdamO-Rebuttal-300/Reviewer_9BCY/Q1.4-a.png), the actor is fully frozen, so the action distribution never changes. AdamO therefore cannot help by gradually inducing fewer OOD actions because there is no actor adaptation. Under this fixed distribution, Adam diverges while AdamO remains stable. We will release the detailed reproduction code for this control after the rebuttal period.
> > >
> > > AdamO is designed for offline RL critics, where the dataset is fixed. Online RL introduces additional replay nonstationarity, so whether AdamO can also resolve critic collapse there remains future work.
> > >
> > > Perfect causal separation is difficult in offline RL because actor and critic are coupled. But the evidence does not support a purely actor-driven explanation. Across the IQL evidence, TD3+BC analysis, BC-removal ablation, partial-freezing controls, and frozen-actor control, **the reduced critic loss is best explained by a genuine critic-side optimizer benefit of AdamO**. Fewer OOD actions remain necessary for strong offline RL performance.

---

### Decision · Program_Chairs · 2026-04-30

**Decision:**

Accept (regular)

**Comment:**

The paper offers a novel and empirically effective optimizer for offline RL. However, causal isolation of the optimizer’s direct benefit (9BCY) and rigorous theoretical grounding of the proposed mechanism (VrAN) remain partially unresolved, limiting full confidence in the fundamental claims.

In general, i feel this is a good paper that is promising but not very strong in the current form.
The paper would benefit from stronger mechanistic evidence and tighter theoretical linking before its contribution can be considered fully established.

Despite the limitations, almost all the reviewers see good potential in this paper, and thus i recommend this paper for acceptance.